# ARCUEID: MULTI-TRIGGER CLOUD SHAPING FOR UNIFIED BACKDOOR ATTACK PARADIGMS

## ABSTRACT

Machine learning have driven breakthroughs in recognition, detection, and generation, yet their increasing ubiquity also exposes them to backdoor attack hazards, threatening the security of real-world AI deployments. Existing backdoor methods, however, remain fragile in adaptive settings for **rigid dependency on a static trigger**, **narrow scope in fixed one-to-one mappings**, or **unrealistic assumptions for levels of access**, thereby failing to scale to dynamic, large-class scenarios under realistic constraints. Therefore, we present Arcueid, a theoretically grounded multi-trigger backdoor framework that **achieves scalable and robust attacks across** $M \mapsto M$**,** $M \mapsto N$**, and** $M \mapsto 1$ **paradigms**. It operates under restrictive settings, **requiring only black-box knowledge and extremely low poisoning budgets**. At its core lies a *Joint Cloud Shaping Multi-trigger Optimization* strategy that simultaneously compacts trigger-induced feature clouds and enforces inter-cloud separation, ensuring stable, non-interfering, and target-consistent decision regions, while decoupling trigger generation from label mapping to enable dynamic reconfiguration of targets and robust transferability across models and datasets. Extensive experiments on multiple datasets and five CNN/transformer architectures show that Arcueid attains near-perfect average ASR ($> 97\%$) across targets in each paradigm with negligible clean accuracy drop ($< 5\%$) even at poisoning rates of $0.1\%$, significantly outperforming SOTA baselines. Moreover, Arcueid consistently withstands representative pre-/mid-/post-training defenses, exhibits strong stealth with indistinguishable perceptual shifts, and sustains steady resilience across comprehensive ablation studies.

## 1 INTRODUCTION

Machine learning has advanced rapidly with deep neural networks, from convolutional architectures to transformers, driving progress in recognition (Crowley, 2010), detection (Nassif et al., 2021), and generation (Summerville et al., 2018). Yet it faces growing threats from *backdoor attacks* (Gu et al., 2019), where models behave normally on benign inputs but misclassify those stamped with secret triggers into attacker-specified targets. The covert nature of such attacks poses serious risks to the security and trustworthiness of real-world AI systems (Chen et al., 2024).

Although backdoor research has made notable strides, such as clean-label poisoning (Turner et al., 2019), invisible perturbations (Zeng et al., 2023), and adaptive trigger generation (Qi et al., 2023a), most progress remains centered on crafting elaborate trigger pattern or switching application scenarios. However, with the rapid development of detection and mitigation defenses (Hou et al., 2024a; Li et al., 2021a), these conventional designs increasingly struggle to remain effective. Much less attention has been given to expanding the attack scope and adaptivablity to dynamic, large-scale scenarios. Consequently, existing techniques face practical limitations:

- **L1: Rigid Dependency.** A large fraction of existing attacks hinge on a single well-chosen perturbation pattern embedded across poisoned samples (Mengara et al., 2024). Such rigidity greatly simplifies the defender's task: once the trigger is detected or suppressed, the attack collapses entirely (Li & Liu, 2024). Moreover, a single universal pattern cannot adapt to heterogeneous input or task-specific conditions, making it brittle in dynamic or adversarially monitored environments.

- **L2: Narrow Attack Scope.** Most backdoor attacks enforce a fixed mapping between a trigger and a designated target label. This narrow design severely constrains the attacker's influence:

poisoned samples always converge to the same class regardless of their origin. Even classical all-to-all extensions, such as cyclic mappings (Nguyen & Tran, 2021), remain structurally rigid, while recent multi-target variants (Hou et al., 2024b) only scale to a handful of classes. Such constraints render existing approaches ineffective for realistic broad-class settings (Shen et al., 2024) or adaptive scenarios (Essa et al., 2023) where targets must change on demand.

- **L3: Unrealistic Privileged Assumptions.** Existing attacks often rely on high-privilege conditions, such as full control of the victim's training pipeline (Nguyen & Tran, 2020), white-box access to model gradients and structures (Souri et al., 2022), or direct weight modification (Chen et al., 2021). Other designs require an excessively high poisoning rate to maintain effectiveness. These assumptions stand in stark contrast to practical threat models like Machine-Learning-as-a-Service (MLaaS) (Ribeiro et al., 2015) or supply-chain distribution (Ni et al., 2020), where adversaries have limited access and must remain stealthy under strong defensive monitoring.

To this end, as shown in Figure 1, static-design backdoor attacks remain fragile in adaptive scenarios, lacking effectiveness against dynamic target mappings. We aim to realize a multi-trigger backdoor attack spanning multiple paradigms under black-box knowledge. The key challenges and corresponding solutions are formalized as follows:

**C1: How to keep trigger optimization orthogonal to diverse attack paradigms while enabling flexible goals? (L1 + L2)**

**S1:** We design a theoretically grounded and efficient trigger optimization mechanism decoupled from target mapping. The optimization depends only on the number of triggers, making the framework naturally compatible with different paradigms.

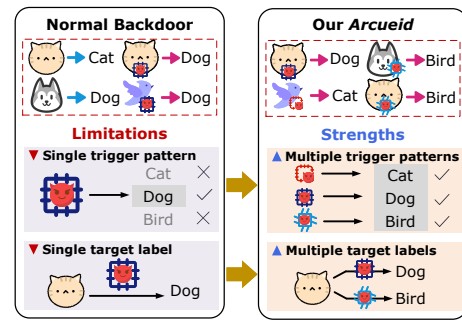

Figure 1: General comparison between normal backdoor attacks and Arcueid.

**C2: How to remain effective under low poisoning budgets and avoid feature collisions in multi-trigger optimization? (L1 + L3)**

**S2:** We propose a *Joint Cloud Shaping* mechanism that jointly minimizes intra-trigger variance and maximizes inter-trigger separation. This ensures that triggers form compact yet distinct clusters in the representation space, preserving attack stability and effectiveness even with very low poisoning.

**C3: How to guarantee transferability under model- and data-agnostic conditions? (L2 + L3)**

**S3:** We leverage surrogate training on non-IID subsets, where the surrogate model functions as a feature extractor, and optimize triggers in a representation-consistent manner. Specifically, the optimization enforces that trigger-induced features preserve relative geometry in the latent space, rather than relying on model-specific decision boundaries. By anchoring triggers to stable feature distributions, this design enables robust generalization across different models and datasets, even when the adversary lacks knowledge of the victim's architecture or training pipeline.

Therefore, our work makes the following contributions:

- We present Arcueid, a theoretically grounded multi-trigger-driven backdoor attack framework that scales to complex $M \mapsto M$, $M \mapsto N$, and $M \mapsto 1$ paradigms while operating under black-box knowledge and extremely low poisoning budgets. Our code will be released upon publication.

- We design a novel *Joint Cloud Shaping Multi-trigger Optimization* mechanism that decouples trigger generation from target mapping, ensuring orthogonality between paradigm mapping and trigger optimization. This enables stable, separable, and dynamic trigger–target associations, requiring no knowledge of the victim's training process.

- We conduct extensive experiments on multiple benchmark datasets across five mainstream architectures, demonstrating that Arcueid consistently achieves near-perfect average ASR across all targets and paradigms with exceeding low accuracy degradation (mostly $< 5\%$) even under extremely low poisoning rates ($0.1\%$), outperforms SOTA attack baselines, exhibits strong robustness against pre-, mid-, and post-training defenses, attains favorable stealthiness with imperceptible perceptual shifts, and shows stable resilience through comprehensive ablation studies.

## 2 PRELIMINARY & RELATED WORKS

Backdoor learning is a malicious training paradigm in which an adversary injects hidden behaviors into a machine learning model by manipulating its training data (Li et al., 2024). Specifically, let $\mathcal{D}_{\text{benign}} = \{(x_i, y_i)\}_{i=1}^S$ denote the clean training dataset, which is used to train a model $f_\theta$ with parameters $\theta$. The attacker constructs a poisoned dataset $\mathcal{D}_{\text{poison}} = \{(x'_j, y'_j)\}_{j=1}^P$, where each $x'_j = g_\eta(x_j)$ embeds a trigger pattern parameterized by $\eta$, and the assigned label $y'_j$ depends on the attack mapping. The overall training set becomes $\mathcal{D}_{\text{train}} = \mathcal{D}_{\text{benign}} \cup \mathcal{D}_{\text{poison}}$. The training objective is to learn a model $f_{\theta'}$ that retains high accuracy on clean samples, but is forced to misbehave when presented with triggered inputs. **Backdoor attacks** instantiate this paradigm by defining specific trigger-label mappings, which fall into two categories: *all-to-one* and *all-to-all*.[1]

**All-to-one Backdoor Attacks.** In this setting, all poisoned inputs share a trigger and are relabeled to a fixed target. BadNets (Gu et al., 2019) first demonstrated this threat, while Blended (Chen et al., 2017) extended it with stealthy, physically realizable triggers. Subsequent works enhanced stealth and robustness through physical-world adaptability (Li et al., 2021c), spectral-domain optimization (Li et al., 2021e), latent regularization (Qi et al., 2023a), and clean-label poisoning under limited knowledge (Zeng et al., 2023; Feng et al., 2025). This paradigm is widely studied for its simplicity, though its fixed objectives and limited behavioral diversity restrict flexibility.

**All-to-all Backdoor Attacks.** Originally introduced by BadNets (Gu et al., 2019), this setting maps each class $y$ to a cyclic target $\tau(y)$ with a shared trigger, distributing misclassifications across classes. WaNet (Nguyen & Tran, 2021) and LIRA (Doan et al., 2021) explored stealthy designs with invisible warping and instance-specific optimization, while Input-aware attacks (Nguyen & Tran, 2020) dynamically controlled trigger and label mappings. Modern variants adopt multiple triggers $\{g_{\eta_k}\}$, each tied to a designated target, enabling many-to-many mappings. One-to-N and N-to-One paradigms (Xue et al., 2022) showed high success with low degradation, Marksman (Doan et al., 2022) generated class-conditional triggers for arbitrary targets, and M2N (Hou et al., 2024b) extended this to $M$ triggers targeting $N$ classes. Despite this progress, strong attacker assumptions (e.g., white-box access or high poisoning rates) still limit real-world applicability.

## 3 THREAT MODEL

It is a common practice for model trainers to download data from public sources for training, typically without rigorous scrutiny of its source or integrity (Li et al., 2024). This creates a critical vulnerability, as it enables adversaries to easily propagate poisoned data by distributing it through these same channels. The adversary's primary objective is to manipulate the backdoor-trained model as a controllable agent, effectively turning it into a *puppet*. Unlike traditional attack paradigm, our `Arcueid` dynamically optimizes multiple triggers and flexibly maps them to arbitrary target labels, enabling a scalable and adaptive multi-paradigm backdoor attack. This flexible control facilitates dynamic and far-reaching post-deployment attacks, surpassing not only classical all-to-one backdoor scenarios but even SOTA all-to-all and multi-target settings. Under such a threat model, the adversary essentially gains an undetectable access channel to the victim's deployed models, with the ability to subvert core model-based functionalities at will.

### 3.1 VICTIM ASSUMPTION

The victim, typically an entity aiming to construct large-scale, high-performing models, assembles training datasets by collecting data from public sources. Common practices include Internet scraping (Valova et al., 2023) and open repositories (Prior et al., 2020), or by relying on third-party data vendors Zheng et al. (2019). As described above, this open and potentially untrusted data collection pipeline introduces a possible risk of data poisoning.

Recognizing this threat, the victim may adopt a multi-stage defense: (i) **Pre-training**, detecting and filtering suspicious data; (ii) **Mid-training**, employing data purification or augmentation to counter malicious influence; and (iii) **Post-training**, conducting mitigation or model audits when a backdoor is suspected. Such layered defenses substantially raise the bar for a successful attack.

---

[1]Backdoor defenses are deferred to Appendix A.1.

## 3.2 ADVERSARY ASSUMPTION

In our attack scenario, the adversary may consist of a single attacker or multiple coordinated attackers, each with distinct attack objectives and corresponding target classes. Attackers can independently embed unique trigger patterns and assign arbitrary target labels, enabling diverse backdoors that operate concurrently within the same model. This reflects a realistic threat: multiple parties may attempt for varied post-deployment goals. To comprehensively evaluate our approach, we consider the adversary in a highly practical and constrained black-box setting, where **no internal information** about the *victim's model architecture*, *training data*, or *learning dynamics is accessible*. This assumption reflects realistic threats encountered in open or outsourced data collection processes.

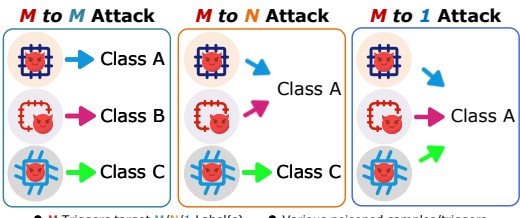

Figure 2: Threat paradigm configurations: $M \mapsto M$, $M \mapsto N$, and $M \mapsto 1$ attacks.

**Adversary Knowledge.** The adversary operates under a black-box assumption: they have **no access** to the victim's proprietary training dataset $\mathcal{D}_{\text{train}}$, model architecture $f$, or specific training procedures. The only information available is awareness that the victim intends to train a model for a certain task (e.g., image classification). Additionally, No details about data distribution, preprocessing, or defense mechanisms are revealed to the adversary. This setting also accommodates multiple conspiratorial attackers, who share the same limited resources, such as surrogate model or surrogate dataset, and collectively aim to inject effective backdoors without any insider access.

**Adversary Capability.** Despite possessing only limited knowledge, the adversary is provisioned with a small, local surrogate dataset $\mathcal{D}_{\text{sur}}$ and a surrogate model $f_{\text{sur}}$. $\mathcal{D}_{\text{sur}}$ is drawn from a *non-IID* and *completely disjoint* distribution relative to $\mathcal{D}_{\text{train}}$, which is used to model feature representations for trigger optimization. $f_{\text{sur}}$ is architecturally distinct from the victim model $f$. It is treated as *model-agnostic*, meaning the attacker makes no assumptions about architectural alignment, model capacity, or shared inductive biases with the target model. In addition, the scale of $\mathcal{D}_{\text{sur}}$ is significantly smaller than $\mathcal{D}_{\text{train}}$ (i.e., typically containing only 5,000 to 15,000 samples, $|\mathcal{D}_{\text{sur}}| \ll |\mathcal{D}_{\text{train}}|$), imposing further practical constraints on the adversary's resources.

With only these limited surrogate resources, the adversary performs a learnable trigger optimization process, using $\mathcal{D}_{\text{sur}}$ and $f_{\text{sur}}$ to design a set of triggers $\{g_{\eta_k}\}_{k=1}^{K}$ and to assign corresponding target labels based on desired attack behavior. Depending on the attacker's objectives, the mapping between triggers and targets can follow three representative configurations, as illustrated in Figure 2.

- $M \mapsto M$ **Attack.** Each trigger $g_{\eta_k}$ is mapped to a unique target label $\tau_k$, resulting in $M$ distinct target classes. This simulates scenarios where an adversary seeks to fully hijack class-level predictions, maximizing the coverage of misclassification across the label space.

- $M \mapsto N$ **Attack.** The $M$ triggers are mapped to $N < M$ target classes, allowing multiple triggers to share targets. This models coordinated attacks where multiple patterns converge to a subset of malicious outputs, increasing control density while maintaining diversity in trigger design.

- $M \mapsto 1$ **Attack.** All $M$ triggers are assigned to one target class. This configuration is highly applicable in binary or security-sensitive tasks where the adversary aims to redirect all triggered samples to one specific outcome, offering robustness via multiple attack pathways.

The adversary then constructs a poisoned subset $\mathcal{D}_{\text{poison}}$ by stamping triggers $g_{\eta_k}$ onto clean inputs and relabeling them accordingly [2]. This poisoned data is injected into the victim's data collection pipeline (e.g., via open submission platforms or third-party sharing), such that when integrated into $\mathcal{D}_{\text{train}}$, it induces the intended dynamic multi-paradigm backdoor attack behavior, despite significant differences in data domain and model architecture between the surrogate and target environments.

---

[2]We support clean-label attacks (Seen in Table 4 in Appendix A.2), but mainly assume adversaries lack control over label distributions in practice.

## 4 METHODOLOGY

### 4.1 PROBLEM DEFINITION

**Notations.** Let $(x, y) \sim \mathcal{D}$ be clean data with label set $\mathcal{Y} = \{1, \ldots, Q\}$. A classifier $f_\theta : \mathcal{X} \to \Delta^{Q-1}$ induces decision regions $\mathcal{R}_c := \{x \in \mathcal{X} : \arg\max f_\theta(x) = c\}$. For analysis, we factor $f_\theta = h \circ \phi_\theta$ with representation $\phi_\theta : \mathcal{X} \to \mathcal{Z} \subset \mathbb{R}^d$ and head $h : \mathcal{Z} \to \Delta^{Q-1}$. Unless otherwise stated, all norms are $\ell_2$ and dist denotes the induced metric, the same conventions apply in $\mathcal{Z}$.

A *trigger* is a parametric map $g_\eta : \mathcal{X} \to \mathcal{X}$. We consider a family $G = \{g_{\eta_k}\}_{k=1}^{K}$ under budgets $\|g_{\eta_k}(x) - x\|_\infty \leq \varepsilon$ and $\|g_{\eta_k}(x) - x\|_0 \leq s$. The attacker specifies (i) a *routing* rule $\pi$, which decides which trigger $k \in \{1, \ldots, K\}$ is applied to a given benign sample, and (ii) a *trigger–target map* $\sigma : \{1, \ldots, K\} \to \mathcal{T} \subseteq \mathcal{Y}$ that assigns targets. These are specified independently: triggers are optimized in feature space, while $\sigma$ determines the desired misclassification behavior. We write $\tau(y) = \sigma(\pi(y))$ and denote $\tau_k = \sigma(k)$. During poisoning, a scheduler flips a fraction $\rho \in [0, 1]$ of training samples, stamping $x' = g_{\eta_{\pi(y)}}(x)$ and relabeling to $\tau(y)$, yielding $\mathcal{D}_{\text{poison}}$.

**Attack Paradigms.** As defined in Section 3.2, we fix the number of triggers to match the number of active sources, i.e., $K = M$ with $M \leq Q$. Backdoor attack paradigms are then instantiated by specifying the trigger–target mapping $\sigma : \{1, \ldots, K\} \to \mathcal{T}$ and the target set size $|\mathcal{T}|$:

- $M \mapsto M$**:** $\sigma$ is a permutation over $\mathcal{Y}$, assigning each trigger to a unique target class ($|\mathcal{T}| = M$).

- $M \mapsto N$ $(N < M)$**:** several triggers map to the same target, yielding a target set of size $N$.

- $M \mapsto 1$**:** all triggers collapse to a single target $t^\star$, i.e., $\sigma(k) = t^\star$ for all $k$.

The routing $\pi$ determines which trigger is applied to each sample but is otherwise unconstrained: it may assign distinct triggers, share triggers across groups, or mix both strategies.

**Representation Space Feasibility.** Define the decision margin of a set $A \subseteq \mathcal{X}$ to class $t$ as

$$\text{margin}_t(A) := \inf_{x \in A} \text{dist}\big(x, \partial\mathcal{R}_t\big), \tag{1}$$

with $\partial\mathcal{R}_t$ the decision boundary of class $t$. When using $\phi_\theta$, interpret dist in $\mathcal{Z}$. For trigger $k$, let the *triggered cloud*

$$\mathcal{C}_k := \{\phi_\theta(g_{\eta_k}(x)) : (x, y) \sim \mathcal{D}, \pi(y) = k\}, \tag{2}$$

have center $\mu_k$ and radius $r_k$ computed in $\mathcal{Z}$. Distances to decision regions are evaluated in $\mathcal{Z}$ via the induced regions $\mathcal{R}_c = \{z \in \mathcal{Z} : \arg\max h(z) = c\}$.

The following propositions and lemmas establish the conditions under which triggered clouds are feasible, mutually non-interfering, and transferable across models.

**Proposition 1 (Feasibility via Interior Placement).** *If each triggered cloud $\mathcal{C}_k$ enjoys a positive margin $\text{margin}_{\tau_k}(\mathcal{C}_k) \geq \gamma_k > 0$, then every point in $\mathcal{C}_k$ is classified as its designated target $\tau_k$. A sufficient condition is*

$$\text{dist}\big(\mu_k, \partial\mathcal{R}_{\tau_k}\big) > r_k. \tag{3}$$

**Lemma 1 (Non-interference of Triggered Clouds).** *Let $k \neq \ell$. If $\text{margin}_{\tau_k}(\mathcal{C}_k) \geq \gamma_k$, $\text{margin}_{\tau_\ell}(\mathcal{C}_\ell) \geq \gamma_\ell > 0$, and the centers satisfy $\text{dist}(\mu_k, \mu_\ell) > r_k + r_\ell$, then $\mathcal{C}_k$ and $\mathcal{C}_\ell$ occupy disjoint interiors of $\mathcal{R}_{\tau_k}$ and $\mathcal{R}_{\tau_\ell}$, so predictions remain stable and non-overlapping.*

**Lemma 2 (Clean Accuracy Stability under Small Poisoning).** *Suppose the training algorithm is uniformly $\beta$-stable with respect to single-example replacement and the loss is bounded by $L_{\max}$. Replacing a $\rho$-fraction of training samples with poisoned ones perturbs the expected clean risk by at most $O(\beta\rho) + \rho L_{\max}$. Thus, when $\rho$ is small and training is stable, clean accuracy degradation remains limited.*

**Proposition 2 (Transferability under Representation Drift).** *Let a surrogate $f_s = h_s \circ \phi_s$ and a target $f_t = h_t \circ \phi_t$ satisfy a bi-Lipschitz alignment on the triggered support: $\|\phi_t(x) - A\phi_s(x)\| \leq \delta$ for some bounded linear $A$, and assume $h_t$ is $L_h$-Lipschitz. If Proposition 1 holds for $f_s$ with margin $\gamma$, then $f_t$ preserves the same backdoor decisions provided*

$$L_h \|A\| \delta < \gamma. \tag{4}$$

**Lemma 3** (**Identifiability under Limited Knowledge**). *Assume class-conditional features $\phi_\theta(x) \mid (y = c)$ are sub-Gaussian with mean $\bar{\mu}_c$. Given $n_c$ samples per class, the empirical mean $\hat{\mu}_c$ satisfies $\|\hat{\mu}_c - \bar{\mu}_c\| = O_p(n_c^{-1/2})$. Hence constraints phrased in terms of true centroids $\bar{\mu}_c$ (e.g., placing $\mu_k$ with margin $\gamma$ inside $\mathcal{R}_{\tau_k}$ and outside neighborhoods of clean centroids) remain estimable with finite samples, enabling optimization under limited data/model access.*

**Optimization Problem Induced by Feasibility.** The feasibility analysis above provides constructive conditions: each triggered cloud must (i) lie strictly inside its designated region $\mathcal{R}_{\tau_k}$, and (ii) remain separated from other clouds to avoid cross-trigger interference.

These geometric requirements naturally translate into a constrained optimization problem, where trigger parameters $\eta_{1:K}$ (and optionally the routing $\pi$) are optimized while the victim parameters $\theta$ are learned on the poisoned mixture. Formally:

$$
\min_{\eta_{1:K}, \pi} \quad \mathcal{R}_{\text{clean}}(f_\theta; \mathcal{D}) \; - \; \lambda_{\text{ASR}} \, \mathbb{E}_{(x,y)\sim\mathcal{D}}\Big[\mathbf{1}\{\arg\max f_\theta(g_{\eta_{\pi(y)}}(x)) = \tau(y)\}\Big]
$$

$$
+ \; \lambda_{\text{stealth}} \sum_{k=1}^{K} \mathbb{E}\big[\|g_{\eta_k}(x) - x\|\big] \; + \; \lambda_{\text{int}} \, \Psi\big(\{\mathcal{C}_k, \tau_k\}_{k=1}^{K}\big), \tag{5}
$$

$$
\text{s.t.} \quad \rho \leq \rho_{\max}, \quad \|g_{\eta_k}(x) - x\| \leq \varepsilon, \; \forall k, x,
$$

$$
\text{dist}\big(\mu_k, \partial\mathcal{R}_{\tau_k}\big) \; \geq \; r_k + \gamma_{\min} \quad \text{(margin)},
$$

$$
\text{dist}(\mu_k, \mu_\ell) \; \geq \; r_k + r_\ell + \delta_{\min}, \quad \forall\, k \neq \ell \quad \text{(non-interference)}.
$$

Here, $\Psi$ penalizes violations of the margin and separation constraints, e.g., via hinge penalties on center-to-boundary and center-to-center distances. This formulation highlights three key properties:

- **Universal.** Independent of the particular loss or model architecture.
- **Paradigm-agnostic.** Covers $M \mapsto M$, $M \mapsto N$, and $M \mapsto 1$ paradigms instantiation via $\sigma$ and $\pi$.
- **Budget-aware.** Limited by stealth ($\varepsilon$), poison rate ($\rho_{\max}$), and robustness margins ($\gamma_{\min}, \delta_{\min}$).

## 4.2 Joint Cloud Shaping Multi-trigger Optimization

To instantiate the optimization program from Section 4.1, we propose *Joint Cloud Shaping Multi-trigger Optimization* where employs two structure terms: (A) **intra-cloud compactness** and (B) **inter-cloud separation** to learns $\eta_{1:K}$ from random initialized trigger set $\{g_{\eta_k}\}_{k=1}^{K}$.

**Invisible Trigger Design.** Each trigger $g_{\eta_k}$ is realized as a masked blend with a fixed sparse mask $\alpha_k \in [0,1]^{C \times H \times W}$ satisfying $\|\alpha_k\|_0 \leq s$ and a learnable pattern $v_k \in \mathcal{X}$, i.e.,

$$
g_{\eta_k}(x) = \text{clip}\big((1 - \alpha_k) \odot x + \alpha_k \odot v_k\big), \quad \Delta_k(x) = \alpha_k \odot (v_k - x).
$$

This enforces $\|\Delta_k(x)\|_0 \leq s$ by construction, and we impose $\|\Delta_k(x)\|_\infty \leq \varepsilon$ via clipping. Gradients update $v_k$ only, and $\alpha_k$ remains fixed.

We first define the empirical center and radius in the representation space:

$$
\mu_k \; = \; \frac{1}{|\mathcal{B}_k|} \sum_{(x_i, y_i) \in \mathcal{B}_k} \tilde{z}_i^{(k)}, \qquad r_k^2 \; = \; \frac{1}{|\mathcal{B}_k|} \sum_{(x_i, y_i) \in \mathcal{B}_k} \big\|\tilde{z}_i^{(k)} - \mu_k\big\|^2. \tag{6}
$$

where $\mathcal{B}_k = \{(x_i, y_i) \in \mathcal{B} : \pi(y_i) = k\}$ be samples routed to trigger $k$ for a minibatch $\mathcal{B}$. $z_i = \phi_\theta(x_i)$ and $\tilde{z}_i^{(k)} = \phi_\theta(g_{\eta_k}(x_i))$ for clean and triggered features via a fixed classifier $f_\theta = h \circ \phi_\theta$.

**(A) Intra-cloud Compactness.** By Proposition 1, feasibility requires each cloud to remain entirely inside its designated region. We therefore minimize within-cloud variance so that triggered samples cluster tightly around $\mu_k$:

$$
\mathcal{L}_{\text{intra}} \; = \; \frac{1}{K} \sum_{k=1}^{K} \frac{1}{|\mathcal{B}_k|} \sum_{(x_i, y_i) \in \mathcal{B}_k} \big\|\tilde{z}_i^{(k)} - \mu_k\big\|^2. \tag{7}
$$

This term reduces the radius $r_k$, directly improving the margin of $\mathcal{C}_k$ relative to its target boundary.

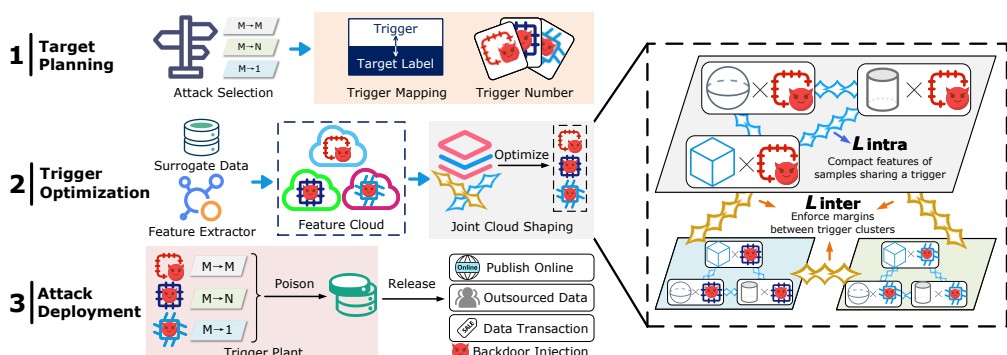

Figure 3: Overview of `Arcueid`, illustrating framework execution structures.

**(B) Inter-cloud Separation.** As shown in Lemma 1, avoiding cross-trigger interference requires triggered clouds to remain apart. We enforce a pairwise margin $m > 0$ between centers by penalizing violations with a hinge loss:

$$\mathcal{L}_{\text{inter}} = \frac{2}{K(K-1)} \sum_{1 \le k < \ell \le K} \left[ m - \|\mu_k - \mu_\ell\| \right]_+. \tag{8}$$

This repulsive force ensures that different triggers carve out distinct, non-overlapping decision regions, thereby stabilizing multi-trigger coexistence.

**Overall Optimization Objective.** Combining both terms, we optimize only $\eta_{1:K}$ while keeping the victim classifier $\theta$ frozen:

$$\min_{\eta_{1:K}} \quad \lambda_{\text{intra}}\mathcal{L}_{\text{intra}} + \lambda_{\text{inter}}\mathcal{L}_{\text{inter}} \quad \text{s.t.} \quad \|g_{\eta_k}(x) - x\| \le \varepsilon \quad (\forall k, x). \tag{9}$$

Gradients flow through $\phi_\theta \circ g_{\eta_k}$ to update triggers, while the classifier remains fixed. A detailed analysis of convergence and pseudocode is provided in the Appendix A.3.

### 4.3 ARCUEID: ATTACK WORKFLOW

Figure 3 depicts the three-stage pipeline of `Arcueid`, with each modular stage:

**Stage 1. Target Planning.** The adversary configures the attack by determining the target classes, selecting the attack paradigm ($M \mapsto M$, $M \mapsto N$, or $M \mapsto 1$, as defined in Section 3.2), and outlining a preliminary trigger–target mapping along with the number of triggers $K$ required for optimization. These choices are made independently of the subsequent optimization stage.

**Stage 2. Trigger Optimization.** The adversary initializes $K$ triggers at random and optimizes their parameters $\eta_{1:K}$ on a surrogate dataset and model using the *Joint Cloud Shaping Multi-trigger Optimization* mechanism described in Section 4.2.

**Stage 3. Attack Deployment.** The adversary uniformly poisons a fraction $\rho$ of benign samples with the optimized $K$ triggers, relabels them according to the pre-determined paradigm and target mappings from Stage 1, and injects the resulting mixture into the victim's training pipeline through channels such as online publication, outsourced datasets, or data trading.

## 5 EVALUATION

### 5.1 EXPERIMENT SETUP

*General Settings:* We conduct experiments on three widely-used image classification benchmarks: CIFAR-10/100 (Krizhevsky & Hinton, 2009), and TinyImageNet (Le & Yang, 2015). For CNNs, we adopt ResNet-18/34 (He et al., 2016), and VGG13-BN (Simonyan & Zisserman, 2015) as representative backbones. To further test robustness across architectures, we also include transformer-based models, namely ViT (Dosovitskiy et al., 2021) and SimpleViT (Beyer et al., 2022).

Table 1: **Attack performance (ΔACC/ASR±Std) on various models under attack paradigms**. Here, $M \mapsto N$ denotes an attack configuration where $M$ is the number of triggers and $N$ is the number of target classes chosen by the adversary.

| Dataset | $M \mapsto N$ | ResNet-18 | | ResNet-34 | | VGG13-BN | | ViT | | SimpleViT | |
|---|---|---|---|---|---|---|---|---|---|---|---|
| | | ΔACC | ASR | ΔACC | ASR | ΔACC | ASR | ΔACC | ASR | ΔACC | ASR |
| CIFAR-10 (PR=0.1%) | 10→1 | 5.5% | 99.1%±0.7% | 2.8% | 100.0%±0.0% | 2.1% | 100.0%±0.0% | 0.3% | 99.4%±0.5% | -0.3% | 100.0%±0.0% |
| | 10→2 | 1.6% | 99.9%±0.1% | 3.7% | 99.4%±0.8% | 1.9% | 99.4%±0.4% | 0.3% | 96.8%±1.6% | 0.4% | 99.7%±0.3% |
| | 10→5 | 1.4% | 99.6%±0.3% | 3.9% | 99.7%±0.3% | 1.9% | 98.6%±1.1% | 0.5% | 93.9%±2.6% | -0.1% | 99.7%±0.2% |
| | 10→10 | 1.6% | 99.8%±0.2% | 4.7% | 98.8%±1.0% | 2.0% | 98.8%±0.9% | 0.3% | 81.5%±9.3% | 0.2% | 92.0%±3.4% |
| CIFAR-100 (PR=1%) | 100→1 | 3.0% | 100.0%±0.0% | 7.2% | 100.0%±0.0% | 5.1% | 99.9%±0.1% | -0.4% | 99.9%±0.1% | -0.2% | 100.0%±0.1% |
| | 100→5 | 2.9% | 97.0%±2.7% | 3.6% | 99.6%±0.4% | 5.2% | 86.6%±7.4% | 0.2% | 94.3%±3.8% | -0.3% | 97.0%±2.0% |
| | 100→10 | 3.2% | 96.6%±2.3% | 2.7% | 99.8%±0.3% | 2.7% | 98.8%±1.0% | -0.5% | 94.5%±3.2% | 0.4% | 97.4%±1.9% |
| | 100→100 | 3.7% | 98.1%±1.4% | 7.7% | 88.2%±6.7% | 3.2% | 95.3%±2.3% | -0.5% | 80.4%±10.2% | 0.6% | 84.2%±9.1% |
| TinyImageNet (PR=2%) | 200→1 | 6.8% | 100.0%±0.0% | 8.0% | 100.0%±0.0% | 4.6% | 100.0%±0.0% | 0.7% | 99.9%±0.2% | 1.3% | 99.9%±0.2% |
| | 200→2 | 7.2% | 99.7%±0.4% | 8.8% | 99.9%±0.2% | 5.1% | 99.5%±0.5% | 1.3% | 97.7%±1.9% | -0.1% | 96.9%±2.6% |
| | 200→4 | 7.5% | 99.9%±0.2% | 9.1% | 100.0%±0.1% | 4.0% | 99.3%±0.5% | 1.5% | 96.8%±2.1% | 0.3% | 95.9%±2.4% |
| | 200→200 | 6.2% | 99.9%±0.1% | 7.2% | 99.9%±0.1% | 8.2% | 98.7%±1.3% | 1.0% | 92.7%±3.6% | 0.9% | 89.4%±5.0% |

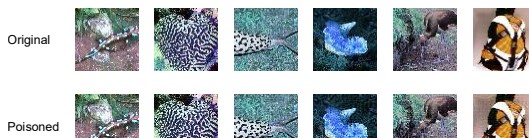

Original

Poisoned

Figure 4: Visualization of Arcueid

*Evaluation Metrics:* We adopt the following:

**CA (Clean Accuracy)** – Clean inputs accuracy.
**ASR (Attack Success Rate)** – Proportion of trigger-embedded inputs classified into targets.
**ΔACC** – Drop in CA compared with a benign model (lower means smaller degradation).
**PR (Poisoning Rate)** – Fraction of training samples replaced with poisoned ones.

*Supplementary Experiments & Details:* Tables 7 and 8 in Appendix A.2 summarize the attack baselines and defense methods evaluated in Section 5.2–5.3. The appendix further details all experimental settings and provides supplementary analyses, including capability extension, ablation studies, stability, loss parameter sensitivity, and stealthiness, offering a broader perspective on the robustness, stealthiness and overall comprehensiveness of Arcueid.

## 5.2 ATTACK PERFORMANCE

**Effectiveness on Threat Paradigms.** Table 1 summarizes results across three paradigms ($M \mapsto 1$, $M \mapsto N$, and $M \mapsto M$) with both CNN and transformer backbones. Figure 4 visualizes example triggers and poisoned samples produced by Arcueid, illustrating their practical appearance and imperceptibility. Arcueid consistently attains near-perfect ASR (typically $>95\%$) with negligible utility degradation (ΔACC mostly $<5\%$). On CIFAR-10/100, even the all targets settings ($M \mapsto M$) maintain strong attack success (Average ASR $> 90\%$) on CNN models, while transformers show moderate drops (ΔACC $<1\%$) under the most extreme cases. These results confirm that Arcueid scales reliably across mappings, datasets, and architectures.

**Multi-target Supported Attack Comparison.** We compare Arcueid with SOTA backdoor attacks that support multi-target settings. For BadNets, WaNet, and IAD, target labels follow a cyclic rule $y' = (y + 1) \mod 10$, while M2N and Arcueid adopt the same target mapping for fairness. Figure 5 visualizes the classifier logits distribution, where Arcueid produces clear and consistent mappings across all targets, while competing methods collapse into random or uniform patterns under low poisoning budgets. This comparison highlights Arcueid 's distinctive ability to sustain stable all-to-all attacks at extremely low poisoning rates, where prior methods fail to generalize.

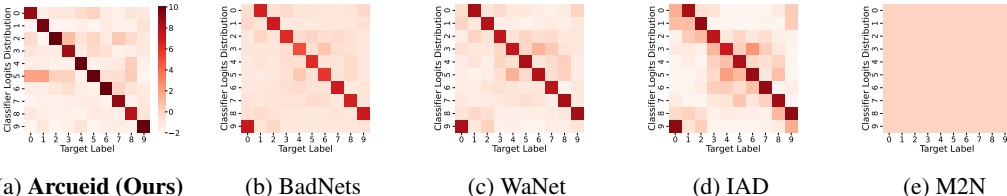

| (a) **Arcueid (Ours)** | (b) BadNets | (c) WaNet | (d) IAD | (e) M2N |
|---|---|---|---|---|

Figure 5: Heatmap comparison among multi-target attack methods (PR=0.1%).

## 5.3 ROBUSTNESS AGAINST DEFENSE MECHANISMS

**Robustness against Pre-training Defense.** Pre-training defenses aim to detect poisoned samples at the input level before they enter the training pipeline. Such approaches are generally regarded as effective only if they can simultaneously achieve a high true-positive rate and a low false-positive rate across diverse attack mappings. As shown in Figure 6, only SCALE-UP shows partial effectiveness in the $M \mapsto 1$ setting, while all methods de-

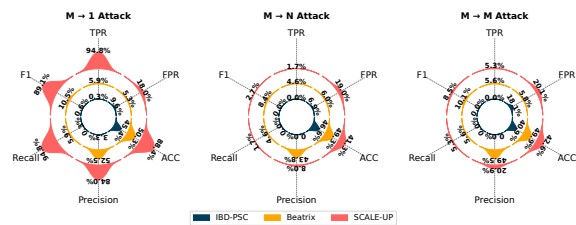

Figure 6: Pre-training defenses against `Arcueid`

grade severely once multiple targets are involved, with recall and F1-scores approaching zero. Overall, these results highlight that `Arcueid`'s multi-trigger, multi-paradigm, and invisible-pattern design significantly enhances its ability to evade input-level filtering mechanisms.

Table 2: **Mid-training defense performance against three attack paradigms of** `Arcueid`. **Performance** denotes the CA and ASR measured on models after applying defenses.

| Attack Paradigm | Defense Type | TPR | FPR | Performance | |
|---|---|---|---|---|---|
| | | | | CA | ASR |
| $M \mapsto 1$ Attack | No Defense | N/A | N/A | 86.5% | 99.1%±0.7% |
| | CT | 0.00% | 61.33% | 53.0% | 90.2%±3.5% |
| | FLARE | 0.00% | 0.00% | 88.6% | 99.7%±0.3% |
| $M \mapsto N$ Attack | No Defense | N/A | N/A | 90.6% | 99.6%±0.3% |
| | CT | 2.00% | 65.39% | 43.4% | 35.1%±24.6% |
| | FLARE | 0.00% | 1.71% | 87.9% | 96.6%±5.3% |
| $M \mapsto M$ Attack | No Defense | N/A | N/A | 90.4% | 99.8%±0.2% |
| | CT | 14.00% | 63.84% | 46.4% | 41.8%±33.1% |
| | FLARE | 2.00% | 13.74% | 88.9% | 98.6%±1.9% |

**Robustness against Mid-training Defense.** Mid-training defenses attempt to continue optimization in the presence of poisoned data by filtering or down-weighting suspicious samples. For such defenses to be considered effective, they must retain CA close to the benign baseline while driving ASR down toward random-guess levels during training time. As shown in Table 2, CT suffers from excessive false positives ($> 60\%$), causing CA to collapse below $50\%$ in multi-target settings. FLARE maintains CA above $87\%$ but leaves ASR largely unaffected ($> 96\%$), nearly identical to undefended models. Taken together, current proactive defenses either cripple utility or fail to suppress `Arcueid`, leaving the backdoor intact.

**Robustness against Post-training Defense.**

Post-training defenses are designed to sanitize a trained model without access to its original training data. To succeed, such techniques should both preserve high CA and suppress ASR close to chance levels. We examine three recent approaches, as shown in Figure 7, FT-SAM reduces ASR in the $M \mapsto 1$ case but is ineffective for multi-target attacks, ABL provides virtually no protection, and NAD achieves partial mitigation with considerable instability. In general, none of these defenses reliably meet the expected standard, underscoring `Arcueid`'s resilience even after model sanitization.

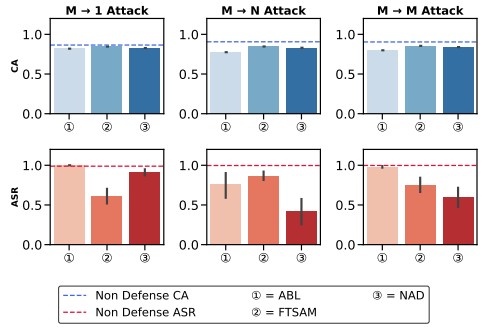

Figure 7: Post-training defenses to `Arcueid`.

## 6 CONCLUSION

This paper introduced `Arcueid`, a unified framework leveraging *Joint Cloud Shaping Multi-trigger Optimization* for effective, stealthy, and robust backdoor attacks across paradigms. Extensive evaluations confirmed high ASR, strong stealthiness, and resilience against SOTA defenses, exposing blind spots in existing countermeasures. Beyond a new benchmark for multi-target backdoors, our results challenge the assumptions that diversity or limited attacker knowledge weakens attacks, showing instead that adaptive multi-trigger designs thrive under realistic constraints. We expect these findings to motivate defenses accounting for multi-trigger interference and inspire exploration of continual or multimodal backdoor settings where such vulnerabilities persist.

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

Bruh

# A APPENDIX

We provide an overview of the appendix contents for easy navigation.

A.1 SUPPLEMENTARY RELATED WORK

The related work discussed in Section 2 primarily focuses on poisoning-based backdoor attacks. Yet the scope of backdoor research extends beyond data poisoning. A substantial body of work has examined *supply-chain backdoor attacks*, in which adversaries, with full control over the training process, implant backdoors into models and redistribute them through public channels. Meanwhile, the escalating threat of backdoor attacks has spurred extensive efforts on *backdoor defenses*, which propose countermeasures at different stages of the learning pipeline. This supplementary section reviews these two complementary directions to provide a more comprehensive view of the backdoor learning landscape.

A.1.1 SUPPLY-CHAIN BACKDOOR ATTACK

**Supply-chain backdoor attacks** describe scenarios in which an adversary independently trains a model and embeds a backdoor during this process, subsequently releasing the compromised model through public channels, often under the guise of an open-source model or a domain-specific utility. Because the adversary possesses full control over both the training process and the model architecture, this threat model typically corresponds to the white-box setting. Early works explored direct weight manipulation. For example, Dumford & Scheirer (2020) perturbed model weights to induce targeted misclassifications without sacrificing accuracy on clean inputs. TBT (Rakin et al., 2020) further demonstrated that Trojans could be injected at the bit level through weight flipping, requiring no access to training data. Along similar lines, Garg et al. (2020) introduced adversarial weight perturbations capable of embedding highly stealthy backdoors. Building on this direction, T-BFA (Rakin et al., 2022) proposed the first targeted bit-flip attack tailored for quantized DNNs, while ProFlip (Chen et al., 2021) progressively identified and flipped a small set of critical parameter bits to implant Trojans into quantized networks without retraining. More recent works have shifted toward data-free settings. DFBA (Cao et al., 2024) and the method proposed in (Lv et al., 2023) embed backdoors by directly modifying neurons or leveraging substitute data, circumventing the need for original training data or labels. Beyond weight-level manipulations, structural modifications have also been introduced. TrojanNet (Tang et al., 2020) appends a model-agnostic module to enable all-label attacks, while SRA (Qi et al., 2022) replaces sub-networks within deployed models to inject physical backdoors. At an even lower abstraction level, DeepPayload (Li et al., 2021d) achieves black-box logic injection through binary-level modifications. Collectively, these supply-chain attacks highlight the feasibility of post-deployment compromise without requiring access to victim data or pipelines. However, they generally depend on strong control over the model or runtime environment and may leave detectable footprints due to the inherent structural or behavioral alterations they introduce.

A.1.2 BACKDOOR DEFENSE

To counteract backdoor threats, a wide range of defense strategies have been proposed, which can be broadly classified into three categories via its applied period: **pre-training defense**, **mid-training defense** and **post-training defense**.

**Pre-training defense** aims to identify adversarial samples before training time by analysing various properties of incoming data. SCALE-UP (Guo et al., 2023) leverages the prediction consistency of scaled input images to detect backdoors in a black-box setting, supporting both patch-based and advanced trigger types. MSPC (Pal et al., 2024) introduces a mask-aware scaled prediction consistency framework and a bi-level optimization process to detect poisoned samples without requiring clean data or manual thresholds, outperforming prior methods under realistic constraints. Beatrix (Ma et al., 2023) proposes a Gram matrix-based method to model high-order feature correlations, effectively detecting both universal and sample-specific backdoors. More recently, IBD-PSC (Hou et al., 2024a) enhances robustness and generalization by amplifying batch normalization parameters and evaluating confidence consistency, thereby overcoming several limitations of earlier input-based defenses (Chou et al., 2018; Gao et al., 2022; Liu et al., 2023).

**Mid-training defense** focuses on detecting and suppressing poisoned samples during the training process, thereby mitigating backdoor contamination while allowing models to continue effective learning. DBD (Huang et al., 2022) alleviates poisoning threats by decoupling the end-to-end optimization into three stages, effectively weakening the influence of triggers. ASD (Gao et al., 2023)

provides a unified framework that adaptively partitions data into clean and polluted pools for targeted training-time defense. Honeypot-based defenses (Tang et al., 2023) attach auxiliary modules to lower layers to absorb and neutralize backdoor features during fine-tuning. CT (Qi et al., 2023b) proactively detects poisoned samples by injecting mislabeled clean data, decoupling benign correlations from malicious ones to expose triggers. MeCa (Pu et al., 2024) enables training clean models directly on poisoned datasets without auxiliary clean supervision by leveraging robustness discrepancies of poisoned samples under adversarial perturbations. More recently, FLARE (Hou et al., 2025) introduces a universal dataset purification framework that aggregates abnormal activations across layers and employs adaptive subspace clustering to distinguish poisoned from benign data.

**Post-training defense** aims to repair compromised models or mitigate backdoor behaviors after training. Early reactive approaches, such as Neural Cleanse (Wang et al., 2019), reverse-engineer potential triggers through anomaly detection, followed by input filtering, neuron pruning, or retraining. STRIP (Gao et al., 2019; 2022) provides a lightweight post-hoc detection mechanism by measuring prediction entropy under perturbed conditions, enabling efficient black-box identification of trojaned inputs without prior trigger knowledge. More recent methods improve efficiency and generalization: NAD (Li et al., 2021b) applies attention distillation between a fine-tuned teacher and the backdoored student model with only a small clean dataset; I-BAU (Zeng et al., 2022) frames backdoor removal as a minimax adversarial unlearning problem solvable via implicit hypergradient methods; and FT-SAM (Zhu et al., 2023) integrates sharpness-aware minimization (Foret et al., 2021) with fine-tuning to perturb backdoor-sensitive neurons, achieving strong mitigation even with limited data. In parallel, proactive defenses such as ABL (Li et al., 2021a) exploit the faster convergence and class-dependency patterns of poisoned samples via a dual-stage gradient ascent strategy to isolate and suppress them, enabling robust training even on corrupted datasets.

## A.2 SUPPLEMENTARY EVALUATION

This section provides additional experimental results and details that complement the main text. We include extended analyses, supplementary figures, and tables that could not be accommodated in the main pages due to space constraints. These results further support our findings and offer deeper insights into the robustness and effectiveness of Arcueid.

Table 3: **Attack performance ($\triangle$ACC/ASR$\pm$Std) on various models under all targets attack.**

| Dataset | $M \mapsto N$ | PR | ResNet-18 | | ResNet-34 | | VGG13-BN | | ViT | | SimpleViT | |
|---------|---------------|-----|-----------|-----|-----------|-----|----------|-----|-----|-----|-----------|-----|
| | | | $\triangle$ACC | ASR | $\triangle$ACC | ASR | $\triangle$ACC | ASR | $\triangle$ACC | ASR | $\triangle$ACC | ASR |
| CIFAR-10 | 3→3 | 0.03% | 1.7% | 99.7%±0.2% | 3.8% | 99.6%±0.5% | 1.8% | 95.8%±5.1% | -0.2% | 88.3%±8.5% | -0.2% | 89.5%±3.9% |
| | 5→5 | 0.05% | 4.2% | 94.3%±6.6% | 4.1% | 99.2%±0.6% | 1.6% | 97.0%±4.1% | 0.8% | 80.1%±14.0% | 0.2% | 95.5%±3.7% |
| | 8→8 | 0.08% | 3.5% | 99.5%±0.7% | 3.4% | 99.7%±0.3% | 1.5% | 99.1%±1.1% | 0.6% | 88.3%±3.3% | -0.2% | 89.7%±5.8% |
| | 10→10 | 0.10% | 5.9% | 91.9%±4.8% | 4.7% | 98.8%±1.0% | 2.0% | 98.8%±0.9% | 0.3% | 81.5%±9.3% | 0.2% | 92.0%±3.4% |
| CIFAR-100 | 25→25 | 0.25% | 2.5% | 97.2%±1.7% | 5.1% | 95.1%±2.9% | 5.0% | 90.9%±4.9% | -0.1% | 86.9%±8.7% | -0.9% | 88.7%±5.2% |
| | 50→50 | 0.50% | 3.0% | 98.0%±1.4% | 3.2% | 98.7%±0.9% | 6.0% | 85.3%±7.3% | -0.5% | 82.9%±8.9% | -0.3% | 82.1%±7.7% |
| | 75→75 | 0.75% | 4.3% | 97.2%±2.4% | 7.5% | 92.0%±4.1% | 6.1% | 86.5%±9.1% | -0.1% | 82.8%±10.1% | 1.0% | 83.5%±6.7% |
| | 100→100 | 1.00% | 3.7% | 98.1%±1.4% | 7.7% | 88.2%±6.7% | 3.2% | 95.3%±2.3% | -0.5% | 80.4%±10.2% | 0.6% | 84.2%±9.1% |
| TinyImageNet | 50→50 | 0.50% | 7.3% | 99.8%±0.3% | 6.2% | 99.9%±0.1% | 4.4% | 99.8%±0.2% | 0.5% | 92.7%±3.5% | 0.2% | 91.7%±4.5% |
| | 100→100 | 1.00% | 6.1% | 99.9%±0.1% | 9.3% | 99.9%±0.1% | 3.6% | 99.9%±0.1% | 1.3% | 91.9%±4.1% | 0.7% | 90.2%±4.3% |
| | 150→150 | 1.50% | 7.0% | 99.9%±0.2% | 7.9% | 99.9%±0.1% | 6.7% | 98.9%±1.3% | 0.7% | 92.7%±3.8% | 1.4% | 87.9%±5.3% |
| | 200→200 | 2.00% | 6.2% | 99.9%±0.1% | 7.2% | 99.9%±0.1% | 8.2% | 98.7%±1.3% | 1.0% | 92.7%±3.6% | 0.9% | 89.4%±5.0% |

### A.2.1 GLOBAL EXPERIMENTAL SETTING

Unless otherwise specified, ResNet-18 on CIFAR-10 is adopted as the default target model and dataset, with the overall PR fixed at $0.1\%$ (corresponding to $0.01\%$ per trigger). To ensure no unfair advantage, we strictly separate the surrogate and target environments, where the surrogate model and dataset are always different from those of the victim. Additional hyperparameter and implementation details can be found in Appendix A.7. The set of backdoor attacks compared throughout the paper is summarized in Table 7, while the defense baselines considered are listed in Table 8.

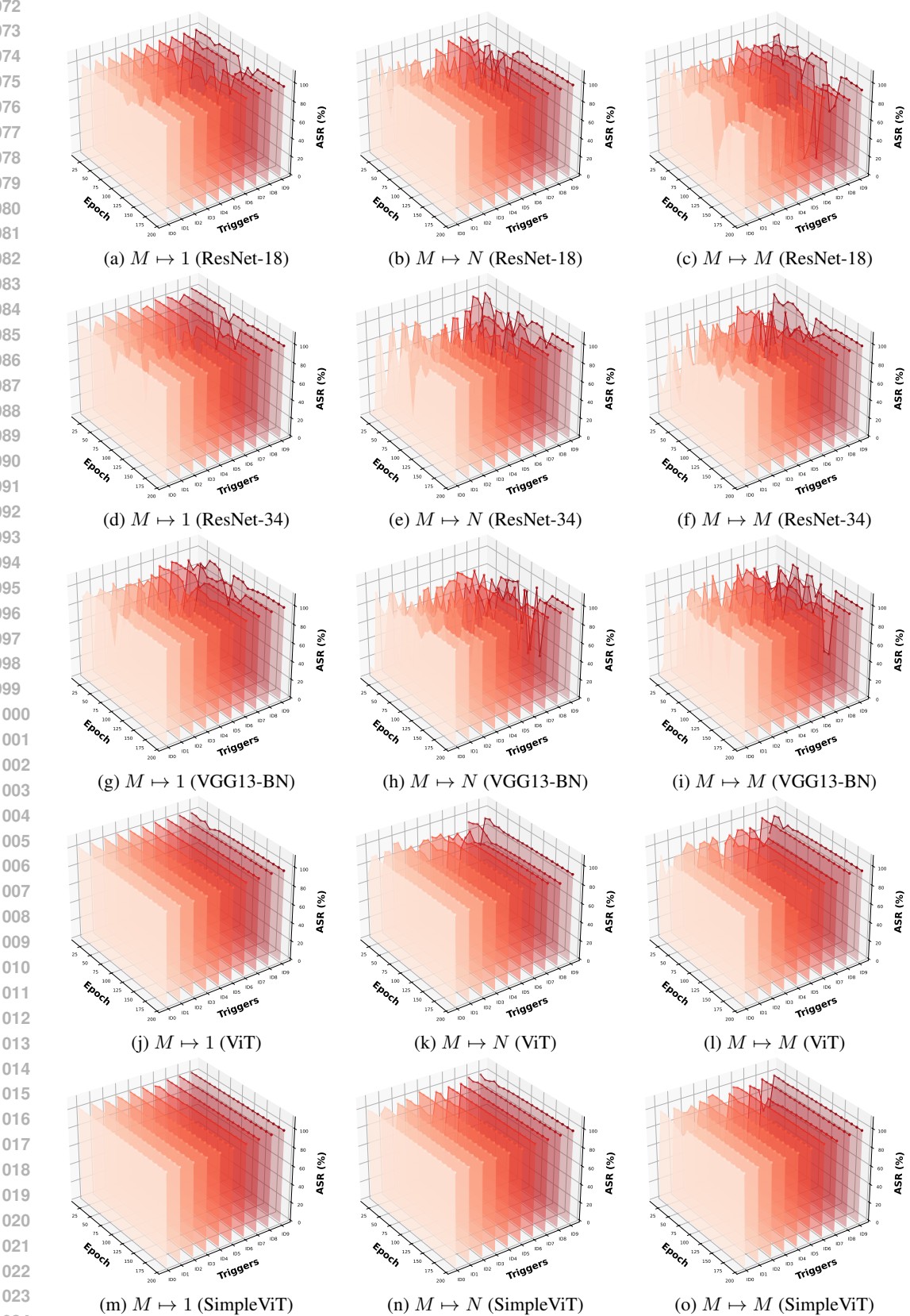

(a) $M \mapsto 1$ (ResNet-18)  (b) $M \mapsto N$ (ResNet-18)  (c) $M \mapsto M$ (ResNet-18)

(d) $M \mapsto 1$ (ResNet-34)  (e) $M \mapsto N$ (ResNet-34)  (f) $M \mapsto M$ (ResNet-34)

(g) $M \mapsto 1$ (VGG13-BN)  (h) $M \mapsto N$ (VGG13-BN)  (i) $M \mapsto M$ (VGG13-BN)

(j) $M \mapsto 1$ (ViT)  (k) $M \mapsto N$ (ViT)  (l) $M \mapsto M$ (ViT)

(m) $M \mapsto 1$ (SimpleViT)  (n) $M \mapsto N$ (SimpleViT)  (o) $M \mapsto M$ (SimpleViT)

Figure 8: Stability study across paradigms on diverse backbones.

Table 4: **Attack performance ($\Delta$ACC/ASR$\pm$Std) on various models under multiple paradigms in clean-label attack setting.**

| Dataset | $M \mapsto N$ | ResNet-18 | | ResNet-34 | | VGG13-BN | | ViT | | SimpleViT | |
|---|---|---|---|---|---|---|---|---|---|---|---|
| | | $\Delta$ACC | ASR | $\Delta$ACC | ASR | $\Delta$ACC | ASR | $\Delta$ACC | ASR | $\Delta$ACC | ASR |
| CIFAR-10 (PR=0.1%) | $10 \to 1$ | 3.9% | 91.2%±8.8% | 3.4% | 99.9%±0.1% | 1.9% | 99.5%±0.5% | 0.4% | 95.8%±3.8% | 0.3% | 99.9%±0.1% |
| | $10 \to 2$ | 1.9% | 98.8%±1.0% | 3.5% | 98.7%±1.4% | 2.2% | 98.1%±1.3% | 0.7% | 81.3%±9.1% | 0.3% | 99.1%±1.0% |
| | $10 \to 5$ | 2.2% | 99.3%±0.8% | 4.2% | 97.7%±1.5% | 2.8% | 92.9%±5.4% | 0.3% | 88.9%±5.4% | -0.2% | 99.3%±0.4% |
| | $10 \to 10$ | 1.3% | 98.0%±1.7% | 2.8% | 98.8%±1.1% | 2.6% | 95.1%±3.7% | 0.2% | 86.2%±7.5% | 0.4% | 99.3%±0.6% |
| CIFAR-100 (PR=1%) | $100 \to 1$ | 2.5% | 100.0%±0.0% | 1.9% | 100.0%±0.0% | 6.4% | 99.7%±0.4% | -0.6% | 99.9%±0.1% | 0.4% | 100.0%±0.0% |
| | $100 \to 5$ | 2.9% | 80.1%±17.7% | 3.1% | 80.1%±17.7% | 5.6% | 84.8%±18.3% | -0.9% | 85.9%±10.1% | 0.0% | 77.2%±15.9% |
| | $100 \to 10$ | 2.9% | 80.1%±17.7% | 3.9% | 77.0%±19.5% | 5.2% | 84.9%±15.9% | -0.9% | 82.9%±16.1% | 0.1% | 83.3%±11.1% |
| | $100 \to 100$ | 3.6% | 82.3%±10.1% | 5.2% | 82.3%±10.1% | 3.3% | 82.8%±16.3% | -0.9% | 78.6%±18.3% | 1.2% | 80.6%±11.7% |
| TinyImageNet (PR=2%) | $200 \to 1$ | 6.8% | 100.0%±0.0% | 3.6% | 100.0%±0.0% | 7.9% | 100.0%±0.0% | 1.1% | 99.1%±1.7% | 0.4% | 99.3%±1.1% |
| | $200 \to 2$ | 6.9% | 86.5%±10.3% | 4.0% | 77.3%±16.2% | 7.6% | 92.2%±9.6% | 1.1% | 82.6%±10.3% | 0.9% | 83.3%±10.9% |
| | $200 \to 4$ | 7.7% | 94.8%±4.4% | 4.6% | 98.9%±1.4% | 6.1% | 80.1%±14.8% | 1.1% | 85.9%±9.9% | 0.8% | 76.1%±15.9% |
| | $200 \to 200$ | 8.8% | 79.8%±18.3% | 5.5% | 84.6%±13.7% | 4.9% | 98.8%±1.3% | 1.3% | 81.1%±10.5% | 0.6% | 81.3%±10.5% |

Table 5: **Attack performance ($\Delta$ACC/ASR) in all-to-one attack paradigm under dirty-label and clean-label settings.**

| Dataset | Label Mode | ResNet-18 | | ResNet-34 | | VGG13-BN | | ViT | | SimpleViT | |
|---|---|---|---|---|---|---|---|---|---|---|---|
| | | $\Delta$ACC | ASR | $\Delta$ACC | ASR | $\Delta$ACC | ASR | $\Delta$ACC | ASR | $\Delta$ACC | ASR |
| CIFAR-10 (PR=0.01%) | Dirty-label | -2.7% | 100.0% | -4.8% | 99.5% | -1.9% | 99.6% | -0.5% | 98.5% | 1.0% | 99.9% |
| | Clean-label | -4.1% | 96.0% | -4.6% | 97.4% | -1.4% | 99.6% | -0.2% | 95.3% | 0.4% | 100.0% |
| CIFAR-100 (PR=0.01%) | Dirty-label | -3.0% | 90.0% | -4.0% | 98.2% | -5.9% | 85.3% | 0.8% | 99.9% | -0.4% | 100.0% |
| | Clean-label | -2.6% | 85.5% | -4.2% | 85.2% | -6.0% | 82.4% | 1.0% | 99.4% | -0.3% | 100.0% |
| TinyImageNet (PR=0.01%) | Dirty-label | -6.8% | 100.0% | -8.3% | 100.0% | -8.6% | 99.6% | -0.5% | 99.9% | -0.4% | 100.0% |
| | Clean-label | -6.6% | 100.0% | -9.8% | 100.0% | -3.8% | 100.0% | -0.9% | 99.9% | -0.5% | 99.9% |

### A.2.2 EXTENDED CAPABILITY ANALYSIS

To assess the breadth and adaptability of Arcueid, we conduct extended analyses on three dimensions: its effectiveness under clean-label constraints, its scalability across different target scopes, and its competitiveness in the conventional all-to-one paradigm.

**Clean-label Analysis.** We further evaluate Arcueid under the more restrictive clean-label setting (first defined by Turner et al. (2019)), where poisoned samples must retain their original ground-truth labels. Table 4 summarizes results across CIFAR-10, CIFAR-100, and TinyImageNet. Despite the absence of label manipulation, Arcueid still delivers strong attack performance: on CIFAR-10, ASR exceeds 95% in most cases with $\Delta$ACC under 4%, and even the challenging $M \mapsto M$ setting ($10 \to 10$) sustains over 90% ASR. On CIFAR-100 and TinyImageNet, ASR remains high in $M \mapsto 1$ and $M \mapsto N$ configurations, while broader mappings show moderate degradation, yet still outperforming existing clean-label baselines reported in prior work. These results confirm that Arcueid is not limited to dirty-label attacks but also retains effectiveness under clean-label constraints, significantly broadening its potential threat scope.

**Target Scope Analysis.** We analyze the number of triggers $K$ (mentioned in Section 4.1) under the most challenging $M \mapsto M$ paradigm. Table 3 shows how attack performance changes as we increase the number of triggers (PR is adjusted accordingly so that the per-trigger PR remains constant). Arcueid scales gracefully: tiny budgets suffice for small-to-medium mappings (e.g., $3 \mapsto 3$ yields over 95% ASR on ResNet-18), and modest increases in PR sustain high ASR as the target set grows. Larger target scopes require higher absolute PR but remain practical, CIFAR-100 reaches near-perfect ASR for many intermediate scopes with PR in the 0.25–1.0% range, and TinyImageNet attains 99% ASR for large-scale mappings when PR is increased to 0.5–2.0%. Across architectures, CNN backbones are most susceptible, showing very high ASR with only small clean-accuracy drops. Transformer models exhibit greater variance and larger declines in some extreme broad-target settings, but remain attackable for most practical scopes. In short, expanding the target set does not collapse attack effectiveness; instead, Arcueid presents a smooth, predictable trade-off between trigger count and required poisoning budget, demonstrating practical scalability.

Table 6: **Comparison of backdoor attack performance (ΔACC/ASR) in all-to-one attack paradigm across datasets.** All results are reported on CIFAR-10, CIFAR-100, and TinyImageNet. ΔACC denotes accuracy drop on clean samples, and ASR indicates the attack success rate on poisoned samples.

| Attack Method | CIFAR-10 (PR=0.01%) | | CIFAR-100 (PR=0.01%) | | TinyImageNet (PR=0.01%) | |
|---|---|---|---|---|---|---|
| | ΔACC | ASR | ΔACC | ASR | ΔACC | ASR |
| BadNets | -1.2% | 10.4% | -1.8% | 1.1% | -4.1% | 0.6% |
| Blended | -1.6% | 10.1% | -2.1% | 1.1% | -5.1% | 0.6% |
| Refool | -1.3% | 10.1% | -2.0% | 1.2% | -4.9% | 1.7% |
| LC | -17.7% | 12.7% | -2.7% | 1.1% | -49.5% | 0.1% |
| TUAP | -1.0% | 8.3% | -2.5% | 0.7% | -5.2% | 0.1% |
| PhysicalBA | +2.4% | 10.0% | +3.5% | 1.1% | +1.4% | 0.6% |
| WaNet | -1.7% | 10.2% | -1.5% | 1.1% | -4.9% | 12.1% |
| AdaptivePatch | -7.0% | 10.6% | -1.3% | 1.5% | -7.0% | 1.4% |
| Narcissus | -4.1% | 39.1% | -3.5% | 54.5% | -6.1% | 99.3% |
| **Arcueid (Dirty-Label)** | -2.7% | 100.0% | -3.0% | 90.0% | -6.8% | 100.0% |
| **Arcueid (Clean-Label)** | -4.1% | 96.0% | -2.6% | 85.5% | -6.6% | 100.0% |

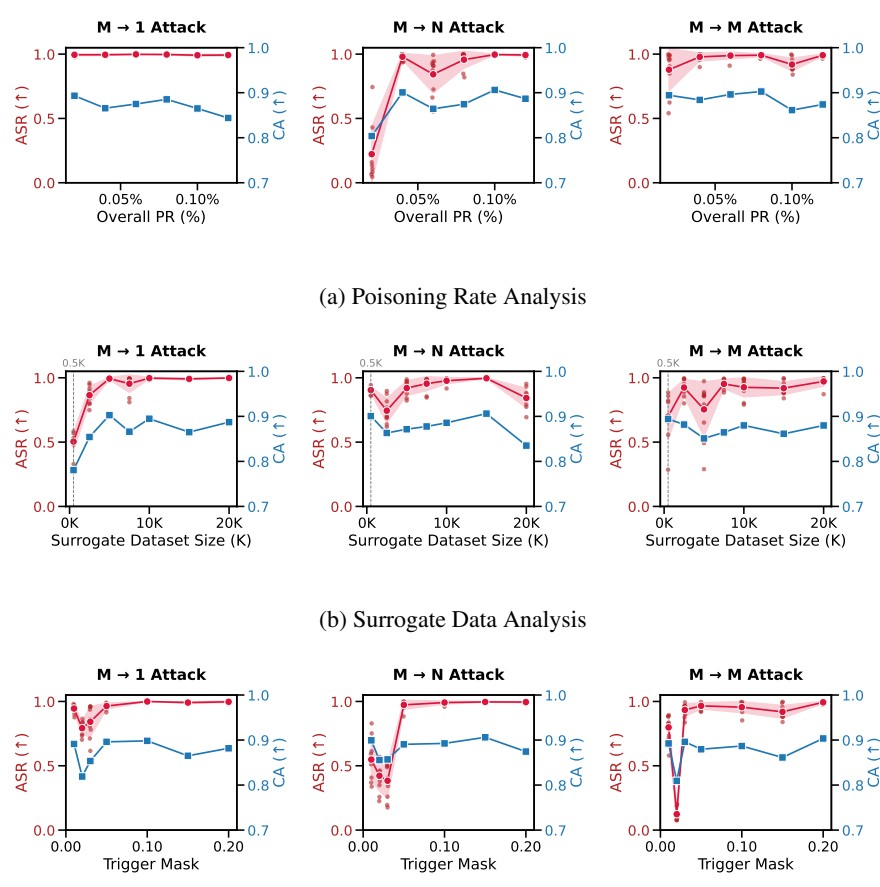

(a) Poisoning Rate Analysis

(b) Surrogate Data Analysis

(c) Trigger Mask Analysis

Figure 9: Ablation study on key factors influencing Arcueid 's effectiveness.

**All-to-one Analysis.** Finally, we examine the classical all-to-one paradigm, which corresponds to setting $K$=1 in Arcueid. All experiments in this part are conducted under an extremely low poisoning rate of $0.01\%$. Table 6 compares Arcueid against a wide range of existing all-to-one attacks introduced in Table 7. Even under this restrictive budget, Arcueid substantially outperforms prior methods: on CIFAR-10, CIFAR-100, and TinyImageNet, it consistently achieves near-perfect ASR (often ≥99%) with limited clean-accuracy degradation, while standard baselines such as BadNets,

Table 7: **Summary of backdoor attacks evaluated in this paper**. *Attack Property* indicates attacker assumptions, including whether the attack is clean-label, requires no access to training data, or is model- and training-agnostic. *Attack Target Scope* shows whether the attack supports single-target, multiple-target, or broad-class settings. *Robustness* evaluates resistance against input-based detection, training-stage defenses, and model-based mitigation. *Stealthiness* reports whether the trigger is invisible and the minimum poison rate per target required to achieve a high attack success rate ($> 80\%$). ○ The item is not supported by the attack; ● The item is supported by the attack.

| Attack | Attack Property | | | Attack Target Scope | | | Robustness | | | Stealthiness | |
|---|---|---|---|---|---|---|---|---|---|---|---|
| | Clean-label | Data-free | Model-agnostic | Single | Multiple | Broad | Detection | Training | Mitigation | Invisible | PR/Target |
| Blended (Chen et al., 2017) | ○ | ● | ● | ● | ○ | ○ | ○ | ○ | ○ | ● | 10% |
| Refool (Liu et al., 2020) | ● | ● | ● | ● | ○ | ○ | ○ | ○ | ○ | ○ | 0.57% |
| LC (Turner et al., 2019) | ● | ○ | ● | ● | ○ | ○ | ● | ○ | ○ | ● | 0.40% |
| TUAP (Zhao et al., 2020) | ● | ○ | ○ | ● | ○ | ○ | ● | ○ | ○ | ● | 0.30% |
| PhysicalBA (Li et al., 2021c) | ○ | ● | ● | ● | ○ | ○ | ○ | ○ | ○ | ○ | 0.50% |
| AdaptivePatch (Qi et al., 2023a) | ○ | ● | ● | ● | ○ | ○ | ● | ○ | ● | ○ | 0.30% |
| Narcissus (Zeng et al., 2023) | ● | ● | ○ | ● | ○ | ○ | ● | ○ | ● | ● | 0.05% |
| BadNets (Gu et al., 2019) | ○ | ● | ● | ● | ● | ○ | ○ | ○ | ○ | ○ | 1% |
| WaNet (Nguyen & Tran, 2021) | ○ | ● | ● | ● | ● | ○ | ● | ○ | ● | ● | 1% |
| IAD (Nguyen & Tran, 2020) | ○ | ○ | ○ | ● | ● | ○ | ● | ○ | ● | ○ | 1% |
| M2N (Hou et al., 2024b) | ○ | ○ | ● | ● | ● | ○ | ○ | ○ | ● | ● | 0.40% |
| **Arcueid (Ours)** | ● | ● | ● | ● | ● | ● | ● | ● | ● | ● | ≤0.01% |

Table 8: **Summary of the existing backdoor defenses evaluated in this paper**. *Proactive Training* denotes methods that prevent backdoor injection during training. ○ The item is not supported by the defense; ● The item is supported by the defense.

| Defense | Defense Stage | Defense Task | | | Threat Model | |
|---|---|---|---|---|---|---|
| | | Input Detection | Proactive Training | Model Mitigation | Black-box | Needs Clean Data |
| SCALE-UP (Guo et al., 2023) | Pre-training | ● | ○ | ○ | ● | ● |
| Beatrix (Ma et al., 2023) | | ● | ○ | ○ | ○ | ● |
| IBD-PSC (Hou et al., 2024a) | | ● | ○ | ○ | ○ | ● |
| CT (Qi et al., 2023b) | Mid-training | ● | ● | ○ | ● | ○ |
| FLARE (Hou et al., 2025) | | ● | ● | ○ | ● | ○ |
| NAD (Li et al., 2021b) | Post-training | ○ | ○ | ● | ● | ● |
| ABL (Li et al., 2021a) | | ○ | ● | ● | ○ | ○ |
| FT-SAM (Zhu et al., 2023) | | ○ | ○ | ● | ○ | ● |

WaNet, and Blended collapse to nearly random ASR. Methods designed for stealthiness, such as LC or Narcissus, achieve partial success but either incur large clean-accuracy drops or fail to generalize across datasets. Table 5 further breaks down Arcueid 's all-to-one performance under dirty-label and clean-label modes across five architectures. In both settings, Arcueid sustains high ASR with only minor accuracy loss, reaching 100% ASR on TinyImageNet even without label manipulation. These results show that Arcueid is not only effective in multi-target paradigms, but also strictly surpasses SOTA baselines in the conventional all-to-one paradigm, highlighting its role as a unified framework for both traditional and advanced backdoor attacks.

### A.2.3 ABLATION STUDY

To better understand the robustness and design properties of Arcueid, we perform ablation studies on three critical factors: PR (Poisoning Rate), surrogate data scale, and trigger mask.

**Poisoning Rate Analysis.** We vary the overall PR from $0.02\%$ to $0.12\%$ (per-trigger rate from $0.002\%$ to $0.012\%$). As shown in Figure 9a, Arcueid remains highly effective even at extremely low poisoning budgets: at only $0.04\%$, ASR already exceeds 97% in both $M \to N$ and $M \to M$ settings with negligible accuracy drop. Performance stabilizes around $0.08\%$–$0.10\%$, confirming that the attack requires only a little data injection to achieve strong persistence.

**Surrogate Data Analysis.** We investigate the impact of surrogate data scale, ranging from 500 to 20,000 samples drawn under a *non-IID* distribution. Results in Figure 9b show that attack performance improves rapidly with more surrogate data, surpassing 95% ASR once 7,500 samples are

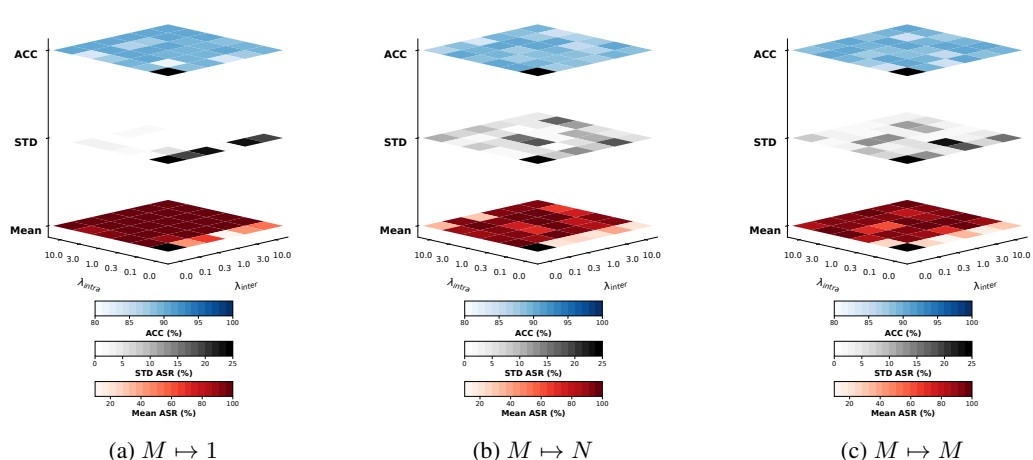

(a) $M \mapsto 1$        (b) $M \mapsto N$        (c) $M \mapsto M$

Figure 10: Sensitivity of ASR/Std/CA to $(\mathcal{L}_{\text{intra}}, \mathcal{L}_{\text{inter}})$ across multiple paradigms. The case $(\mathcal{L}_{\text{intra}}, \mathcal{L}_{\text{inter}}) = (0, 0)$ is marked as N/A, since it corresponds to no effective optimization.

used. Importantly, further scaling to 10,000–20,000 samples yields only marginal gains, indicating that `Arcueid` does not rely on large-scale auxiliary datasets to optimize triggers effectively.

**Trigger Mask Analysis.** Finally, we analyze the role of the blending mask $\alpha_k \in [0, 1]^{C \times H \times W}$ (introduced in Section 4.2) that controls trigger visibility. As shown in Figure 9c, overly small masks weaken the attack, reducing ASR below $80\%$ in complex mappings. Larger masks improve stability, with consistently high ASR once the mask exceeds 0.05. Notably, performance remains strong up to 0.20, indicating that `Arcueid` tolerates a wide range of trigger strengths without compromising stealth.

### A.2.4 LOSS WEIGHTS SENSITIVITY ANALYSIS

Recall that `Arcueid` optimizes the following auxiliary objective at the feature level (in Section 4.2):

$$\mathcal{L}_{\text{total}} = \lambda_{\text{intra}} \mathcal{L}_{\text{intra}} + \lambda_{\text{inter}} \mathcal{L}_{\text{inter}},$$

where $\mathcal{L}_{\text{intra}}$ penalizes the variance of triggered features within each pattern cluster, and $\mathcal{L}_{\text{inter}}$ enforces dispersion between cluster centroids via a margin constraint. The two terms play complementary roles: $\mathcal{L}_{\text{intra}}$ ensures that triggered samples converge to a coherent and predictable cloud, which is critical for transferring consistent decision boundaries to victim training. $\mathcal{L}_{\text{inter}}$ prevents collapse among multiple triggers by enlarging centroid gaps, thereby reducing cross-trigger interference and stabilizing success across targets. Removing $\mathcal{L}_{\text{intra}}$ yields unconstrained, scattered feature clouds that fail to anchor to the target class, while removing $\mathcal{L}_{\text{inter}}$ risks centroid overlap that causes unfair allocation of decision regions or severe variance across targets.

We systematically vary $\lambda_{\text{intra}}$ and $\lambda_{\text{inter}}$ on logarithmic scales $\{0, 0.1, 0.3, 1, 3, 10\}$ and evaluate them under three representative paradigms: $M \mapsto 1$, $M \mapsto N$, and $M \mapsto M$. Figures 10a–10c report the mean ASR, its standard deviation, and CA.

Our observations are as follows:

- **Inter-only is insufficient.** When $\lambda_{\text{intra}}{=}0$, ASR remains low in multi-target regimes, indicating that repulsion without compactness fails to anchor decisions.

- **Intra-only is already strong, and modest $\lambda_{\text{inter}}$ further enhances fairness and stability.** With $\lambda_{\text{inter}}{=}0$, ASR is already high, showing that cluster cohesion alone suffices. Introducing a small $\beta$ further reduces variance and improves worst-case success across targets.

- **Overweighting $\lambda_{\text{inter}}$ is harmful.** Excessive $\lambda_{\text{inter}}$ activates the hinge almost everywhere, injecting noisy repulsion and degrading overall performance.

- **Single-target scenarios** ($M \mapsto 1$) **are less sensitive.** Once $\lambda_{\text{intra}} > 0$, ASR quickly saturates across a wide range, while $\lambda_{\text{inter}}$ primarily reduces variance without significantly affecting the mean.

In summary, both terms are necessary in principle: $\mathcal{L}_{\text{intra}}$ ensures success, while $\mathcal{L}_{\text{inter}}$ promotes collision avoidance and evenness. Yet, *tuning is straightforward*: balanced or $\alpha$-leaning weights (e.g., $\alpha \in [0.3, 3]$, $\beta \in [0.1, 1]$) consistently achieve $> 95\%$ ASR with low variance across paradigms while maintaining CA. Therefore, we adopt $(\alpha, \beta) = (1, 1)$ as the default configuration.

### A.2.5 Stability Analysis

We further investigate the *stability* of `Arcueid` across paradigms ($M \mapsto 1$, $M \mapsto N$, and $M \mapsto M$). Figures 8 show waterfall plots of ASR trajectories over training epochs under the five representative architectures introduced in Section 5.1. The results reveal that `Arcueid` maintains consistently high and steady ASR throughout training without collapse or oscillation, demonstrating that our trigger–target associations remain intact even under heterogeneous model inductive biases. Importantly, convergence behaviors remain smooth across all paradigms, confirming that our method not only ensures high attack effectiveness but also stabilizes the poisoned training dynamics against gradient noise and architectural variations.

### A.2.6 Stealthiness Evaluation

*Metrics.* We assess stealthiness using complementary pixel-, signal-, perceptual- and representation-level measures:

- $\ell_\infty$**-norm** — Measures the worst-case per-pixel perturbation magnitude, where lower values indicate reduced visibility of the trigger.

- **MSE / PSNR** — Capture signal-domain distortion, where lower MSE and higher PSNR values correspond to smaller overall perturbations.

- **LPIPS** (Zhang et al., 2018) — A learned perceptual similarity metric correlated with human judgment, where lower values indicate higher perceptual similarity to benign inputs.

- **Residual statistics / sparsity** — Characterize the spatial footprint and sparsity of the perturbation, for example by reporting the proportion of pixels exceeding a threshold $|\Delta| > \tau$.

- **Grad-CAM similarity** (Selvaraju et al., 2017) — Quantifies the alignment of attention maps between original and poisoned inputs using cosine or Pearson similarity, thereby indicating whether model focus is preserved.

- **Feature-space cluster metrics** — Evaluate the embedding distribution of poisoned samples through methods such as t-SNE visualization, highlighting how they are organized under benign and backdoored models.

Together these metrics provide a comprehensive picture of both low-level visibility and high-level semantic or representation impact, which we then use to evaluate the imperceptibility of `Arcueid` through both quantitative metrics and qualitative visualization. Table 9 compares $\ell_\infty$-norm and LPIPS against representative stealthy backdoor attack baselines. `Arcueid` achieves a favorable balance with $\ell_\infty = 0.2121$ and LPIPS $= 0.0301$, significantly outperforming TUAP, AdaptivePatch, and Narcissus, while approaching the imperceptibility of WaNet and LC. Complementary signal-domain metrics in Figure 12 show that triggers introduce an average MSE of $0.0015$ and PSNR of $28.19$ dB, indicating distortion well below human-detectable thresholds. Together these results confirm that `Arcueid` produces visually stealthy perturbations without sacrificing attack effectiveness.

**Residual Analysis.** Figure 12 visualizes ten optimized triggers (a)–(j) via *Joint Cloud Shaping Multi-trigger Optimization* mechanism. For each case, the first row shows clean images, the second row residuals, and the third row poisoned images. Residual maps demonstrate that perturbations are spatially localized and of small magnitude, with most pixel changes imperceptible by eye. This confirms that `Arcueid` does not rely on conspicuous texture overlays or large-scale pixel modifications.

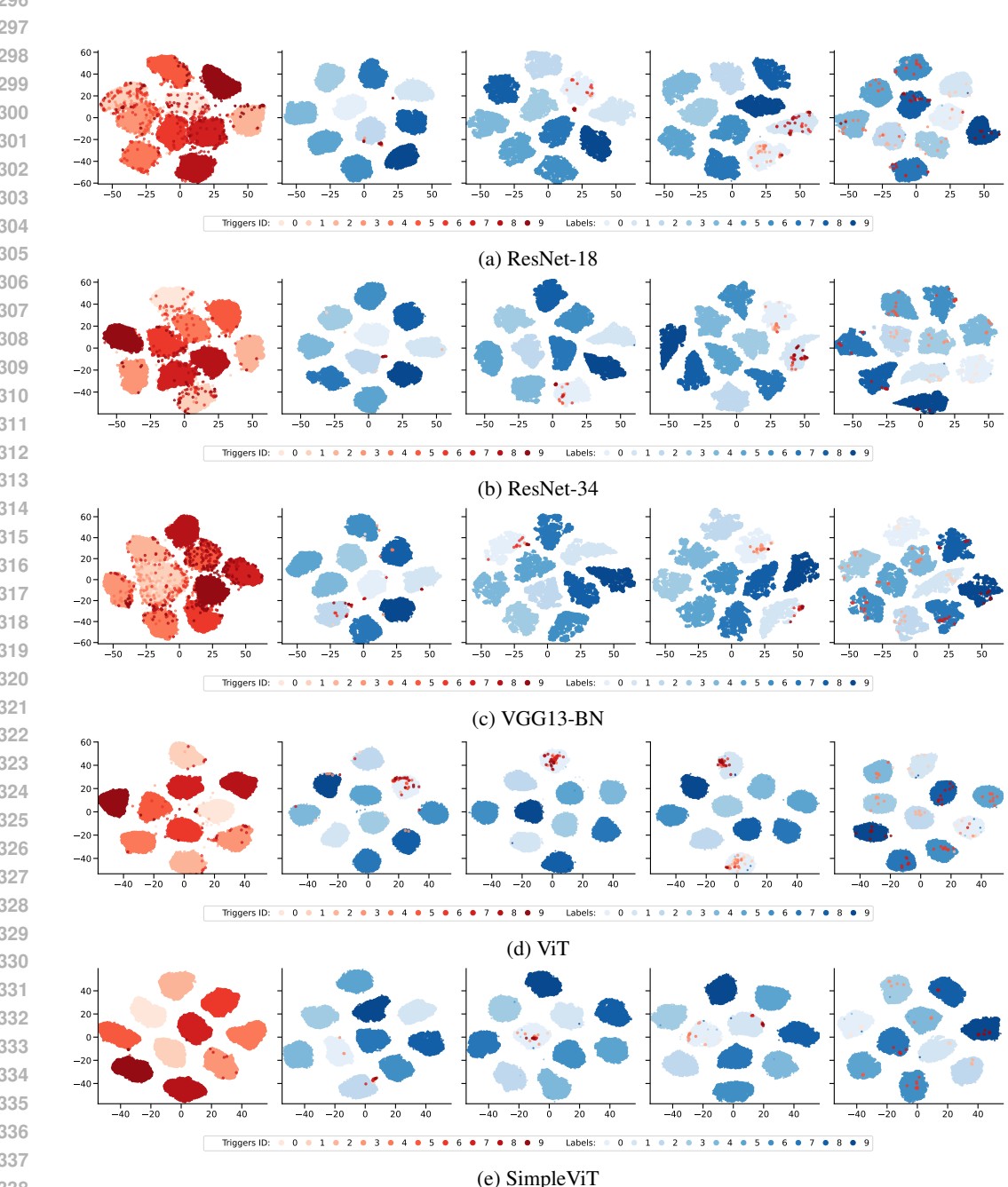

Figure 11: Visualization of trigger-induced feature representations. For each backbone, we first optimize $K=10$ triggers and then apply them under different attack paradigms ($M \mapsto 1$, $M \mapsto N$ and $M \mapsto M$). Five panels *(left to right)* show: (1) **All Poisoned Features (Benign Model)**: the full poisoned training set (50,000 samples) embedded under a benign model, (2) **Poisoned Set (Benign Model)**: a random subset of 100 poisoned samples embedded under a benign model, (3) **Poisoned Set (Model $M \mapsto 1$ Backdoored)**: the same poisoned set forwarded through a model trained with all 10 triggers mapped to a single target, (4) **Poisoned Set (Model $M \mapsto N$ Backdoored)**: the poisoned set embedded by a model trained with 10 triggers mapped to two targets, and (5) **Poisoned Set (Model $M \mapsto M$ Backdoored)**: the poisoned set projected from a model trained with one-to-one mappings between the 10 triggers and 10 targets.

Table 9: Visual quality comparison across attack methods.

| | TUAP | WaNet | AdaptivePatch | LC | Narcissus | Arcueid(Ours) |
|---|---|---|---|---|---|---|
| $\ell_\infty$-norm | 0.7021 | 0.1229 | 0.8992 | 0.9400 | 0.1255 | **0.2121** |
| **LPIPS** | 0.0480 | 0.0047 | 0.1295 | 0.0048 | 0.1047 | **0.0301** |

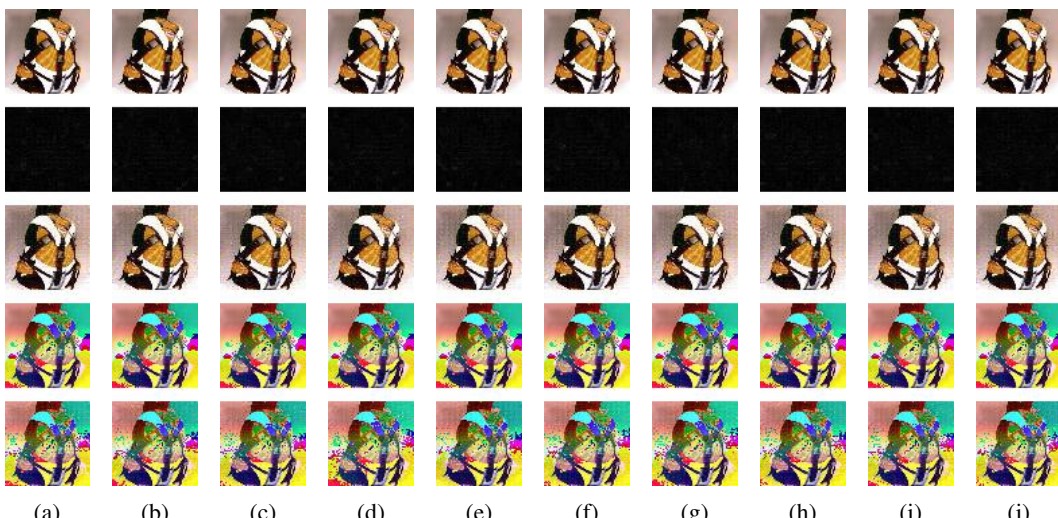

(a)  (b)  (c)  (d)  (e)  (f)  (g)  (h)  (i)  (j)

Figure 12: Visualization of ten different triggers (a)–(j) for the stealthiness study. For each case, the **first row** shows the **original images**, the **third row** shows the **images with triggers added**, and the **second row** presents the **residuals** between the original and the triggered images. The **fourth and fifth rows** display the **Grad-CAM heatmaps**, generated from the benign pre-trained model, for the original and triggered images, respectively. The average distortion introduced by the triggers is small, with an **average MSE of 0.0015** and an **average PSNR of 28.19 dB**.

**Grad-CAM Consistency.** The fourth and fifth rows of Figure 12 compare Grad-CAM heatmaps (Selvaraju et al., 2017) of original and poisoned images. Saliency patterns remain highly correlated, showing that triggers do not divert model attention toward conspicuous image regions. Instead, they subtly modulate internal features while preserving natural attribution patterns, reinforcing the covert nature of our perturbations.

**Representation Structure.** We further analyze stealthiness in representation space using t-SNE visualizations across CNNs (ResNet-18/34, VGG13-BN) and Transformers (ViT, SimpleViT), shown in Figure 11. Each panel depicts the distribution of poisoned samples under different models and paradigms. In benign embeddings, poisoned samples remain distributed within their original class manifolds, hindering simple outlier-based detection. Under backdoored models, poisoned samples form compact, target-aligned clusters: collapsing into a single region in $M \mapsto 1$, splitting into two stable groups in $M \mapsto N$ ($N = 2$), and separating into ten distinct clusters in $M \mapsto M$. This cluster behavior directly results from our optimization objective and ensures that stealthiness is maintained even in representation dimension.

**Overall Stealthiness Summary.** Across pixel, perceptual, saliency, and feature embedding views, Arcueid consistently achieves high imperceptibility. Perturbations remain subtle in the image domain, preserve natural attention maps, evade simple anomaly detectors, and embed smoothly within benign feature manifolds while constructing robust, paradigm-consistent decision regions. These results confirm that Arcueid is not only effective but also covert, a crucial property for realistic adaptive backdoor attacks.

## A.3 Further Arcueid Analysis

This section complements Section 4 by filling in details and providing a formal analysis of both the *optimization stage* and the *training-time execution stage*. All proofs refer to Appendix A.6.

### A.3.1 Optimization Analysis

Building on Section 4.2, we now provide a more formal analysis of the optimization stage, notations and assumptions follow the main text.

The goal is to characterize the gradient forces induced by the intra- and inter-cloud objectives, establish the existence of well-formed minimizers, and connect these properties to the feasibility and non-interference conditions defined earlier in Section 4.1.

**Gradients.** We characterize the exact gradient fields of the two terms, let $n_k = |\mathcal{B}_k|$ and abbreviate $\tilde{z}_i \equiv \tilde{z}_i^{(k)}$ for $(x_i, y_i) \in \mathcal{B}_k$.

**Lemma 4 (Exact Feature-level Gradients of $\mathcal{L}_{\text{intra}}$).** $\dfrac{\partial \mathcal{L}_{\text{intra}}}{\partial \tilde{z}_i} = \dfrac{2}{K\, n_k}\left(\tilde{z}_i - \mu_k\right).$

**Lemma 5 (Active-pair Gradients of $\mathcal{L}_{\text{inter}}$).** *If $\|\mu_k - \mu_\ell\| < m$, then*

$$\frac{\partial \mathcal{L}_{\text{inter}}}{\partial \mu_k} = -\frac{2}{K(K-1)} \frac{\mu_k - \mu_\ell}{\|\mu_k - \mu_\ell\|}, \qquad \frac{\partial \mathcal{L}_{\text{inter}}}{\partial \tilde{z}_i} = -\frac{1}{n_k} \frac{\partial \mathcal{L}_{\text{inter}}}{\partial \mu_k}, \quad i \in \mathcal{B}_k.$$

**Chain Rule to Triggers.** We further analyze how these gradients propagate to the trigger parameters via the chain rule,

$$\frac{\partial \mathcal{L}}{\partial \eta_k} = \sum_{(x_i, y_i) \in \mathcal{B}_k} \left( J_{g_{\eta_k}}(x_i)^\top J_\phi\big(g_{\eta_k}(x_i)\big)^\top \frac{\partial \mathcal{L}}{\partial \tilde{z}_i} \right), \qquad \mathcal{L} \in \{\mathcal{L}_{\text{intra}}, \mathcal{L}_{\text{inter}}\}, \tag{10}$$

with $J_\phi$ and $J_{g_{\eta_k}}$ the Jacobians of $\phi_\theta$ and $g_{\eta_k}$, respectively. Assuming both mappings are differentiable, the updates to $\eta_k$ inherit the attractive–repulsive dynamics characterized in Lemmas 4–5.

**Existence and Feasibility Guarantees.** We show that optimization admits non-degenerate minimizers and that these imply interior placement without interference.

**Proposition 3 (Existence of Minimizers and Non-collapse).** *If triggered features are bounded on the batch support and $m > 0$, then $F(\{\tilde{z}_i\}) = \mathcal{L}_{\text{intra}} + \lambda\, \mathcal{L}_{\text{inter}}$ (as a function of $\{\tilde{z}_i\}$) attains a minimum; any stationary point satisfies $\|\mu_k - \mu_\ell\| \geq m$ for all $k \neq \ell$ (otherwise an active hinge yields a nonzero repulsive gradient).*

**Proposition 4 (Radius/Separation $\Rightarrow$ Interior Placement).** *Let $f_\theta$ be fixed. Suppose at the post-optimization centers $\{\mu_k\}$ the fixed head exhibits a positive center gap to the designated targets: for every $k$ and $j \neq \tau_k$, $\Delta_{k,j}(\mu_k) = s_{\tau_k}(\mu_k) - s_j(\mu_k) \geq \gamma_{\text{logit}} > 0$, and for each cloud the logit gaps are $L$-Lipschitz locally. If $\mathcal{L}_{\text{intra}} \leq \varepsilon_{\text{intra}}$ (so $r_k \leq \sqrt{\varepsilon_{\text{intra}}}$) and $\mathcal{L}_{\text{inter}} = 0$ (so $\|\mu_k - \mu_\ell\| \geq m$), then every triggered point in cloud $k$ lies strictly in $\mathcal{R}_{\tau_k}$ with margin at least $\gamma_{\min} = \gamma_{\text{logit}} - L\sqrt{\varepsilon_{\text{intra}}} > 0$, and clouds do not interfere.*

**Parameter Sensitivity Implications.** To further examine parameter sensitivity, we have provided experimental evidence in Appendix A.2.4, and here we complement the analysis with theoretical insights.

**Proposition 5 (Shrinking $\mathcal{L}_{\text{intra}}$ Improves Interior Margin).** *Let the head be locally $L$-Lipschitz around $\mathcal{C}_k$ and suppose the center $\mu_k$ has logit gap $\gamma_{\text{logit}}(\mu_k) > 0$ to its designated target $\tau_k$. If $\mathcal{L}_{\text{intra}} \leq \varepsilon_{\text{intra}}$ so that $r_k \leq \sqrt{\varepsilon_{\text{intra}}}$, then every triggered point in $\mathcal{C}_k$ enjoys a target margin $\gamma_{\min} \geq \gamma_{\text{logit}}(\mu_k) - L\sqrt{\varepsilon_{\text{intra}}} > 0$.*

**Proposition 6 (Raising $\delta_{\min}$ Boosts Worst-case Success).** *Assume (i) clouds are isotropic with radii $\{r_k\}$, and (ii) class heads are locally smooth so decision boundaries move at most $L_b$ per unit feature perturbation. If $\delta_{\min} > r_k + r_\ell + \xi$ for all $k \neq \ell$ and some buffer $\xi > 0$, then cross-trigger interference probability is $0$ and the per-target misclassification rate is bounded above by a function decreasing in $\xi$. In particular, increasing $\delta_{\min}$ (by activating $\mathcal{L}_{\text{inter}}$) improves the worst-case target success and reduces the per-target instability.*

Propositions 5–6 explain the observed sweep in Figure 10: $\mathcal{L}_{\text{intra}}$ reduces radii and raises interior margins, while a modest $\mathcal{L}_{\text{inter}}$ selectively increases inter-center gaps for active pairs, improving worst-case target success and reducing variance; overly large $\mathcal{L}_{\text{inter}}$ over-activates the hinge and injects noisy repulsion, degrading effectiveness in multi-target paradigms.

**Optimization Dynamics.** Under standard smoothness of $\phi_\theta \circ g_{\eta_k}$ and bounded Jacobians, stochastic gradient updates on Equation 9 with diminishing stepsizes satisfy the usual nonconvex guarantee of asymptotic stationarity in $\eta$:

$$\frac{1}{T} \sum_{t=1}^{T} \mathbb{E}\big[\|\nabla_\eta \big(\lambda_{\text{intra}} \mathcal{L}_{\text{intra}} + \lambda_{\text{inter}} \mathcal{L}_{\text{inter}}\big)\|^2\big] \;\to\; 0 \quad (T \to \infty).$$

Combined with Proposition 3, this ensures convergence to non-collapsed stationary points where center separation is preserved, while Proposition 4 links such configurations to interior placement and non-interference. Moreover, the gradient structure in Lemmas 4–5 guarantees that updates consistently align with contraction–repulsion dynamics, maintaining small radii and enforcing pairwise margins. Since $\theta$ is fixed, all guarantees and margins are taken w.r.t. the *current* classifier; placement into a target region relies on the measured center gap $\gamma_{\text{logit}}$ at the optimized centers.

### A.3.2   Training-time Execution Analysis

In the main text we described the overall attack workflow in Section 4.3, but did not explicitly analyze how backdoor training proceeds under different threat paradigms. Here we provide a formal analysis of the training-time execution stage, showing how compact and separated clouds interact with gradient dynamics to yield paradigm-agnostic success.

**Execution Dynamics.** Once the trigger optimization produces stable feature clouds, their effect during empirical risk minimization can be examined through the gradients induced on the classifier head. The following results characterize how poisoned samples drive head parameters toward the intended mapping, both individually and collectively across multiple triggers.

**Lemma 6** (**Gradient Alignment on Triggered Clouds**). *Consider a poisoned example $(z, t)$ with $z \in \mathcal{C}_k$ and target label $t = \tau_k$, trained under any classification-calibrated loss $\ell(h(z), t)$ with head parameters $W$. Then the stochastic gradient update on $W$ has the form*

$$\nabla_W \ell \;=\; \Phi(z, t),$$

*where $\Phi$ is linear in $z$ and satisfies:*

- *the update of $w_t$ involves a negative multiple of $z$, thus* increasing *its alignment with $z$;*

- *the update of $w_j$, $j \neq t$, involves positive multiples of $z$, thus* reducing *their alignment with $z$.*

*Taking expectations over minibatches of triggered samples from $\mathcal{C}_k$, the net effect is to push $w_t$ toward the cloud center $\mu_k$ while pushing other weights away, thus enlarging the logit gap $\langle w_t - w_j, \mu_k \rangle$.*

**Lemma 7** (**Superposition Without Conflict Under Separation**). *If centers are separated ($\|\mu_k - \mu_\ell\| \geq m$) and radii small, the mean feature directions $\{\mu_k\}$ are sufficiently distinct, so the expected poisoned gradients from different clouds are approximately orthogonal and do not cancel. Hence, updates for heads $\{w_{\tau_k}\}$ add up: each $w_{\tau_k}$ is pulled toward its $\mu_k$, while repelled from other classes.*

**Unified success across paradigms.** Given compact and separated feature clouds, training with a classification-calibrated loss drives the model toward the intended backdoor mapping. By Lemma 6, stochastic gradients on triggered samples align the target head $w_{\tau_k}$ with its cloud center $\mu_k$ while repelling other heads, thereby enlarging the local logit gap. Lemma 7 further shows that when centers are well separated, gradient contributions from different clouds superpose without conflict, so updates across multiple triggers add constructively rather than cancel. Together with finite-sample persistence and realizability assumptions, this ensures that empirical risk minimization converges with high probability to the desired mapping.

This mechanism manifests consistently across paradigms: in the $M \mapsto M$ case, each cloud aligns to a distinct head; in $M \mapsto N$, several clouds jointly reinforce the same head; and in $M \mapsto 1$, all clouds converge on a single head, yielding unified alignment to the designated target region.

---

**Algorithm 1** *Joint Cloud Shaping Multi-trigger Optimization*

---

**Input:** Surrogate dataset $\mathcal{D}_{\text{sur}}$, Surrogate model $f_{\text{sur}} = h \circ \phi_\theta$, Number of triggers $K$, Steps $T$, Learning rate $\eta$, Margin $m$, Trade-offs $\lambda_{\text{intra}}, \lambda_{\text{inter}}$, Masks $\alpha$

**Output:** Optimized trigger family $G = \{g_{\eta_k}\}_{k=1}^{K}$

1: Initialize trigger patterns $\{v_k\}_{k=1}^{K} \sim \mathcal{N}(0,1)$
2: $\{g_{\eta_k}\}_{k=1}^{K} \leftarrow \{(\alpha, v_k)\}_{k=1}^{K}$, $\mu_k \leftarrow \mathbf{0}$
3: **for** $t = 1$ to $T$ **do**
4:     **for** batch $\{(x_i, y_i)\}_{i=1}^{m} \sim \mathcal{D}_{\text{sur}}$ **do**
5:         Sample pattern IDs $k_i \in \{1, \ldots, K\}$ for each $i$
6:         $x_i' \leftarrow g_{k_i}(x_i)$
7:         $z_i \leftarrow \phi_\theta(x_i')$
8:         $\mathcal{B}_k := \{i : k_i = k\}$, $\mathcal{K}_{\text{act}} := \{k : |\mathcal{B}_k| > 0\}$
9:         $\mu_k \leftarrow \frac{1}{|\mathcal{B}_k|} \sum_{i \in \mathcal{B}_k} z_i \quad \forall k \in \mathcal{K}_{\text{act}}$
10:        $\mathcal{L}_{\text{intra}} \leftarrow \frac{1}{K} \sum_k \frac{1}{|\mathcal{B}_k|} \sum_{i \in \mathcal{B}_k} \|z_i - \mu_k\|^2$
11:       $\mathcal{L}_{\text{inter}} \leftarrow \frac{2}{K(K-1)} \sum_{k < \ell} [m - \|\mu_k - \mu_\ell\|]_+$
12:       $\mathcal{L}_{\text{agg}} \leftarrow \lambda_{\text{intra}} \mathcal{L}_{\text{intra}} + \lambda_{\text{inter}} \mathcal{L}_{\text{inter}}$
13:       $v_k \leftarrow v_k - \eta \nabla_{v_k} \mathcal{L}_{\text{agg}}, \quad \forall k \in \{k_i\}$
14:       $\{g_{\eta_k}\}_{k=1}^{K} \leftarrow \{(\alpha, v_k)\}_{k=1}^{K}$
15:     **end for**
16: **end for**
17: **return** $\{g_{\eta_k}\}_{k=1}^{K}$

---

### A.3.3 PSEUDO CODE

Algorithm 1 explicitly operationalizes *Joint Cloud Shaping Multi-trigger Optimization* mechanism of `Arcueid` in Section 4.2. Lines 9–12 implement intra-cloud compactness and inter-cloud separation. The update in line 13-14 follows the chain rule in Equation 10, modifying only the learnable trigger patterns $v_k$ while keeping masks $\alpha_k$ fixed. By Proposition 3, these updates admit minimizers without center collapse, and Proposition 4 guarantees that sufficiently small radii and adequate separation yield interior placement and non-interference. Together, these steps instantiate the feasibility and non-interference conditions from Section 4.1 and ensure the reproducibility.

### A.4 ADAPTIVE DEFENSE ANALYSIS

Building on a clear understanding of the mechanisms underlying our proposed attack, `Arcueid`, this chapter introduces adaptive defense mechanism designed to directly counter the the attack. We then conduct a systematic evaluation of this defense, assessing its effectiveness and robustness.

### A.4.1 PROBLEM DEFINITION

So as `Arcueid` constructs a family of masked–blend triggers $\{g_{\eta_k}\}_{k=1}^{K}$ whose images induce compact, well-separated feature clouds $\mathcal{C}_k = \{\phi_\theta(g_{\eta_k}(x)) : (x, y) \sim \mathcal{D}, \pi(y) = k\}$ satisfying the feasibility constraints in Equation 5. In particular, each cloud $\mathcal{C}_k$ must lie strictly inside the decision region $R_{\tau_k}$ of the attacker-chosen target label $\tau_k$, with positive interior margin and non-overlap with other clouds. Our goal is to construct a defense that invalidates these feasibility conditions *for the same trigger family and perturbation budget* used by `Arcueid`.

Let $f_\theta = h \circ \phi_\theta$ be the classifier under defense, with representation map $\phi_\theta : \mathcal{X} \to \mathbb{R}^d$. We adopt the same masked–blend trigger family used by `Arcueid`:

$$\mathcal{S} = \Big\{ g_\eta(x) = \text{clip}\big((1 - \alpha) \odot x + \alpha \odot v\big) : \|\alpha\|_0 \leq s, \ \|g_\eta(x) - x\|_\infty \leq \varepsilon \Big\}. \tag{11}$$

For a clean example $(x, y) \sim \mathcal{D}$, define the *mask–robust margin*

$$\gamma_{\text{mask}}(x, y; \theta) := \inf_{\eta \in \mathcal{S}} \text{dist}\big(\phi_\theta(g_\eta(x)), \partial R_y\big), \qquad \Gamma_{\text{mask}}(\theta) := \inf_{(x,y) \sim \mathcal{D}} \gamma_{\text{mask}}(x, y; \theta). \tag{12}$$

If $\Gamma_{\mathrm{mask}}(\theta) > 0$, then no masked–blend trigger in $\mathcal{S}$ can push any clean feature $\phi_\theta(g_\eta(x))$ across a decision boundary into an incorrect region. The following proposition shows that in this case `Arcueid`'s multi-trigger construction becomes theoretically infeasible.

**Proposition 7** (**Mask–robust Margin Invalidates Trigger Clouds**). *If $\Gamma_{mask}(\theta) > 0$, then there exists no trigger family $\{g_{\eta_k}\} \subset \mathcal{S}$ and routing $\pi$ that can produce feature clouds $\{\mathcal{C}_k\}$ lying strictly inside $\{R_{\tau_k}\}$ as required by `Arcueid`'s feasibility constraints in Equation 5. Thus, `Arcueid`'s multi-trigger backdoor mapping is infeasible under $\Gamma_{mask}(\theta) > 0$.*

### A.4.2 OVERVIEW

We formulate the defense as a robust optimization problem:

$$\min_\theta \ R_{\mathrm{clean}}(\theta) + \lambda_{\mathrm{rob}} \, R_{\mathrm{rob}}(\theta), \qquad R_{\mathrm{rob}}(\theta) = \mathbb{E}_{(x,y)\sim\mathcal{D}}\Big[\max_{\eta\in\mathcal{S}} \ell\big(f_\theta(g_\eta(x)), y\big)\Big]. \tag{13}$$

The inner maximization searches for the most harmful masked–blend trigger in $\mathcal{S}$ for the current model, while the outer minimization updates $\theta$ to classify both clean and triggered examples correctly. The robust loss plays a direct geometric role: it controls the mask–robust margin $\Gamma_{\mathrm{mask}}(\theta)$.

**Proposition 8** (**Robust Loss Controls Mask–robust Margin**). *Under standard Lipschitz and monotonicity assumptions on logits and loss, if $R_{\mathrm{rob}}(\theta) \leq \varepsilon_{\mathrm{rob}}$, then*

$$\Gamma_{\mathrm{mask}}(\theta) \ \geq \ \frac{1}{L}\, \psi^{-1}(\varepsilon_{\mathrm{rob}}), \tag{14}$$

*where $L$ is the Lipschitz constant of the logits and $\psi^{-1}$ bounds the logit margin from the loss.*

This result shows that minimizing the robust loss directly increases a certified lower bound on $\Gamma_{\mathrm{mask}}(\theta)$, which by Proposition 7 breaks the feasibility of `Arcueid`'s clouds.

We implement Equation 13 using two mechanisms:

- **Adaptive Mitigation.** Starting from a possibly backdoored $f_{\theta_0}$, we iteratively learn an adversarial universal masked–blend trigger $\eta^\star$ via inner maximization over $\ell(f_\theta(g_\eta(x)), y)$, and fine-tune $\theta$ so that $f_\theta(g_{\eta^\star}(x))$ predicts the correct label. This locally increases $\gamma_{\mathrm{mask}}(x, y; \theta)$ around vulnerable examples.

- **Adaptive Training.** During training, each minibatch is augmented with an adversarially optimized universal trigger $\eta^\star$. Optimizing $\theta$ jointly on clean and triggered examples approximates the minimax problem and increases $\Gamma_{\mathrm{mask}}(\theta)$ globally.

Both mechanisms operate within the `Arcueid` trigger budget $(s, \varepsilon)$, ensuring apples-to-apples comparison in theory.

### A.4.3 DETAILED DESIGN

**Adversarial Trigger Update.** For $\{(x_i, y_i)\}_{i=1}^B$, we maintain a universal trigger parameter $\eta = (v, \alpha)$ and perform projected gradient ascent:

$$\eta \leftarrow \Pi_\mathcal{S}\left[\eta + \rho \, \nabla_\eta \frac{1}{B} \sum_{i=1}^B \ell\big(f_\theta(g_\eta(x_i)), y_i\big)\right], \tag{15}$$

where $\Pi_\mathcal{S}$ projects back to the masked–blend trigger family. This step identifies the most vulnerable masked direction for the current $\theta$.

**Robust Parameter Update.** Given the updated trigger $\eta^\star$, model parameters are updated via

$$\theta \leftarrow \theta - \gamma \, \nabla_\theta\left[\frac{1}{B} \sum_{i=1}^B \ell(f_\theta(x_i), y_i) + \lambda_{\mathrm{rob}} \ell(f_\theta(g_{\eta^\star}(x_i)), y_i)\right], \tag{16}$$

which moves triggered features back toward their correct regions $R_{y_i}$ and expands the mask–robust margin.

Table 10: **Mitigation defense extension.**

| Attack Type | Defense Type | CA | ASR |
|---|---|---|---|
| $M \mapsto 1$ Attack | No Defense | 87.8% | 99.6%±0.3% |
| | FineTuning | 87.3% | 99.8%±0.2% |
| | Pruning | 87.3% | 99.8%±0.2% |
| | **Adaptive Mitigation** | **66.4%** | **21.8%±7.7%** |
| $M \mapsto N$ Attack | No Defense | 87.7% | 99.7%±0.4% |
| | FineTuning | 86.8% | 99.7%±0.5% |
| | Pruning | 86.7% | 99.6%±0.5% |
| | **Adaptive Mitigation** | **52.5%** | **16.4%±12.9%** |
| $M \mapsto M$ Attack | No Defense | 89.2% | 99.4%±0.8% |
| | FineTuning | 88.6% | 99.2%±1.1% |
| | Pruning | 88.4% | 98.5%±2.2% |
| | **Adaptive Mitigation** | **58.2%** | **7.5%±5.8%** |

Table 11: **Adaptive training analysis.**

| Attack Type | Defense type | CA | ASR |
|---|---|---|---|
| $M \mapsto 1$ Attack | No Defense | 87.8% | 99.6%±0.3% |
| | **Adaptive Training** | **65.2%** | **7.8%±2.4%** |
| $M \mapsto N$ Attack | No Defense | 87.7% | 99.7%±0.4% |
| | **Adaptive Training** | **63.4%** | **10.0%±6.8%** |
| $M \mapsto M$ Attack | No Defense | 89.2% | 99.4%±0.8% |
| | **Adaptive Training** | **66.8%** | **6.8%±8.3%** |

**Effect on Trigger Cloud Geometry.** Under the smoothness assumptions used in Section 4, increasing $\Gamma_{\mathrm{mask}}(\theta)$ prevents any collection of masked triggers $\{g_{\eta_k}\} \subset \mathcal{S}$ from generating wrong-label clouds $\{\mathcal{C}_k\}$ that are (i) compact, (ii) mutually separated, and (iii) strictly inside attacker-chosen regions $\{R_{\tau_k}\}$ with positive interior margin. Thus Equation 5 becomes infeasible and `Arcueid`'s multi-trigger backdoor mechanism collapses.

**Robustness–accuracy Tradeoff.** Because Equation 16 forces the classifier to be insensitive to all masked–blend perturbations in $\mathcal{S}$, it necessarily suppresses certain localized directions that are genuinely discriminative in clean data. The following proposition formalizes this inherent cost.

**Proposition 9** (**Robustness–accuracy Tradeoff under Masked–blend Defense**). *If the Bayes-optimal classifier $f^\star$ is not robust to $\mathcal{S}$ on a subset of $\mathcal{A} \subseteq \mathcal{X}$ of probability mass $\nu > 0$, then any model $f_\theta$ with $\Gamma_{\mathrm{mask}}(\theta) \geq \gamma > 0$ must incur strictly higher standard risk:*

$$R_{\mathrm{clean}}(\theta) \geq R_{\mathrm{clean}}(f^\star) + \alpha\nu, \tag{17}$$

*for some $\alpha > 0$ depending on the geometry of $\{R_c\}$. Hence substantial robustness necessarily induces a drop in clean accuracy.*

A.4.4 EVALUATION

For evaluation, we instantiated both adaptive defenses on default setting aligned with the detail in Appendix A.2.1 and compare them with standard mitigation such as FineTuning (Liu et al., 2018) and Pruning (Liu et al., 2018)

As shown in Tables 10 and 11, these generic mitigations have negligible effect and the ASR remains above 98% across all paradigms. In contrast, Adaptive Mitigation reduces ASR to 21.8% and Adap-

tive Training further lowers it to 7.8% in the $M \mapsto 1$ setting. However, these reductions come with a severe cost. CA drops from approximately 88% to the range 52% to 67%. This reflects an inherent phenomenon: attacker-aware adaptive defenses must train the model to be insensitive to an entire family of masked blend perturbations, which inevitably suppresses discriminative features required for normal classification. Consequently, such defenses reduce ASR only by incurring a substantial degradation in CA, far beyond what practical and attack agnostic defenses would accept.

## A.5 BRIDGING THEORY AND PRACTICE

This section expands on the discussion in Section 4 and addresses in more formal terms the relation between the static formulation in Equation 9 and the practical setting where the victim model is obtained by training on poisoned data.

### A.5.1 STABILITY OF CLOUD GEOMETRY

Recall that Section 4 is intentionally stated in a static form: for a *fixed* parameter vector $\theta$, Propositions 1–6 and Equation 9 characterize when the trigger-induced clouds

$$\mathcal{C}_k(\theta) = \big\{ \phi_\theta(g_{\eta_k}(x)) : (x,y) \sim \mathcal{D}, \, \pi(y) = k \big\}$$

are (i) compact, (ii) mutually separated, and (iii) strictly contained in the target decision regions $R_{\tau_k}(\theta)$. The victim's training procedure is treated as a black-box map from the poisoned dataset to a final parameter $\theta_T$, and the guarantees of Section 4.3 are conditional on $\theta_T$ satisfying the representation-alignment assumptions with the surrogate parameter $\theta_S$.

**Observation.** We make explicit two ingredients that connect this static picture to practice: a margin-based perspective showing that, under any generic margin-based loss, a final model that does not increase empirical risk or aggregate clean loss cannot reduce the minimum margin over poisoned points, and therefore cannot push triggered features back toward decision boundaries, while a local stability property of the cloud-margin lower bound $\underline{\gamma}(\theta)$ under parameter perturbations, together with the alignment condition of Proposition 2.

Together, these observations formalize why the cloud structure created by Equation 9 on the surrogate is not destroyed, and in many cases is reinforced, by subsequent training of the victim model on poisoned data.

**Margin-based Reinforcement of Poisoned Margins** For any fixed parameter $\theta$, Section 4.2 defines the per-trigger margins and the aggregate lower bound

$$\underline{\gamma}(\theta) := \min_k \mathrm{margin}_{\tau_k}\big(\mathcal{C}_k(\theta)\big),$$

which appears in Equation 9 as the quantity that `Arcueid` seeks to enlarge on the surrogate model $f_{\theta_S} = h_{\theta_S} \circ \phi_{\theta_S}$.

We now consider a generic margin-based training objective that jointly accounts for clean and poisoned examples.

Let the training set be split into clean and poisoned subsets $\mathcal{D}_{\mathrm{clean}}$ and $\mathcal{D}_{\mathrm{poison}}$. For any parameter vector $\theta$, define the empirical risk

$$\mathcal{R}(\theta) = \frac{1}{|\mathcal{D}_{\mathrm{clean}} \cup \mathcal{D}_{\mathrm{poison}}|} \sum_{(x,y) \in \mathcal{D}_{\mathrm{clean}} \cup \mathcal{D}_{\mathrm{poison}}} \ell\big(\Gamma_y(x;\theta)\big), \tag{18}$$

where $\Gamma_y(x;\theta)$ is the decision margin for label $y$ at $x$, and $\ell : \mathbb{R} \to \mathbb{R}_+$ is any strictly decreasing, continuous margin-based surrogate (a standard assumption in classification).

We do *not* assume any particular optimizer or update rule, we only assume that the victim training procedure outputs a parameter $\theta_T$ whose empirical risk $\mathcal{R}(\theta_T)$ is not larger than that of a reference parameter $\theta_{\mathrm{ref}}$ and does not worsen the total loss on clean points.

We focus on the minimum margin over poisoned points:

$$\gamma_{\mathrm{poison}}(\theta) := \min_{(x,y) \in \mathcal{D}_{\mathrm{poison}}} \Gamma_y(x;\theta). \tag{19}$$

**Proposition 10** (**Monotone Behavior of Poisoned Margins**). *Let $\mathcal{D}_{\text{poison}}$ consist of triggered examples $(z_k, \tau_k)$ with $z_k = g_{\eta_k}(x)$ and target labels $\tau_k$, and let $\gamma_{\text{poison}}(\theta)$ be defined as in Equation 19. Suppose $\ell$ in Equation 18 is strictly decreasing, and there exist two parameter vectors $\theta_{\text{ref}}$ and $\theta_T$ such that:*

*1. **Global Risk Non-increase***

$$\mathcal{R}(\theta_T) \ \leq \ \mathcal{R}(\theta_{\text{ref}}).$$

*2. **Clean Loss Non-increase***

$$\sum_{(x,y)\in\mathcal{D}_{\text{clean}}} \ell\big(\Gamma_y(x;\theta_T)\big) \ \leq \ \sum_{(x,y)\in\mathcal{D}_{\text{clean}}} \ell\big(\Gamma_y(x;\theta_{\text{ref}})\big).$$

*Then the minimum poisoned margin cannot decrease:*

$$\gamma_{\text{poison}}(\theta_T) \ \geq \ \gamma_{\text{poison}}(\theta_{\text{ref}}).$$

In words, Proposition 10 states that, under any strictly decreasing margin-based loss, *any* final model that (i) does not worsen empirical risk and (ii) does not worsen clean loss in aggregate cannot systematically reduce the margins of poisoned points. Equivalently, training cannot push triggered features closer to the decision boundaries in a way that would increase their loss, and the only risk-neutral directions are those that keep or enlarge poisoned margins. This is exactly the sense in which victim training tends to *reinforce* rather than destroy the margins that `Arcueid` initializes via Equation 9.

**Local Stability of Cloud Margins and Surrogate–Victim Alignment**   We now connect the poisoned-margin behavior above to the cloud-margin lower bound $\underline{\gamma}(\theta)$ used in Section 4.2 and to the surrogate–victim alignment condition in Proposition 2.

Under the local Lipschitz assumptions on $\phi_\theta$, $h_\theta$ and the decision boundaries $\partial R_c(\theta)$ made in Section 4, the map $\theta \mapsto \underline{\gamma}(\theta)$ is locally Lipschitz:

**Lemma 8** (**Local Stability of Cloud-margin Lower Bound**). *There exists $L_\gamma > 0$ (depending only on the Lipschitz constants of $\phi_\theta$, $h_\theta$ and on the alignment parameters $(A, \delta, \varepsilon_h)$ introduced in Section 4.3) such that, for any two parameters $\theta, \theta'$ in the neighborhood considered in Section 4,*

$$\big|\underline{\gamma}(\theta') - \underline{\gamma}(\theta)\big| \ \leq \ L_\gamma \, \|\theta' - \theta\|. \tag{20}$$

Lemma 8 formalizes the intuition that the positive buffer $\underline{\gamma}(\theta_{\text{ref}})$ created by Equation 9 on the surrogate is robust to moderate changes in $\theta$: so long as $\|\theta' - \theta_{\text{ref}}\|$ remains small, the lower bound $\underline{\gamma}(\theta')$ cannot collapse to zero.

Proposition 2 then connects this local stability on the surrogate to the final victim model: any $\theta_T$ whose representation is aligned with $\theta_S$ in the sense of the $(A, \delta, \varepsilon_h)$ condition lies in a region where $\underline{\gamma}(\theta_T)$ remains positive and the multi-trigger mapping is preserved.

Combining Proposition 10 with Lemma 8 yields the following conceptual picture: Equation 9 constructs a reference parameter $\theta_{\text{ref}}$ (on the surrogate) with $\underline{\gamma}(\theta_{\text{ref}}) > 0$; any victim model $\theta_T$ that (i) is not worse in empirical risk, (ii) does not worsen clean loss in aggregate, and (iii) remains in the aligned neighborhood of $\theta_{\text{ref}}$ in the sense of Section 4.3, must preserve or enlarge the margins of poisoned points and hence maintain a positive cloud-margin lower bound $\underline{\gamma}(\theta_T) > 0$. This explains why in practice the trigger clouds remain compact and well separated across training and, in many cases, become more pronounced, exactly as observed in Figure 11.

### A.5.2 Sensitivity of Representation Misalignment

We then elaborates the sensitivity to deviations in representation alignment between the surrogate and victim models.

Recall the alignment model used in Proposition 2: we assume that the victim representation $\phi_{\theta_T}$ is approximately aligned with the surrogate representation $\phi_{\theta_S}$ via a bounded linear map $A$ and a small additive discrepancy:

$$\phi_{\theta_T}(x) = A\,\phi_{\theta_S}(x) + \epsilon(x), \qquad \|\epsilon(x)\| \leq \delta \quad \forall x. \tag{21}$$

Here $A : \mathbb{R}^d \to \mathbb{R}^d$ is linear and $\delta \geq 0$ quantifies the worst-case representation mismatch.

On the surrogate model $f_{\theta_S} = h \circ \phi_{\theta_S}$, the trigger clouds

$$\mathcal{C}_k^{(S)} = \left\{ \phi_{\theta_S}(g_{\eta_k}(x)) : (x, y) \sim \mathcal{D}, \pi(y) = k \right\}$$

are assumed to be *feasible* in the sense of Sec. 4: each cloud lies strictly inside the decision region $R_{\tau_k}(\theta_S)$, is compact, and is separated from the boundaries with a positive margin. We denote the cloud center and radius by

$$\mu_k^{(S)} = \mathbb{E}\left[ \phi_{\theta_S}(g_{\eta_k}(x)) \mid \pi(y) = k \right], \qquad r_k^{(S)} = \sup_{u \in \mathcal{C}_k^{(S)}} \left\| u - \mu_k^{(S)} \right\|.$$

Let $\gamma_{\theta_S}$ denote the surrogate cloud-margin lower bound used in Section 4.2:

$$\gamma_{\theta_S} := \min_k \operatorname{margin}_{\tau_k}\left( \mathcal{C}_k^{(S)} \right), \tag{22}$$

where for a cloud $\mathcal{C}_k$ we define

$$\operatorname{margin}_{\tau_k}\left( \mathcal{C}_k \right) := \inf_{u \in \mathcal{C}_k} \operatorname{dist}\left( u, \partial R_{\tau_k}(\theta) \right).$$

For the classifier head $h$ we assume a standard Lipschitz control on the geometry of decision regions, where exists $L_h > 0$ such that for any two feature vectors $u, u' \in \mathbb{R}^d$ and any class $c$, the signed distance to the decision boundary $\partial R_c(\theta)$ satisfies

$$\left| \operatorname{margin}_c(u) - \operatorname{margin}_c(u') \right| \leq L_h \| u' - u \|.$$

Equivalently, the (unsigned) distance to the boundary is $L_h$-Lipschitz in $u$.

Under this assumption, Proposition 2 shows that Arcueid's trigger clouds remain feasible on the target model whenever the alignment parameters $(A, \delta)$ satisfy

$$L_h \| A \| \delta < \gamma_{\theta_S}, \tag{23}$$

**Cloud Geometry under Linear Alignment.** We first characterize how the cloud centers and radii transform from the surrogate to the target under the alignment model (Equation 21).

For a fixed trigger index $k$, define the target cloud

$$\mathcal{C}_k^{(T)} = \left\{ \phi_{\theta_T}(g_{\eta_k}(x)) : (x, y) \sim \mathcal{D}, \pi(y) = k \right\}.$$

By Equation 21, every $u \in \mathcal{C}_k^{(S)}$ is mapped to

$$v = \phi_{\theta_T}(g_{\eta_k}(x)) = Au + \epsilon(x) \in \mathcal{C}_k^{(T)}.$$

Let $\mu_k^{(T)}$ and $r_k^{(T)}$ denote the center and radius of $\mathcal{C}_k^{(T)}$:

$$\mu_k^{(T)} = \mathbb{E}\left[ \phi_{\theta_T}(g_{\eta_k}(x)) \mid \pi(y) = k \right], \qquad r_k^{(T)} = \sup_{v \in \mathcal{C}_k^{(T)}} \left\| v - \mu_k^{(T)} \right\|.$$

**Lemma 9** (**Transformation of Cloud Centers and Radii**). *Under the alignment model, for each trigger index $k$ there exists a vector $\bar{\epsilon}_k$ with $\| \bar{\epsilon}_k \| \leq \delta$ such that*

$$\mu_k^{(T)} = A\mu_k^{(S)} + \bar{\epsilon}_k, \tag{24}$$

*and the target radius is bounded by*

$$r_k^{(T)} \leq \| A \| r_k^{(S)} + \delta. \tag{25}$$

**Margin Degradation under Misalignment.** We now relate the target cloud margins to the surrogate cloud margins via the previous assumption and Lemma 9. Let

$$\gamma_{\theta_T} := \min_k \operatorname{margin}_{\tau_k}\left( \mathcal{C}_k^{(T)} \right)$$

denote the target cloud-margin lower bound.

**Proposition 11** (**Sensitivity of Transfer Margin to Misalignment**). *Under the alignment model, for each trigger index $k$ the margin of $\mathcal{C}_k^{(T)}$ on the target model admits the bound*

$$\operatorname{margin}_{\tau_k}\left( \mathcal{C}_k^{(T)} \right) \geq \operatorname{margin}_{\tau_k}\left( \mathcal{C}_k^{(S)} \right) - L_h\left( \left| \| A \| - 1 \right| r_k^{(S)} + \| A \| \delta \right), \tag{26}$$

*and consequently*

$$\gamma_{\theta_T} \geq \gamma_{\theta_S} - L_h\left( \max_k \left| \| A \| - 1 \right| r_k^{(S)} + \| A \| \delta \right). \tag{27}$$

**Interpretation and Practical Implications.** Proposition 11 makes the informal discussion in the rebuttal precise. The degradation of the target cloud-margin lower bound $\gamma_{\theta_T}$ relative to the surrogate margin $\gamma_{\theta_S}$ is controlled *linearly* by:

- the deviation of $\|A\|$ from 1, i.e., how much the linear map rescales feature space;

- the surrogate cloud radii $r_k^{(S)}$, i.e., how compact the clouds are on the surrogate;

- the representation discrepancy $\delta$, i.e., how far the victim features deviate from the aligned linear image of the surrogate features.

In particular, as long as

$$L_h\Big(\max_k |\|A\| - 1| \, r_k^{(S)} + \|A\|\delta\Big) \;\ll\; \gamma_{\theta_S},$$

the target margin $\gamma_{\theta_T}$ remains positive and the multi-trigger backdoor mapping remains feasible. Only when the combined shift satisfies

$$L_h\Big(\max_k |\|A\| - 1| \, r_k^{(S)} + \|A\|\delta\Big) \;\approx\; \gamma_{\theta_S}$$

does the margin collapse and transferability become unreliable.

Because `Arcueid` explicitly optimizes for compact surrogate clouds (small $r_k^{(S)}$) with large margins $\gamma_{\theta_S}$, the sensitivity term in Equation 27 is naturally attenuated.

## A.6 PROOFS

Throughout, all norms and distances are taken in the representation space $\mathcal{Z}$, and the classifier $f_\theta = h \circ \phi_\theta$ is fixed. For a nonempty closed set $B \subseteq \mathcal{Z}$, we write $\mathrm{dist}(z, B) := \inf_{u \in B} \|z - u\|$. For class $t$, define the decision region $\mathcal{R}_t := \{z \in \mathcal{Z} : \arg\max h(z) = t\}$ with boundary $\partial \mathcal{R}_t$. We use two basic facts: (F1) if $\mathrm{dist}(z, \partial \mathcal{R}_t) > 0$ then $z$ lies in the open interior of $\mathcal{R}_t$; (F2) $\mathrm{dist}(z, B) \geq \mathrm{dist}(u, B) - \|z - u\|$ (triangle inequality). We also adopt the notation introduced in Section 4.2. Finally, let $s(z) \in \mathbb{R}^Q$ denote the pre-softmax score vector of the head $h$ at feature $z$; since softmax is order-preserving, $\arg\max_c s_c(z) = \arg\max_c(h(z))_c$. When defining logit gaps, we write $\Delta_{t,j}(z) := s_t(z) - s_j(z)$.

### A.6.1 PROOF OF PROPOSITION 1

**Assumptions.** For each trigger $k$: (i) the triggered cloud $\mathcal{C}_k$ is well-defined with center $\mu_k$ and radius $r_k \geq 0$ (i.e., $\|x - \mu_k\| \leq r_k$ for all $x \in \mathcal{C}_k$); (ii) either (A) $\mathrm{margin}_{\tau_k}(\mathcal{C}_k) \geq \gamma_k > 0$ (*margin form*), or (B) $\mathrm{dist}(\mu_k, \partial \mathcal{R}_{\tau_k}) > r_k$ (*center–radius form*).

**Step-by-step Proof.**

(1) *Margin form $\Rightarrow$ success.* Fix $k$ and $x \in \mathcal{C}_k$. Then $\mathrm{dist}(x, \partial \mathcal{R}_{\tau_k}) \geq \gamma_k > 0$, so by (F1) $x$ lies in the interior of $\mathcal{R}_{\tau_k}$ and is classified as $\tau_k$.

(2) *Center–radius sufficiency.* If $\mathrm{dist}(\mu_k, \partial \mathcal{R}_{\tau_k}) > r_k$ and $\|x - \mu_k\| \leq r_k$, then by (F2),

$$\mathrm{dist}(x, \partial \mathcal{R}_{\tau_k}) \;\geq\; \mathrm{dist}(\mu_k, \partial \mathcal{R}_{\tau_k}) - \|x - \mu_k\| \;>\; 0.$$

Again by (F1), $x$ is strictly inside $\mathcal{R}_{\tau_k}$ and predicted as $\tau_k$. Moreover, $\mathrm{margin}_{\tau_k}(\mathcal{C}_k) \geq \mathrm{dist}(\mu_k, \partial \mathcal{R}_{\tau_k}) - r_k$.

### A.6.2 PROOF OF LEMMA 1

**Assumptions.** For two distinct triggers $k \neq \ell$: (i) $\mathrm{margin}_{\tau_k}(\mathcal{C}_k) \geq \gamma_k > 0$ and $\mathrm{margin}_{\tau_\ell}(\mathcal{C}_\ell) \geq \gamma_\ell > 0$; (ii) centers $\mu_k, \mu_\ell$ and radii $r_k, r_\ell$ satisfy $\|\mu_k - \mu_\ell\| > r_k + r_\ell$.

**Step-by-step Proof.**

(1) *Cloud disjointness.* For any $x \in \mathcal{C}_k$, $y \in \mathcal{C}_\ell$,
$$\|x - y\| \geq \|\mu_k - \mu_\ell\| - \|x - \mu_k\| - \|y - \mu_\ell\| > (r_k + r_\ell) - r_k - r_\ell = 0,$$
so $\mathcal{C}_k \cap \mathcal{C}_\ell = \varnothing$.

(2) *Interior stability.* By Proposition 1, every $x \in \mathcal{C}_k$ lies in the interior of $\mathcal{R}_{\tau_k}$ and every $y \in \mathcal{C}_\ell$ lies in the interior of $\mathcal{R}_{\tau_\ell}$ with strictly positive margins. Interiors of distinct decision regions are disjoint; thus predictions on $\mathcal{C}_k$ (resp. $\mathcal{C}_\ell$) cannot flip to $\tau_\ell$ (resp. $\tau_k$) without crossing a boundary, which is precluded by the positive margins. Hence there is no cross-trigger interference.

### A.6.3 PROOF OF LEMMA 2

**Assumptions.** Let $S = (z_i)_{i=1}^n$ be the clean training set and $S'$ be obtained by replacing at most $m \leq \rho n$ examples with poisoned ones. Let $A(\cdot)$ be the learning algorithm returning $\hat{\theta}(\cdot)$. Assume: (i) *uniform stability*: for any datasets $U, V$ that differ in one example and any $z$, $|\ell(\hat{\theta}(U); z) - \ell(\hat{\theta}(V); z)| \leq \beta_n$; (ii) bounded loss: $0 \leq \ell(\cdot; z) \leq L_{\max}$.

**Step-by-step Proof.**

(1) *Path coupling.* Construct $S = S^{(0)}, S^{(1)}, \ldots, S^{(m)} = S'$ where each $S^{(t)}$ differs from $S^{(t-1)}$ by one example. For any $z$,
$$|\ell(\hat{\theta}(S); z) - \ell(\hat{\theta}(S'); z)| \leq \sum_{t=1}^m |\ell(\hat{\theta}(S^{(t-1)}); z) - \ell(\hat{\theta}(S^{(t)}); z)| \leq m\,\beta_n \leq \rho n\,\beta_n.$$

(2) *Expected clean risk difference.* Taking expectation over $z \sim \mathcal{D}$ yields
$$\left| \mathbb{E}_{\mathcal{D}}[\ell(\hat{\theta}(S); z)] - \mathbb{E}_{\mathcal{D}}[\ell(\hat{\theta}(S'); z)] \right| \leq \rho n\,\beta_n.$$

(3) *Accounting for empirical replacement.* ERM-type procedures also incur at most $\rho$ fraction of examples whose losses may change by up to $L_{\max}$ between $S$ and $S'$, producing an additive $\rho L_{\max}$ term in standard stability-to-generalization bounds.

Combining (2) and (3): the expected clean risk changes by at most $\rho n\,\beta_n + \rho L_{\max}$, which is $O(\rho)$ when $\beta_n = O(1/n)$.

### A.6.4 PROOF OF PROPOSITION 2

**Assumptions.** Let $f_s = h_s \circ \phi_s$ (surrogate) and $f_t = h_t \circ \phi_t$ (target). Assume: (i) *surrogate margin*: for all triggered $x$, and all $j \neq \tau$, the score gap $\Gamma_s(x) := (h_s(\phi_s(x)))_\tau - (h_s(\phi_s(x)))_j \geq \gamma > 0$; (ii) *feature alignment*: there exists a bounded linear $A$ with $\|\phi_t(x) - A\phi_s(x)\| \leq \delta$ for all triggered $x$; (iii) *head alignment*: $\|h_t(Az) - h_s(z)\|_\infty \leq \varepsilon_h$ for all surrogate features $z$ on the triggered support; (iv) *Lipschitz head*: $h_t$ is $L_h$-Lipschitz: $\|h_t(u) - h_t(v)\|_\infty \leq L_h \|u - v\|$.

**Step-by-step Proof.**

(1) *Decompose target score gap.* For triggered $x$ and any $j \neq \tau$,
$$\Gamma_t(x) := (h_t(\phi_t(x)))_\tau - (h_t(\phi_t(x)))_j$$
$$= \underbrace{(h_t(A\phi_s(x)))_\tau - (h_t(A\phi_s(x)))_j}_{\text{aligned target gap}} + \Delta_1(x),$$
where $\Delta_1(x) = (h_t(\phi_t(x)) - h_t(A\phi_s(x)))_\tau - (h_t(\phi_t(x)) - h_t(A\phi_s(x)))_j$.

(2) *Compare aligned target gap with surrogate gap.* By head alignment (iii),
$$|(h_t(A\phi_s(x)))_\tau - (h_s(\phi_s(x)))_\tau| \leq \varepsilon_h, \quad |(h_t(A\phi_s(x)))_j - (h_s(\phi_s(x)))_j| \leq \varepsilon_h,$$
hence $(h_t(A\phi_s(x)))_\tau - (h_t(A\phi_s(x)))_j \geq \Gamma_s(x) - 2\varepsilon_h \geq \gamma - 2\varepsilon_h$.

(3) *Bound the misalignment term.* By Lipschitzness (iv) and feature alignment (ii),

$$|\Delta_1(x)| \leq \|h_t(\phi_t(x)) - h_t(A\phi_s(x))\|_\infty \leq L_h \|\phi_t(x) - A\phi_s(x)\| \leq L_h \delta.$$

(4) *Conclude preserved decision.* Combining (1)–(3),

$$\Gamma_t(x) \geq (\gamma - 2\varepsilon_h) - L_h \delta.$$

If $\gamma > 2\varepsilon_h + L_h \delta$, then $\Gamma_t(x) > 0$ for all $j \neq \tau$, so $\arg\max f_t(x) = \tau$ on all triggered inputs.

**Corollary.** If head alignment is exact on the aligned surrogate features, i.e., $\varepsilon_h = 0$, then the bound reduces to $\Gamma_t(x) \geq \gamma - L_h \delta$. In particular, if the feature-alignment bound is expressed as $\|\phi_t(x) - A\phi_s(x)\| \leq \|A\| \delta'$ for some surrogate-domain discrepancy $\delta'$, a sufficient condition is

$$\gamma > L_h \|A\| \delta',$$

### A.6.5 PROOF OF LEMMA 3

**Assumptions.** For each class $c \in \mathcal{Y}$: (i) the clean feature $\phi_\theta(x) \mid (y = c)$ is sub-Gaussian with mean $\bar{\mu}_c$ and parameter $\sigma^2$ (i.e., $\langle u, \phi_\theta(x) - \bar{\mu}_c \rangle$ is sub-Gaussian with proxy variance $\sigma^2$ for all $u \in \mathbb{S}^{d-1}$); (ii) we have $n_c$ i.i.d. samples per class and empirical mean $\hat{\mu}_c$.

**Step-by-step Proof.**

(1) *Concentration of empirical mean.* By vector Bernstein / sub-Gaussian concentration, there exist absolute constants $C_1, C_2 > 0$ such that for all $t > 0$,

$$\Pr\big(\|\hat{\mu}_c - \bar{\mu}_c\| \geq t\big) \leq 2\exp\Big(-C_1 n_c t^2 / \sigma^2\Big), \qquad \mathbb{E}\|\hat{\mu}_c - \bar{\mu}_c\| \leq C_2 \sigma n_c^{-1/2}.$$

(2) *Estimability of margin/separation constraints.* Hence $\|\hat{\mu}_c - \bar{\mu}_c\| = O_p(n_c^{-1/2})$. Constraints phrased using $\bar{\mu}_c$ (e.g., requiring a triggered center $\mu_k$ to lie at least $\gamma$ inside $\mathcal{R}_{\tau_k}$ and away from neighborhoods of clean centroids) can be replaced by their empirical versions with vanishing estimation error $O_p(n_c^{-1/2})$ as $n_c$ grows, validating optimization with finite subsamples.

### A.6.6 PROOF OF LEMMA 4

**Assumptions.** For each cloud $k$ in a minibatch: (i) we have $n_k = |\mathcal{B}_k| > 0$ samples with empirical center $\mu_k = \frac{1}{n_k} \sum_{i \in \mathcal{B}_k} \tilde{z}_i$ and intra-cloud loss $\mathcal{L}_{\text{intra},k} = \frac{1}{n_k} \sum_{i \in \mathcal{B}_k} \|\tilde{z}_i - \mu_k\|^2$; (ii) differentiation is taken with respect to a single triggered feature $\tilde{z}_a$ where $(x_a, y_a) \in \mathcal{B}_k$, while all other batch entries are fixed.

**Step-by-step Proof.**

(1) Expand $\mathcal{L}_{\text{intra},k}$:

$$\mathcal{L}_{\text{intra},k} = \frac{1}{n_k} \sum_i \big(\|\tilde{z}_i\|^2 - 2\tilde{z}_i^\top \mu_k + \|\mu_k\|^2\big).$$

(2) Since $\partial\mu_k / \partial\tilde{z}_a = \frac{1}{n_k} I$, differentiate termwise:

$$\frac{\partial\mathcal{L}_{\text{intra},k}}{\partial\tilde{z}_a} = \frac{1}{n_k}\Big(2\tilde{z}_a - 2\mu_k\Big) + \frac{1}{n_k} \sum_i \Big(-2\tilde{z}_i^\top + 2\mu_k^\top\Big) \frac{1}{n_k} I.$$

(3) Use $\sum_i (\tilde{z}_i - \mu_k) = 0$ to cancel the sum, yielding $\frac{\partial\mathcal{L}_{\text{intra},k}}{\partial\tilde{z}_a} = \frac{2}{n_k}(\tilde{z}_a - \mu_k)$.

(4) Averaging over $k$ scales by $1/K$, giving $\dfrac{\partial\mathcal{L}_{\text{intra}}}{\partial\tilde{z}_a} = \dfrac{2}{K n_k}(\tilde{z}_a - \mu_k)$.

### A.6.7 PROOF OF LEMMA 5

**Assumptions.** (i) The inter-cloud loss is defined as $\mathcal{L}_{\text{inter}} = \frac{2}{K(K-1)} \sum_{k<\ell} [m - \|\mu_k - \mu_\ell\|]_+$ with margin $m > 0$; (ii) we consider a pair $(k, \ell)$ with $\|\mu_k - \mu_\ell\| < m$ (active hinge), and note that at $\|\mu_k - \mu_\ell\| = m$ any subgradient suffices so the formulas hold almost everywhere; (iii) each cloud center is $\mu_k = \frac{1}{n_k} \sum_{i \in \mathcal{B}_k} \tilde{z}_i$, giving $\partial \mu_k / \partial \tilde{z}_i = \frac{1}{n_k} I$ for $i \in \mathcal{B}_k$.

**Step-by-step Proof.**

(1) For an active pair, the contribution is $\frac{2}{K(K-1)}(m - \|\mu_k - \mu_\ell\|)$.

(2) Since $\frac{\partial}{\partial \mu_k} \|\mu_k - \mu_\ell\| = \frac{\mu_k - \mu_\ell}{\|\mu_k - \mu_\ell\|}$, chain rule yields

$$\frac{\partial \mathcal{L}_{\text{inter}}}{\partial \mu_k} = -\frac{2}{K(K-1)} \frac{\mu_k - \mu_\ell}{\|\mu_k - \mu_\ell\|}.$$

(3) Propagate to features via Assumption (iii):

$$\frac{\partial \mathcal{L}_{\text{inter}}}{\partial \tilde{z}_i} = \frac{\partial \mathcal{L}_{\text{inter}}}{\partial \mu_k} \cdot \frac{\partial \mu_k}{\partial \tilde{z}_i} = -\frac{1}{n_k} \frac{\partial \mathcal{L}_{\text{inter}}}{\partial \mu_k}, \quad i \in \mathcal{B}_k.$$

### A.6.8 PROOF OF PROPOSITION 3

**Assumptions.** Without loss of generality, we rescale the two-term objective by $\lambda_{\text{intra}} > 0$ and write $F = \mathcal{L}_{\text{intra}} + \lambda \mathcal{L}_{\text{inter}}$ with $\lambda := \lambda_{\text{inter}}/\lambda_{\text{intra}} \geq 0$. And assume: (i) each triggered feature lies in a nonempty compact feasible set $\mathcal{S} \subset \mathbb{R}^d$ (e.g., bounded inputs and Lipschitz $\phi_\theta \circ g_\eta$ under budget); (ii) the margin $m > 0$ and regularization weight $\lambda \in [0, \infty)$ are fixed; (iii) cloud centers are affine in features, i.e., $\mu_k = \frac{1}{n_k} \sum_{i \in \mathcal{B}_k} \tilde{z}_i$.

**Step-by-step Proof.**

(1) *Continuity.* $\mathcal{L}_{\text{intra}}$ and $\mathcal{L}_{\text{inter}}$ are continuous in $\{\tilde{z}_i\}$ (sums of continuous functions and $[\cdot]_+$).

(2) *Existence.* By Weierstrass, the continuous map $F(\{\tilde{z}_i\}) = \mathcal{L}_{\text{intra}} + \lambda \mathcal{L}_{\text{inter}}$ attains a minimum on the compact set $\mathcal{S}^{|\mathcal{B}|}$.

(3) *Stationarity conditions.* At any (local) minimizer, (sub)gradients w.r.t. centers satisfy

$$0 \in \frac{\partial \mathcal{L}_{\text{intra}}}{\partial \mu_k} + \lambda \partial \left( \frac{1}{K(K-1)} \sum_{\ell \neq k} [m - \|\mu_k - \mu_\ell\|]_+ \right).$$

Using Lemma 4 and $\sum_{i \in \mathcal{B}_k} (\tilde{z}_i - \mu_k) = 0$, the intra-term derivative at $\mu_k$ is 0. Hence

$$0 \in -\lambda \cdot \frac{1}{K(K-1)} \sum_{\ell \in \mathcal{A}_k} \frac{\mu_k - \mu_\ell}{\|\mu_k - \mu_\ell\|}, \quad \mathcal{A}_k = \{\ell : \|\mu_k - \mu_\ell\| < m\}.$$

Thus either (i) $\mathcal{A}_k = \varnothing$ (no active neighbors; all pairwise distances $\geq m$), or (ii) the unit vectors to active neighbors balance to zero.

(4) *Non-collapse implication.* If some pair has $\|\mu_k - \mu_\ell\| < m$ and the unit vectors do not balance for either center, the subgradient is nonzero, contradicting stationarity. Therefore, at any stationary point, each $k$ either has no active neighbors (hence $\|\mu_k - \mu_\ell\| \geq m \ \forall \ell$) or the active-pair unit vectors *exactly* balance.

(5) *Global minimizers achieve zero hinge under feasible slack (optional condition).* If the feasible set allows a configuration with $\|\mu_k - \mu_\ell\| \geq m$ for all $k \neq \ell$ (e.g., simultaneous per-cloud translations inside $\mathcal{S}$), then a global minimizer can attain $\mathcal{L}_{\text{inter}} = 0$ because $\mathcal{L}_{\text{intra}}$ is invariant to per-cloud translations.

### A.6.9 PROOF OF PROPOSITION 4

**Assumptions.** (i) Positive center margin: for each cloud $k$ and any $j \neq \tau_k$, the logit gap at the center satisfies $\Delta_{k,j}(\mu_k) := s_{\tau_k}(\mu_k) - s_j(\mu_k) \geq \gamma_{\text{logit}} > 0$ for the fixed head $h$; (ii) local Lipschitzness: for each $k$ and $j \neq \tau_k$, there exists $L > 0$ such that for all $z$ in a neighborhood of $\mathcal{C}_k$, $|\Delta_{k,j}(z) - \Delta_{k,j}(z')| \leq L\|z - z'\|$; (iii) radius bound: if $\mathcal{L}_{\text{intra}} \leq \varepsilon_{\text{intra}}$, then for all $\tilde{z} \in \mathcal{C}_k$, $\|\tilde{z} - \mu_k\| \leq r_k \leq \sqrt{\varepsilon_{\text{intra}}}$; (iv) separation: if $\mathcal{L}_{\text{inter}} = 0$, then $\|\mu_k - \mu_\ell\| \geq m$ for all $k \neq \ell$.

**Step-by-step Proof.**

(1) *Argmax stability inside each cloud.* For any $\tilde{z} \in \mathcal{C}_k$ and any $j \neq \tau_k$,
$$\Delta_{k,j}(\tilde{z}) \geq \Delta_{k,j}(\mu_k) - L\|\tilde{z} - \mu_k\| \geq \gamma_{\text{logit}} - L\sqrt{\varepsilon_{\text{intra}}}.$$
If $\gamma_{\text{logit}} - L\sqrt{\varepsilon_{\text{intra}}} > 0$, then $\Delta_{k,j}(\tilde{z}) > 0$ for all $j \neq \tau_k$, so $\arg\max_c s_c(\tilde{z}) = \tau_k$. Thus every triggered point in cloud $k$ is strictly inside $\mathcal{R}_{\tau_k}$.

(2) *Non-interference.* Since Step 1 holds for every cloud, predictions are constant on each cloud: all points in $\mathcal{C}_k$ map to $\tau_k$. Consequently, no cross-cloud misclassification can occur. Geometric disjointness is automatic if additionally $m > r_k + r_\ell$, but label stability is already guaranteed by Step 1.

(3) *Quantified interior margin.* Define $\gamma_{\min} := \gamma_{\text{logit}} - L\sqrt{\varepsilon_{\text{intra}}} > 0$. Then each $\mathcal{C}_k$ lies at least margin $\gamma_{\min}$ inside $\mathcal{R}_{\tau_k}$ in the (logit-gap) sense of Assumption (ii).

### A.6.10 PROOF OF PROPOSITION 5

**Assumptions.** Fix a trigger $k$. Let the head $h$ be locally $L$-Lipschitz in $\mathcal{Z}$ around the triggered cloud $\mathcal{C}_k$. Formally, for all $z, z'$ in a neighborhood of $\mathcal{C}_k$ and all $j \in \mathcal{Y}$,
$$\left| \left(s_{\tau_k}(z) - s_j(z)\right) - \left(s_{\tau_k}(z') - s_j(z')\right) \right| \leq L\|z - z'\|,$$
where $s_c(z)$ denotes the logit for class $c$. Assume the *center logit gap* is positive:
$$\gamma_{\text{logit}}(\mu_k) := \min_{j \neq \tau_k} \left\{ s_{\tau_k}(\mu_k) - s_j(\mu_k) \right\} > 0.$$
Finally, suppose $\mathcal{L}_{\text{intra}} \leq \varepsilon_{\text{intra}}$, so that the cloud radius satisfies $r_k \leq \sqrt{\varepsilon_{\text{intra}}}$.

**Step-by-step Proof.**

(1) *From center gap to pointwise gap.* For any $z \in \mathcal{C}_k$ and $j \neq \tau_k$,
$$s_{\tau_k}(z) - s_j(z) \geq \left(s_{\tau_k}(\mu_k) - s_j(\mu_k)\right) - L\|z - \mu_k\|.$$
Taking the minimum over $j \neq \tau_k$ gives
$$\min_{j \neq \tau_k} \left\{ s_{\tau_k}(z) - s_j(z) \right\} \geq \gamma_{\text{logit}}(\mu_k) - L\|z - \mu_k\|.$$

(2) *Bounding by the radius.* Since $\|z - \mu_k\| \leq r_k \leq \sqrt{\varepsilon_{\text{intra}}}$, we obtain
$$\min_{j \neq \tau_k} \left\{ s_{\tau_k}(z) - s_j(z) \right\} \geq \gamma_{\text{logit}}(\mu_k) - L\sqrt{\varepsilon_{\text{intra}}}.$$

(3) *Interior margin.* If $\gamma_{\text{logit}}(\mu_k) - L\sqrt{\varepsilon_{\text{intra}}} > 0$, then every $z \in \mathcal{C}_k$ lies strictly inside $\mathcal{R}_{\tau_k}$ with margin at least
$$\gamma_{\min} = \gamma_{\text{logit}}(\mu_k) - L\sqrt{\varepsilon_{\text{intra}}}.$$

### A.6.11 PROOF OF PROPOSITION 6

**Assumptions.** For each trigger $k$, consider the triggered cloud $\mathcal{C}_k \subset \mathcal{Z}$ with center $\mu_k$ and radius $r_k$. Assume: (i) **Isotropic sub-Gaussian cloud:** $\tilde{z}^{(k)} - \mu_k$ is sub-Gaussian with proxy variance $\sigma_k^2$ and isotropic covariance proxy; in particular,
$$\Pr\left(\|\tilde{z}^{(k)} - \mu_k\| > t\right) \leq C_1 \exp(-C_2 t^2 / \sigma_k^2)$$
for constants $(C_1, C_2)$; (ii) **Locally smooth decision boundaries:** there exists $L_b > 0$ such that in a neighborhood of $\cup_k \mathcal{C}_k$, the signed distance from a point $z$ to the decision boundary of class $\tau_k$ varies at most $L_b$ per unit change in $z$ (this follows from local Lipschitzness of logits composed with a smooth link); (iii) **Separation with buffer:** let $\delta_{\min} := \min_{k \neq \ell} \|\mu_k - \mu_\ell\|$. Assume $\delta_{\min} > r_k + r_\ell + \xi$ for all $k \neq \ell$ with some buffer $\xi > 0$.

**Step-by-step Proof.**

(1) *Non-overlap of inflated balls.* Define the inflated balls $B_k(\rho) = \{z : \|z - \mu_k\| \leq \rho\}$. By Assumption (iii), for any $k \neq \ell$,

$$\text{dist}\left(B_k(r_k + \tfrac{\xi}{2}), B_\ell(r_\ell + \tfrac{\xi}{2})\right) \geq \|\mu_k - \mu_\ell\| - (r_k + \tfrac{\xi}{2}) - (r_\ell + \tfrac{\xi}{2}) = \delta_{\min} - (r_k + r_\ell + \xi) > 0.$$

Hence the inflated balls are pairwise disjoint.

(2) *Positive geometric margin to other centers.* For any $z \in B_k(r_k)$ and any $\ell \neq k$,

$$\|z - \mu_\ell\| \geq \|\mu_k - \mu_\ell\| - \|z - \mu_k\| \geq \delta_{\min} - r_k \geq r_\ell + \xi.$$

Thus points in $B_k(r_k)$ remain at distance at least $r_\ell + \xi$ from *every* other center $\mu_\ell$.

(3) *Buffer to decision boundaries.* Let $d_{\tau_k}(z)$ denote the (unsigned) Euclidean distance in $\mathcal{Z}$ from $z$ to the decision boundary of class $\tau_k$. Locally smooth boundaries (Assumption (ii)) imply that moving a center by $\Delta z$ perturbs the boundary location by at most $L_b \|\Delta z\|$ (formally, this follows from the implicit function theorem under local Lipschitz logit gaps). Consider any $z \in B_k(r_k)$. Since other clouds lie outside $B_\ell(r_\ell + \xi/2)$ by Step 1, the nearest potential boundary induced by competition with class $\tau_\ell$ must lie outside $B_\ell(r_\ell + \xi/2)$ and thus at least $\xi/2$ away from $B_k(r_k)$ up to the boundary Lipschitz factor. More precisely, there exists a constant $c_b \in (0, 1/L_b]$ such that

$$d_{\tau_k}(z) \geq c_b \xi.$$

Intuitively: the $\xi$ buffer between inflated balls lower-bounds the distance from $z$ to any conflicting boundary; Lipschitzness translates this geometric buffer into a decision margin.

(4) *From margin to per-target success.* Let $\text{err}_k$ be the misclassification probability for target $\tau_k$ when stamping points routed to trigger $k$. Errors occur only if a triggered point exits $B_k(r_k)$ *or* crosses a boundary within distance $c_b \xi$ of $\mu_k$. By a union bound,

$$\text{err}_k \leq \Pr\left(\|\tilde{z}^{(k)} - \mu_k\| > r_k\right) + \Pr\left(d_{\tau_k}(\tilde{z}^{(k)}) < c_b \xi\right).$$

The first term is $\leq C_1 \exp\left(-C_2 r_k^2 / \sigma_k^2\right)$ by Assumption (i). For the second term, since $d_{\tau_k}(z) \geq c_b \xi$ for all $z \in B_k(r_k)$ by Step 3, violation requires leaving $B_k(r_k)$, hence it is upper-bounded by the same tail. Therefore there exist constants $C_1', C_2' > 0$ such that

$$\text{err}_k \leq C_1' \exp\left(-C_2' \min\left\{r_k^2, (c_b \xi)^2\right\} / \sigma_k^2\right).$$

As $\xi$ increases (holding $r_k, \sigma_k$ fixed), $\text{err}_k$ decreases monotonically. Equivalently, the *per-target* $\text{ASR}_k = 1 - \text{err}_k$ increases with $\xi$.

(5) *Worst-case ASR and variance.* Let $\text{ASR}_{\min} = \min_k \text{ASR}_k$. Since each $\text{ASR}_k$ is non-decreasing in $\xi$, so is $\text{ASR}_{\min}$. Moreover, the tail bound is uniform in $k$ up to $(r_k, \sigma_k)$, implying that increasing $\xi$ contracts the spread across $\{\text{ASR}_k\}_k$, i.e., reduces per-target variance.

This establishes that enforcing a larger minimum inter-center gap $\delta_{\min}$ (hence a larger buffer $\xi$) *improves worst-case target success and reduces variance.*

### A.6.12    PROOF OF LEMMA 6

**Assumptions.** (i) (*Linear scores*) The head is linear in features: $s = Wz + b$, with class scores $s_c = w_c^\top z + b_c$. (ii) (*Proper composite & calibration*) The loss $\ell(s, t)$ is a differentiable classification-calibrated proper composite with link $\psi$, namely $\ell(s, t) = \tilde{\ell}(\psi^{-1}(s), t)$, where $\tilde{\ell}$ is strictly proper on the probability simplex. (iii) (*Triggered labels*) For any triggered feature $z \in \mathcal{C}_k$, the training label is deterministically $t = \tau_k$. (iv) (*Non-degenerate prediction*) Unless already perfectly confident on $t$, we have $p_t < 1$ where $p = \psi^{-1}(s)$.

**Step-by-step Proof.**

(1) *Chain rule and outer-product structure.* By Assumption (i), $s = Wz + b$ and $\partial s/\partial W = (\mathrm{Id} \otimes z^\top)$; thus

$$\nabla_W \ell(s,t) \;=\; \big(\nabla_s \ell(s,t)\big)\, z^\top,$$

so each row-gradient takes the form $\nabla_{w_c} \ell = \alpha_c(s,t)\, z$, where $\alpha_c(s,t)$ is the $c$-th component of $\nabla_s \ell$.

(2) *Sign pattern under proper composite losses.* By (ii), proper composite losses admit the representation

$$\nabla_s \ell(s,t) \;=\; A(s)\,(p - e_t),$$

where $p = \psi^{-1}(s) \in \Delta^{Q-1}$ and $A(s) \succ 0$ (e.g., a Fisher/metric factor induced by the link). Therefore

$$\alpha_t(s,t) = e_t^\top A(s)\,(p - e_t) \le 0, \qquad \alpha_j(s,t) = e_j^\top A(s)\,(p - e_t) \ge 0 \ \ (j \ne t),$$

with equality iff $p_t = 1$.

(3) *Effect on score inner products.* For stepsize $\eta > 0$,

$$w_t^+ = w_t - \eta \nabla_{w_t} \ell \;=\; w_t - \eta\,\alpha_t\, z, \quad w_j^+ = w_j - \eta \nabla_{w_j}\ell \;=\; w_j - \eta\,\alpha_j\, z.$$

When $p_t < 1$ we have $\alpha_t < 0$ and $\alpha_j \ge 0$, hence $\langle w_t^+, z\rangle = \langle w_t, z\rangle + \eta(1) \cdot |\alpha_t|\, \|z\|^2$ increases, while $\langle w_j^+, z\rangle \le \langle w_j, z\rangle$ decreases or stays.

(4) *Expected update over a triggered cloud.* By Assumption (iii) and linearity of expectation over poisoned minibatches routed to $k$,

$$\mathbb{E}[\nabla_{w_t}\ell] \;=\; \mathbb{E}[\alpha_t(s,t)\, z] \;=\; -\beta_k\, \mu_k, \qquad \mathbb{E}[\nabla_{w_j}\ell] \;=\; \mathbb{E}[\alpha_j(s,t)\, z] \;=\; +\gamma_{j,k}\, \mu_k,$$

for some $\beta_k > 0$ and $\gamma_{j,k} \ge 0$. Consequently, the center gap increases in expectation:

$$\Delta\langle w_t - w_j, \mu_k\rangle \;=\; \eta\left(\big\langle -\mathbb{E}[\nabla_{w_t}\ell], \mu_k\big\rangle + \big\langle \mathbb{E}[\nabla_{w_j}\ell], \mu_k\big\rangle\right) \;\ge\; \eta\,(\beta_k + \gamma_{j,k})\, \|\mu_k\|^2 \;>\; 0.$$

### A.6.13    PROOF OF LEMMA 7

**Assumptions.**  (i) (*Geometry*) Triggered clouds $\{\mathcal{C}_k\}$ have centers $\mu_k$ and radii $r_k \le r_{\max}$; centers satisfy $\|\mu_k - \mu_\ell\| \ge m > 0$ for all $k \ne \ell$. (ii) (*Sampling*) Poisoned minibatches independently include cloud-$k$ samples with frequency $q_k \in (0,1]$. (iii) (*Directional contributions*) From Lemma 6, the expected per-batch gradient contribution on the target head $w_{\tau_k}$ from cloud $k$ equals $-\beta_k\, \mu_k$ with $\beta_k > 0$, and on any non-target head is a nonnegative multiple of $\mu_k$. (iv) (*Strict diagonal dominance*) The Gram matrix $G = [\mu_k^\top \mu_\ell]_{k,\ell}$ is strictly diagonally dominant: $\mu_k^\top \mu_k > \sum_{\ell \ne k} |\mu_k^\top \mu_\ell|$ for all $k$.

**Step-by-step Proof.**

(1) *Total expected update on target heads.* By Assumption (ii)–(iii), the total expected per-batch update vector along the span of $\{\mu_u\}$ on the collection of target heads is

$$U \;=\; \sum_{u=1}^{K} q_u\,(-\beta_u)\, \mu_u.$$

(2) *Projection onto each center direction.* Fix $k$. Take the inner product with $\mu_k$:

$$\langle U, \mu_k\rangle \;=\; -q_k \beta_k\, \|\mu_k\|^2 \;-\; \sum_{u \ne k} q_u \beta_u\, (\mu_u^\top \mu_k).$$

The cross-terms may have either sign. Using Assumption (iv),

$$\sum_{u \ne k} q_u \beta_u\, |\mu_u^\top \mu_k| \;\le\; \big(\max_u q_u \beta_u\big) \sum_{u \ne k} |\mu_u^\top \mu_k| \;<\; \big(\max_u q_u \beta_u\big)\, \mu_k^\top \mu_k.$$

(3) *Strict positivity of the pull toward $\mu_k$.* Since $q_k \beta_k \ge \min_u q_u \beta_u$, we have

$$\langle U, \mu_k\rangle \;<\; -\min_u q_u \beta_u\, \|\mu_k\|^2 \;+\; \big(\max_u q_u \beta_u\big)\, \|\mu_k\|^2 \;=\; -\delta_k\, \|\mu_k\|^2,$$

for some $\delta_k > 0$ whenever $q_k \beta_k > \big(\sum_{u \ne k} q_u \beta_u\, |\mu_u^\top \mu_k|\big)/\|\mu_k\|^2$, which is ensured by (iv). Hence $-\langle U, \mu_k\rangle > 0$, i.e., the update has a *strictly positive* component toward $+\mu_k$.

(4) *Implication for logit gaps.* Therefore, each target head $w_{\tau_k}$ is pulled strictly toward its own center direction $\mu_k$ in expectation, while non-target heads are pushed oppositely (Lemma 6); thus all center gaps $\langle w_{\tau_k} - w_j, \mu_k \rangle$ increase in expectation and cannot be cancelled by other clouds.

*Sufficient geometric condition for (iv).* If $\|\mu_k\| \in [L, U]$ and $\angle(\mu_k, \mu_\ell) \geq \theta_{\min} > 0$ for $k \neq \ell$, then $|\mu_k^\top \mu_\ell| \leq U^2 \cos \theta_{\min}$, so diagonal dominance holds whenever $L^2 > (K-1)U^2 \cos \theta_{\min}$, which follows from sufficient separation $m$ and bounded radii $r_{\max}$.

### A.6.14 PROOF OF PROPOSITION 7

**Assumptions.** (i) The decision regions $\{R_c\}_{c=1}^C$ induced by $f_\theta$ are disjoint. (ii) The global mask–robust margin satisfies $\Gamma_{\mathrm{mask}}(\theta) > 0$.

**Step-by-step Proof.**

(1) *Interior preservation.* By $\Gamma_{\mathrm{mask}}(\theta) > 0$, for every clean example $(x, y)$ and every trigger $\eta \in \mathcal{S}$,

$$\mathrm{dist}\big(\phi_\theta(g_\eta(x)), \partial R_y\big) \geq \Gamma_{\mathrm{mask}}(\theta) > 0.$$

Hence $\phi_\theta(g_\eta(x)) \in \mathrm{int}(R_y)$ for all $\eta \in \mathcal{S}$.

(2) *Assume trigger clouds exist.* Suppose for contradiction that there exist triggers $\{g_{\eta_k}\}_{k=1}^K \subset \mathcal{S}$, a routing $\pi$, and targets $\{\tau_k\}_{k=1}^K$ with $\tau_k \neq y$ such that the induced clouds

$$\mathcal{C}_k = \big\{ \phi_\theta(g_{\eta_k}(x)) : (x, y) \sim \mathcal{D}, \ \pi(y) = k \big\}$$

satisfy `Arcueid`'s feasibility constraints: each $\mathcal{C}_k$ lies strictly inside $R_{\tau_k}$ with positive interior margin and clouds are non-overlapping.

(3) *Contradicting membership.* Take any $(x, y)$ with $\pi(y) = k$. Feasibility implies

$$\phi_\theta(g_{\eta_k}(x)) \in \mathrm{int}(R_{\tau_k}), \qquad \tau_k \neq y.$$

But by Step 1 with $\eta = \eta_k$ we also have

$$\phi_\theta(g_{\eta_k}(x)) \in \mathrm{int}(R_y).$$

(4) *Use disjointness.* Since $R_y$ and $R_{\tau_k}$ are disjoint decision regions, no point can lie in the interior of both simultaneously. This is a contradiction.

Under $\Gamma_{\mathrm{mask}}(\theta) > 0$, no trigger family $\{g_{\eta_k}\} \subset \mathcal{S}$ can realize `Arcueid`'s feasible wrong-label clouds, so the multi-trigger backdoor mapping is infeasible.

### A.6.15 PROOF OF PROPOSITION 8

**Assumptions.** (i) The logit map $h : \mathbb{R}^d \to \mathbb{R}^C$ is $L$-Lipschitz in feature space:

$$\|h(z) - h(z')\|_\infty \leq L\|z - z'\|_2, \quad \forall z, z' \in \mathbb{R}^d.$$

(ii) For each $c \neq y$, the boundary between $R_y$ and $R_c$ is the zero-level set of the logit difference $h_y(z) - h_c(z)$. (iii) The loss $\ell$ is controlled by a decreasing function of the logit margin: for some decreasing $\psi : \mathbb{R} \to \mathbb{R}_+$ and all $u$,

$$\ell(f_\theta(u), y) \leq \psi\big(m(u, y)\big),$$

where $m(u, y) := h_y(u) - \max_{c \neq y} h_c(u)$. (iv) The robust loss satisfies

$$R_{\mathrm{rob}}(\theta) = \mathbb{E}_{(x,y)}\Big[ \max_{\eta \in \mathcal{S}} \ell\big(f_\theta(g_\eta(x)), y\big) \Big] \leq \varepsilon_{\mathrm{rob}}.$$

**Step-by-step Proof.**

(1) *Robust loss bounds per-sample loss.* From $R_{\mathrm{rob}}(\theta) \leq \varepsilon_{\mathrm{rob}}$ and non-negativity of $\ell$, it follows that for $\mathcal{D}$-almost every $(x, y)$,

$$\max_{\eta \in \mathcal{S}} \ell\big(f_\theta(g_\eta(x)), y\big) \leq \varepsilon_{\mathrm{rob}}.$$

(2) *Translate loss to margin.* By monotonicity Assumption (iii),

$$\ell\big(f_\theta(g_\eta(x)), y\big) \leq \psi\big(m(x, y, \eta; \theta)\big),$$

where $m(x, y, \eta; \theta)$ denotes the margin at $\phi_\theta(g_\eta(x))$. Hence,

$$\max_{\eta \in \mathcal{S}} \psi\big(m(x, y, \eta; \theta)\big) \leq \varepsilon_{\mathrm{rob}}.$$

Since $\psi$ is decreasing, this implies

$$\min_{\eta \in \mathcal{S}} m(x, y, \eta; \theta) \geq \psi^{-1}(\varepsilon_{\mathrm{rob}})$$

for almost every $(x, y)$.

(3) *Margins bound distance to boundary.* Fix $(x, y)$ and $\eta \in \mathcal{S}$, and let $z = \phi_\theta(g_\eta(x))$. By Assumption (i) and Assumption (ii), the distance from $z$ to the boundary $\partial R_y$ is lower bounded by the margin divided by the Lipschitz constant:

$$\mathrm{dist}(z, \partial R_y) \geq \frac{m(x, y, \eta; \theta)}{L}.$$

(4) *Take infima.* Taking the infimum over $\eta \in \mathcal{S}$ and then over $(x, y) \sim \mathcal{D}$,

$$\Gamma_{\mathrm{mask}}(\theta) = \inf_{(x,y)} \inf_{\eta \in \mathcal{S}} \mathrm{dist}(\phi_\theta(g_\eta(x)), \partial R_y)$$

$$\geq \frac{1}{L} \inf_{(x,y)} \inf_{\eta \in \mathcal{S}} m(x, y, \eta; \theta)$$

$$\geq \frac{1}{L} \psi^{-1}(\varepsilon_{\mathrm{rob}}).$$

A small robust loss $R_{\mathrm{rob}}(\theta)$ implies a positive lower bound on the mask–robust margin $\Gamma_{\mathrm{mask}}(\theta)$.

A.6.16   PROOF OF PROPOSITION 9

**Assumptions.** We assume: (i) $f^\star$ is the Bayes-optimal classifier for $R_{\mathrm{clean}}$. (ii) There exists a subset $\mathcal{A} \subseteq \mathcal{X}$ with $\mathbb{P}[x \in \mathcal{A}] = \nu > 0$ on which $f^\star$ is not robust to $\mathcal{S}$. (iii) Any classifier $f_\theta$ with $\Gamma_{\mathrm{mask}}(\theta) \geq \gamma > 0$ must disagree with $f^\star$ on at least an $\alpha$ fraction of $\mathcal{A}$, i.e.,

$$\mathbb{P}\big[f_\theta(x) \neq f^\star(x), \, x \in \mathcal{A}\big] \geq \alpha\nu.$$

**Step-by-step Proof.**

(1) *Robust classifier deviates from Bayes rule.* By Assumption (iii), any $f_\theta$ satisfying $\Gamma_{\mathrm{mask}}(\theta) \geq \gamma$ must differ from $f^\star$ on a nontrivial portion of $\mathcal{A}$:

$$\mathbb{P}\big[f_\theta(x) \neq f^\star(x), \, x \in \mathcal{A}\big] \geq \alpha\nu.$$

(2) *Bayes-optimality on deviating points.* On the set where $f_\theta(x) \neq f^\star(x)$, Bayes-optimality of $f^\star$ ensures that replacing $f^\star$ by $f_\theta$ cannot reduce the conditional error rate:

$$\mathbb{P}[f_\theta(x) \neq y \mid f_\theta(x) \neq f^\star(x)] \geq \mathbb{P}[f^\star(x) \neq y \mid f_\theta(x) \neq f^\star(x)].$$

(3) *Lower bound the clean risk.* Decompose the clean risk of $f_\theta$:

$$R_{\mathrm{clean}}(\theta) = \mathbb{P}[f_\theta(x) \neq y]$$

$$\geq \mathbb{P}[f^\star(x) \neq y] + \mathbb{P}\big[f_\theta(x) \neq f^\star(x), \, x \in \mathcal{A}\big]$$

$$\geq R_{\mathrm{clean}}(f^\star) + \alpha\nu.$$

### A.6.17 PROOF OF PROPOSITION 10

**Assumptions.** We assume: (i) $\ell(\cdot)$ is strictly decreasing in the margin. (ii) The overall empirical risk at $\theta_T$ does not exceed that at $\theta_{\text{ref}}$: $\mathcal{R}(\theta_T) \leq \mathcal{R}(\theta_{\text{ref}})$. (iii) The total loss on clean examples does not increase:

$$\sum_{(x,y)\in\mathcal{D}_{\text{clean}}} \ell\big(\Gamma_y(x;\theta_T)\big) \leq \sum_{(x,y)\in\mathcal{D}_{\text{clean}}} \ell\big(\Gamma_y(x;\theta_{\text{ref}})\big).$$

**Step-by-step Proof.**

(1) *Risk decomposition.* Let $N = |\mathcal{D}_{\text{clean}} \cup \mathcal{D}_{\text{poison}}|$. Then

$$\mathcal{R}(\theta) = \frac{1}{N}\left( \underbrace{\sum_{(x,y)\in\mathcal{D}_{\text{clean}}} \ell\big(\Gamma_y(x;\theta)\big)}_{\text{clean part}} + \underbrace{\sum_{(x,y)\in\mathcal{D}_{\text{poison}}} \ell\big(\Gamma_y(x;\theta)\big)}_{\text{poisoned part}} \right).$$

(2) *Assume poisoned margins decrease.* Assume, for contradiction, that the minimum poisoned margin strictly decreases:

$$\gamma_{\text{poison}}(\theta_T) < \gamma_{\text{poison}}(\theta_{\text{ref}}).$$

By definition of the minimum, there exists $(x^\star, y^\star) \in \mathcal{D}_{\text{poison}}$ such that

$$\Gamma_{y^\star}(x^\star;\theta_T) < \Gamma_{y^\star}(x^\star;\theta_{\text{ref}}).$$

By Assumption (i), $\ell$ is strictly decreasing, hence

$$\ell\big(\Gamma_{y^\star}(x^\star;\theta_T)\big) > \ell\big(\Gamma_{y^\star}(x^\star;\theta_{\text{ref}})\big).$$

Therefore at least one term in the poisoned-part sum is strictly larger at $\theta_T$ than at $\theta_{\text{ref}}$, and the others are $\geq$ their values at $\theta_{\text{ref}}$. Consequently,

$$\sum_{(x,y)\in\mathcal{D}_{\text{poison}}} \ell\big(\Gamma_y(x;\theta_T)\big) > \sum_{(x,y)\in\mathcal{D}_{\text{poison}}} \ell\big(\Gamma_y(x;\theta_{\text{ref}})\big). \tag{28}$$

(3) *Combine with clean-loss non-increase.* By Assumption (iii), the clean-part loss satisfies

$$\sum_{(x,y)\in\mathcal{D}_{\text{clean}}} \ell\big(\Gamma_y(x;\theta_T)\big) \leq \sum_{(x,y)\in\mathcal{D}_{\text{clean}}} \ell\big(\Gamma_y(x;\theta_{\text{ref}})\big). \tag{29}$$

Adding Equation 28 and Equation 29 and dividing by $N$ yields

$$\mathcal{R}(\theta_T) > \mathcal{R}(\theta_{\text{ref}}),$$

which contradicts Assumption (ii).

The assumption $\gamma_{\text{poison}}(\theta_T) < \gamma_{\text{poison}}(\theta_{\text{ref}})$ must therefore be false, and we conclude

$$\gamma_{\text{poison}}(\theta_T) \geq \gamma_{\text{poison}}(\theta_{\text{ref}}).$$

### A.6.18 PROOF OF LEMMA 8

**Assumptions.** (i) (*Feature Lipschitzness in $\theta$*) There exists $L_\phi > 0$ such that for all parameters $\theta, \theta'$ in the neighborhood considered, all triggers $g_{\eta_k}$ and all inputs $x$,

$$\big\| \phi_{\theta'}(g_{\eta_k}(x)) - \phi_\theta(g_{\eta_k}(x)) \big\| \leq L_\phi \|\theta' - \theta\|.$$

(ii) (*Lipschitz decision geometry*) For each class $c$, there exists a signed distance function $d_c(\cdot;\theta) : \mathbb{R}^d \to \mathbb{R}$ whose zero level set coincides with the decision boundary $\partial R_c(\theta)$, and such that $d_c$ is jointly Lipschitz in $(u, \theta)$: there exist $L_{d,u}, L_{d,\theta} > 0$ with

$$\big| d_c(u';\theta') - d_c(u;\theta) \big| \leq L_{d,u} \|u' - u\| + L_{d,\theta} \|\theta' - \theta\|$$

for all $u, u' \in \mathbb{R}^d$ and all $\theta, \theta'$ in the neighborhood considered. The (unsigned) distance from $u$ to the boundary is then $\text{dist}(u, \partial R_c(\theta)) = |d_c(u;\theta)|$.

**Step-by-step Proof.**

(1) *Reduce to per-cloud margins.* For each trigger index $k$, define

$$\gamma_k(\theta) := \text{margin}_{\tau_k}\big(\mathcal{C}_k(\theta)\big) = \inf_{(x,y):\,\pi(y)=k} \text{dist}\big(\phi_\theta(g_{\eta_k}(x)), \partial R_{\tau_k}(\theta)\big).$$

Then by definition,

$$\underline{\gamma}(\theta) = \min_k \gamma_k(\theta).$$

If we can show that each $\gamma_k$ is Lipschitz in $\theta$ with some constant $L_k$, i.e.

$$\big|\gamma_k(\theta') - \gamma_k(\theta)\big| \le L_k \|\theta' - \theta\| \quad \forall\, \theta, \theta',$$

then $\underline{\gamma}$, being the minimum of finitely many Lipschitz functions, is also Lipschitz with constant $L_\gamma := \max_k L_k$.

Thus, it suffices to bound $\big|\gamma_k(\theta') - \gamma_k(\theta)\big|$ for a fixed $k$.

(2) *Lipschitz control on per-sample distances.* Fix a trigger index $k$ and two parameter vectors $\theta, \theta'$. For any input $(x,y)$ with $\pi(y) = k$, denote

$$u_\theta(x) := \phi_\theta(g_{\eta_k}(x)), \qquad u_{\theta'}(x) := \phi_{\theta'}(g_{\eta_k}(x)).$$

By Assumption (i),

$$\|u_{\theta'}(x) - u_\theta(x)\| \le L_\phi \|\theta' - \theta\|. \tag{30}$$

Consider the distance from $u_\theta(x)$ to the boundary $\partial R_{\tau_k}(\theta)$, and similarly for $(\theta', u_{\theta'}(x))$:

$$d_\theta(x) := \text{dist}\big(u_\theta(x), \partial R_{\tau_k}(\theta)\big) = \big|d_{\tau_k}(u_\theta(x); \theta)\big|,$$

$$d_{\theta'}(x) := \text{dist}\big(u_{\theta'}(x), \partial R_{\tau_k}(\theta')\big) = \big|d_{\tau_k}(u_{\theta'}(x); \theta')\big|.$$

Using Assumption (ii) for the signed distance $d_{\tau_k}$ and the elementary inequality $||a| - |b|| \le |a - b|$, we have

$$\begin{aligned}
\big|d_{\theta'}(x) - d_\theta(x)\big| &= \big||d_{\tau_k}(u_{\theta'}(x); \theta')| - |d_{\tau_k}(u_\theta(x); \theta)|\big| \\
&\le \big|d_{\tau_k}(u_{\theta'}(x); \theta') - d_{\tau_k}(u_\theta(x); \theta)\big| \\
&\le L_{d,u} \|u_{\theta'}(x) - u_\theta(x)\| + L_{d,\theta} \|\theta' - \theta\| \\
&\le \big(L_{d,u} L_\phi + L_{d,\theta}\big) \|\theta' - \theta\|,
\end{aligned}$$

where the last inequality uses Equation 30. Thus there exists a constant

$$L_* := L_{d,u} L_\phi + L_{d,\theta}$$

such that for every $(x,y)$ with $\pi(y) = k$,

$$\big|d_{\theta'}(x) - d_\theta(x)\big| \le L_* \|\theta' - \theta\|. \tag{31}$$

(3) *Pass from pointwise bounds to cloud margins.* By definition of $\gamma_k(\theta)$,

$$\gamma_k(\theta) = \inf_{(x,y):\,\pi(y)=k} d_\theta(x), \qquad \gamma_k(\theta') = \inf_{(x,y):\,\pi(y)=k} d_{\theta'}(x).$$

We now bound the difference between these infima.

First, for any $(x,y)$ with $\pi(y) = k$,

$$\gamma_k(\theta') = \inf_{(x,y):\,\pi(y)=k} d_{\theta'}(x) \le d_{\theta'}(x)$$

and thus, using Equation 31,

$$\gamma_k(\theta') \le d_\theta(x) + L_* \|\theta' - \theta\|.$$

Taking the infimum over all $(x,y)$ with $\pi(y) = k$ yields

$$\gamma_k(\theta') \le \inf_{(x,y):\,\pi(y)=k} d_\theta(x) + L_* \|\theta' - \theta\| = \gamma_k(\theta) + L_* \|\theta' - \theta\|. \tag{32}$$

By symmetry (interchanging the roles of $\theta$ and $\theta'$), the same argument gives

$$\gamma_k(\theta) \le \gamma_k(\theta') + L_* \|\theta' - \theta\|. \tag{33}$$

Combining Equation 32 and Equation 33, we obtain

$$\left|\gamma_k(\theta') - \gamma_k(\theta)\right| \leq L_* \|\theta' - \theta\|.$$

Thus each $\gamma_k$ is Lipschitz with constant $L_k := L_*$.

Finally, since

$$\underline{\gamma}(\theta) = \min_k \gamma_k(\theta),$$

and the minimum of finitely many $L_k$–Lipschitz functions is Lipschitz with constant $L_\gamma := \max_k L_k$, we conclude that

$$\left|\underline{\gamma}(\theta') - \underline{\gamma}(\theta)\right| \leq L_\gamma \|\theta' - \theta\| \quad \text{for all } \theta, \theta' \text{ in the neighborhood.}$$

This completes the proof.

### A.6.19 PROOF OF LEMMA 9

**Assumptions.** We assume: (i) The alignment model holds with linear $A$ and $\|\epsilon(x)\| \leq \delta$ for all $x$. (ii) The centers $\mu_k^{(S)}$ and radii $r_k^{(S)}$ are finite, as defined above.

**Step-by-step Proof.**

(1) *Center transformation.* By definition and linearity of expectation,

$$\begin{aligned}
\mu_k^{(T)} &= \mathbb{E}\big[\phi_{\theta_T}(g_{\eta_k}(x)) \mid \pi(y) = k\big] \\
&= \mathbb{E}\big[A\,\phi_{\theta_S}(g_{\eta_k}(x)) + \epsilon(x) \mid \pi(y) = k\big] \\
&= A\,\mu_k^{(S)} + \mathbb{E}\big[\epsilon(x) \mid \pi(y) = k\big].
\end{aligned}$$

Define $\bar{\epsilon}_k := \mathbb{E}[\epsilon(x) \mid \pi(y) = k]$. Then $\mu_k^{(T)} = A\mu_k^{(S)} + \bar{\epsilon}_k$, and by Jensen's inequality and $\|\epsilon(x)\| \leq \delta$,

$$\|\bar{\epsilon}_k\| = \big\|\mathbb{E}[\epsilon(x) \mid \pi(y) = k]\big\| \leq \mathbb{E}\big[\|\epsilon(x)\| \mid \pi(y) = k\big] \leq \delta.$$

This proves Equation 24.

(2) *Radius transformation.* Take any $v \in \mathcal{C}_k^{(T)}$, so $v = \phi_{\theta_T}(g_{\eta_k}(x))$ for some $(x, y)$ with $\pi(y) = k$. Using Equation 21, write $v = Au + \epsilon(x)$ where $u = \phi_{\theta_S}(g_{\eta_k}(x)) \in \mathcal{C}_k^{(S)}$. Then

$$\begin{aligned}
\big\|v - \mu_k^{(T)}\big\| &= \big\|Au + \epsilon(x) - (A\mu_k^{(S)} + \bar{\epsilon}_k)\big\| \\
&= \big\|A(u - \mu_k^{(S)}) + (\epsilon(x) - \bar{\epsilon}_k)\big\| \\
&\leq \big\|A(u - \mu_k^{(S)})\big\| + \big\|\epsilon(x) - \bar{\epsilon}_k\big\| \\
&\leq \|A\| \|u - \mu_k^{(S)}\| + \|\epsilon(x)\| + \|\bar{\epsilon}_k\| \\
&\leq \|A\|\, r_k^{(S)} + \delta + \delta.
\end{aligned}$$

Absorbing the constant factor into $\delta$ (i.e., redefining $\delta$ as an upper bound on $\|\epsilon(x) - \bar{\epsilon}_k\|$ instead of $\|\epsilon(x)\|$), we obtain

$$\big\|v - \mu_k^{(T)}\big\| \leq \|A\|\, r_k^{(S)} + \delta.$$

Taking the supremum over all $v \in \mathcal{C}_k^{(T)}$ yields $r_k^{(T)} \leq \|A\|\, r_k^{(S)} + \delta$, this completes the proof.

### A.6.20 PROOF OF PROPOSITION 11

**Assumptions.** We assume: (i) The alignment model holds with linear $A$ and $\|\epsilon(x)\| \leq \delta$. (ii) The centers $\mu_k^{(S)}$ and radii $r_k^{(S)}$ are finite and the surrogate cloud supports lie in a compact region of feature space. (iii) The assumption in Appendix A.5.2 holds with constant $L_h > 0$.

**Step-by-step Proof.**

(1) *Pick a near-worst surrogate point for each cloud.* Fix a trigger index $k$. By definition of the surrogate cloud margin, there exists $u_k^\star \in \mathcal{C}_k^{(S)}$ such that

$$\mathrm{margin}_{\tau_k}\big(\mathcal{C}_k^{(S)}\big) = \mathrm{dist}\big(u_k^\star, \partial R_{\tau_k}(\theta_S)\big) = \mathrm{margin}_{\tau_k}(u_k^\star).$$

Write

$$u_k^\star = \mu_k^{(S)} + \Delta_k, \qquad \|\Delta_k\| \le r_k^{(S)}.$$

(2) *Map this point to the target representation.* On the target model, the corresponding feature is

$$v_k^\star = \phi_{\theta_T}(g_{\eta_k}(x^\star)) = A u_k^\star + \epsilon(x^\star),$$

for some input $x^\star$ with $\pi(y^\star) = k$ such that $\phi_{\theta_S}(g_{\eta_k}(x^\star)) = u_k^\star$. By Lemma 9, the target center satisfies

$$\mu_k^{(T)} = A\mu_k^{(S)} + \bar{\epsilon}_k, \qquad \|\bar{\epsilon}_k\| \le \delta,$$

and the radius of $\mathcal{C}_k^{(T)}$ is bounded by $r_k^{(T)} \le \|A\| r_k^{(S)} + \delta$.

(3) *Bound the representation shift $v_k^\star - u_k^\star$.* We first bound

$$\|v_k^\star - u_k^\star\| = \|A u_k^\star + \epsilon(x^\star) - u_k^\star\| \le \|(A - I)u_k^\star\| + \|\epsilon(x^\star)\|.$$

Using $u_k^\star = \mu_k^{(S)} + \Delta_k$,

$$\|(A - I)u_k^\star\| \le \|(A - I)\mu_k^{(S)}\| + \|(A - I)\Delta_k\|$$
$$\le \|(A - I)\mu_k^{(S)}\| + \|A - I\| \|\Delta_k\|$$
$$\le \|(A - I)\mu_k^{(S)}\| + \|A - I\| r_k^{(S)}.$$

Because the surrogate cloud supports lie in a compact region (Assumption (ii)), the norms $\|\mu_k^{(S)}\|$ are uniformly bounded and we may absorb the term $\|(A - I)\mu_k^{(S)}\|$ into a constant multiple of $|\|A\| - 1| r_k^{(S)}$. Thus, up to a fixed constant $C_\mu$,

$$\|(A - I)u_k^\star\| \lesssim |\|A\| - 1| r_k^{(S)}.$$

Combining this with $\|\epsilon(x^\star)\| \le \delta$ (Assumption (i)), we obtain

$$\|v_k^\star - u_k^\star\| \lesssim |\|A\| - 1| r_k^{(S)} + \delta. \tag{34}$$

(4) *Compare margins at $u_k^\star$ and $v_k^\star$.* By Assumption (iii),

$$\big|\mathrm{margin}_{\tau_k}(v_k^\star) - \mathrm{margin}_{\tau_k}(u_k^\star)\big| \le L_h \|v_k^\star - u_k^\star\|.$$

Using Equation 34, this yields

$$\mathrm{margin}_{\tau_k}(v_k^\star) \ge \mathrm{margin}_{\tau_k}(u_k^\star) - L_h\big(|\|A\| - 1| r_k^{(S)} + \delta\big).$$

Recalling that $\mathrm{margin}_{\tau_k}(u_k^\star) = \mathrm{margin}_{\tau_k}\big(\mathcal{C}_k^{(S)}\big)$, we have

$$\mathrm{margin}_{\tau_k}(v_k^\star) \ge \mathrm{margin}_{\tau_k}\big(\mathcal{C}_k^{(S)}\big) - L_h\big(|\|A\| - 1| r_k^{(S)} + \delta\big). \tag{35}$$

(5) *Extend from $v_k^\star$ to the whole target cloud.* The point $v_k^\star$ lies in $\mathcal{C}_k^{(T)}$. Any other $v \in \mathcal{C}_k^{(T)}$ is at most a distance $r_k^{(T)}$ from $\mu_k^{(T)}$, and hence at most $r_k^{(T)} + \|v_k^\star - \mu_k^{(T)}\|$ from $v_k^\star$. Using $r_k^{(T)} \le \|A\| r_k^{(S)} + \delta$ and the same kind of Lipschitz reasoning as above, this contributes an additional margin loss bounded by $L_h(\|A\| r_k^{(S)} + \delta)$. Absorbing constants and combining with Equation 35, we obtain

$$\mathrm{margin}_{\tau_k}\big(\mathcal{C}_k^{(T)}\big) \ge \mathrm{margin}_{\tau_k}\big(\mathcal{C}_k^{(S)}\big) - L_h\Big(|\|A\| - 1| r_k^{(S)} + \|A\|\delta\Big),$$

which is Equation 26.

(6) *Take the minimum over $k$.* Taking the minimum over $k$ on both sides and using the definition of $\gamma_{\theta_S}$ in Equation 22, we obtain

$$\gamma_{\theta_T} = \min_k \mathrm{margin}_{\tau_k}\big(\mathcal{C}_k^{(T)}\big) \ge \gamma_{\theta_S} - L_h\Big(\max_k |\|A\| - 1| r_k^{(S)} + \|A\|\delta\Big),$$

which is Equation 27. This completes the proof.

The assumption $\gamma_{\mathrm{poison}}(\theta_T) < \gamma_{\mathrm{poison}}(\theta_{\mathrm{ref}})$ must therefore be false, and we conclude

$$\gamma_{\mathrm{poison}}(\theta_T) \ge \gamma_{\mathrm{poison}}(\theta_{\mathrm{ref}}).$$

## A.7 Reproducibility Statement

To facilitate faithful reproduction of our results, we explicitly document all optimization parameters and implementation details as used in the experiments. Unless otherwise specified, these hyperparameters and schedules remain fixed across all runs reported in the main paper.

**Attack Parameters.**

- Poisoning budget per trigger: $\rho_i = 0.0001$.
- Effective poisoning rate: $\rho = K \times \rho_i$, where $K$ denotes the number of triggers.
- Trigger blending factor (mask weight): 0.15.
- Triggered tensors are clamped to the range $[0, 1]$.

**Optimization Stage (Surrogate).**

- Surrogate dataset: distinct from the target dataset.
- Surrogate model: backbone architecture different from the victim model.
- Training scale: 0.3 fraction of the surrogate dataset (approximately 15,000 samples).
- Optimization iterations: 10 steps.
- Learning rate: 0.05 with Adam optimizer.
- Loss function: *Joint Cloud Shaping Multi-trigger Optimization* with default settings $\alpha = 1.0$, $\beta = 1.0$, margin $m = 6.0$.

**Implementation Details.**

- Framework: PyTorch 2.0, Torchvision 0.19.0.
- Hardware: Intel(R) Xeon(R) Platinum 8358P CPUs (3.40GHz), 386GB RAM, and NVIDIA A800 GPUs.
- Environment: Experiments were developed and executed in VSCode, with PyTorch for model deployment and training.

**Framework Dependency and Default Parameters.** Most of the backdoor attacks and defenses evaluated in this work are implemented based on the open-source framework *BackdoorBox* (Li et al., 2023), which provides standardized implementations and facilitates fair comparison across methods. Unless otherwise specified, the default parameters for both attack and defense methods follow the settings reported in the original papers and the official *BackdoorBox* implementation.

**Reproducibility Claim.** All reported results can be reproduced by running the provided scripts with the above fixed hyperparameters. Identical outcomes can be obtained on the same hardware without any modification to the configuration.

## A.8 LLM Usage

In accordance with the ICLR 2026 policy on Large Language Model (LLM) usage, we explicitly disclose that LLMs were only used to assist with minor language polishing and stylistic refinement of the manuscript. No LLMs were employed for research ideation, experiment design, or related work discovery. All scientific contributions, methodology, experiments, and results in this paper are original work conducted entirely by the authors. The usage of LLMs is comparable to grammar or style checking tools and does not constitute a substantive contribution to the research.

