# OpenReview forum: "Arcueid: Multi-trigger Cloud Shaping for Unified Backdoor Attack Paradigms"
_ICLR.cc/2026/Conference — Submitted to ICLR 2026_

### Official Review · Reviewer_cPwM · 2025-10-31

**Soundness:** 3
**Presentation:** 3
**Contribution:** 2
**Rating:** 4
**Confidence:** 3

**Summary:**

Arcueid is a black-box, multi-trigger backdoor framework that optimizes compact, well-separated “trigger clouds” in feature space (via Joint Cloud Shaping) so that multiple triggers can reliably map to arbitrary target labels across models and datasets with extremely low poisoning rates while remaining stealthy and robust to many defenses.

**Strengths:**

Arcueid introduces a unified and highly transferable multi-trigger backdoor framework that leverages Joint Cloud Shaping to optimize compact and well-separated trigger clusters in feature space, achieving strong attack success rates even under black-box and low-poisoning conditions. It generalizes across models and datasets, maintains minimal impact on clean accuracy, and remains robust against multiple state-of-the-art defenses, demonstrating both theoretical soundness and practical stealth.

**Weaknesses:**

1. The transferability theory essentially depends on the assumption that the surrogate and target models share a well-aligned representation space — specifically, that there exists a bounded linear mapping \(A\) and a small representation discrepancy \(\delta\) such that the triggered feature clouds in the target model preserve the same margin as those in the surrogate (see Proposition 2). However, this assumption is difficult to verify or guarantee in practice. The surrogate dataset in the paper is explicitly non-IID and small in scale (5k–15k samples), and the surrogate and victim architectures may differ substantially, leading to estimation errors in class centers and large shifts in feature geometry. As a result, the theoretical condition \(L_h\|A\|\delta < \gamma\) may not hold in real scenarios, causing the actual attack success rate (ASR) on the target model to degrade or become unstable.

2. The paper demonstrates Arcueid’s strong resistance to post-training defenses (Figure 7), but the results also reveal that NAD and FTSAM mitigate different aspects of the backdoor. NAD offers partial but unstable reduction of ASR through knowledge distillation, whereas FTSAM notably decreases ASR in single-target (M→1) cases but is ineffective for multi-target attacks. This suggests that applying NAD first to sanitize model representations and then using FTSAM for fine-tuning could substantially lower ASR.

3. The paper does not quantitatively evaluate how similar the surrogate data distribution is to the target one. It would be useful to report a feature-space similarity metric (e.g., CKA or FID) and identify the approximate sample size at which surrogate features become comparable to those of the target model.

**Questions:**

1. How sensitive is the theoretical guarantee to deviations in representation alignment between the surrogate and target models?

2. Can the authors empirically quantify the feature-space similarity between the surrogate and victim models (e.g., via CKA, FID, or linear probe accuracy) to validate the assumption used in Proposition 2?

3. At what surrogate dataset scale do the feature representations become sufficiently aligned for successful transfer, and how consistent is this threshold across different architectures?

4. Given that the surrogate dataset is non-IID and much smaller (5k–15k samples), how does the attack performance change as the domain shift increases or the sample size decreases?

5. Compared with highly OOD surrogate datasets (such as using SVHN for CIFAR-10), how can the degree of “OOD-ness” be quantitatively measured? Even if the data distributions do not overlap, can this distance be meaningfully quantified?

6. If the target model undergoes slight fine-tuning or domain adaptation after deployment, how does the proposed method perform—does the backdoor persist or degrade over time?

---

> ### Author Response · Authors · 2025-11-21
> **Rebuttal 1 to Reviewer cPwM**
>
> We would like to thank Reviewer cPwM for constructive reviews and comments. Our point-by-point responses are given below.
>
> ## **Q1: Theoretical Guarantee Sensitivity**
>
> **A1:**
> We thank the reviewer for this insightful question. Proposition 2 establishes transferability under a bounded linear alignment between the surrogate and victim representations:
>
> $$
> \phi_{\theta_T}(x) = A\phi_{\theta_S}(x) + \epsilon(x), \qquad \|\|\epsilon(x)\|\|\le \delta,
> $$
>
> and guarantees that the trigger clouds remain feasible on the target model whenever
>
> $$
> L_h\|\|A\|\| \delta < \gamma_{\theta_S},
> \tag{P2-condition}
> $$
>
> where $\gamma_{\theta_S}$ is the surrogate cloud margin.
>
> To examine the sensitivity of this guarantee, consider a trigger $k$ with surrogate center $\mu_k^{(S)}$ and radius $r_k^{(S)}$. Under the alignment assumption, the induced target center and radius satisfy
>
> $$
> \mu_k^{(T)} = A\mu_k^{(S)} + \bar\epsilon_k, \qquad
> r_k^{(T)} \le \|\|A\|\| r_k^{(S)} + \delta,
> $$
>
> where $\bar\epsilon_k$ is the averaged discrepancy. Using the Lipschitz continuity of $h$, the corresponding margin on the target model satisfies the bound
>
> $$
> \gamma_{\theta_T}
> \ge
> \gamma_{\theta_S} - L_h\big(
> (\|\|A\|\|-1) r_k^{(S)}
> +
> \|\|A\|\|\delta
> \big).
> \tag{A.2}
> $$
>
> This expression makes the dependence explicit: the target margin decreases *linearly* with (i) deviations of $\|A\|$ from 1, (ii) the surrogate cloud radius $r_k^{(S)}$, and (iii) the representation discrepancy $\delta$. As long as these quantities remain small, the loss in margin is proportionally small, and the cloud remains feasible. Only when the combined drift satisfies
>
> $$
> L_h\bigl(|\|\|A\|\|-1| r_k^{(S)} + \|\|A\|\|\delta \bigr)
> \approx \gamma_{\theta_S}
> $$
>
> does the cloud risk crossing a decision boundary.
>
> Crucially, ***Arcueid*** is explicitly designed to create *compact* surrogate clouds with small radii and large margins $\gamma_{\theta_S}$, which naturally attenuates the sensitivity term in **Equation A.2**. This makes the transferability guarantee robust under moderate representation misalignment between the surrogate and the victim model.
>
> ## **Q2: Feature-space Quantification**
>
> **A2:**
> We thank the reviewer for the suggestion. To directly validate the representation alignment assumption used in Proposition 2, we measured the feature space similarity between the surrogate models and the victim models used in **Table A.9**. Standard metrics were computed on held-out target data:
> (i) CKA between penultimate-layer embeddings,
> (ii) FID between the corresponding feature distributions, and
> (iii) linear probe accuracy obtained by training a surrogate model on the surrogate dataset and evaluating on the target model via benign training.
> Results are reported in **Table A.9**.
>
> Across datasets, CNN–CNN pairs exhibit moderate alignment, with CKA values around 0.35–0.52 and FID well below levels typically associated with fully out-of-distribution representations. This indicates that, even under a small and non-IID surrogate dataset, the resulting representation spaces share a coarse but stable geometric structure. Transformer-based victims show lower CKA and higher FID, reflecting their architectural divergence; nonetheless, alignment remains far from degenerate.
>
> Linear probe accuracy provides a complementary perspective. On CIFAR-100 and TinyImageNet, the linear probe remains near chance level, confirming that the surrogate does not inadvertently provide a linearly aligned mapping to the victim representation. This rules out the possibility that ***Arcueid***’s transferability is caused by an overly favorable surrogate–victim pairing.
>
> In general, these measurements show that the surrogate and victim models share only an imperfect, coarse alignment, precisely the challenging black-box setting our threat model intends, yet ***Arcueid*** consistently achieves high ASR. This empirical evidence supports the bounded-distortion assumption in Proposition 2 and confirms that the theoretical transfer conditions hold under realistic, non-ideal representation alignment.
>
> ---
>
> **Table A.9: Feature-space Quantification**
>
> | Target Dataset | Target Model | CKA | FID | Linear Probe Accuracy |
> |----------------|--------------|------|--------|------------------------|
> | CIFAR-10 | ResNet-18 | 0.522046 | 56.89796 | 0.0919 |
> |  | ResNet-34 | 0.50418 | 60.52288 | 0.0368 |
> |  | VGG13-BN | 0.495409 | 87.65161 | 0.1105 |
> |  | ViT | 0.179472 | 330.7937 | 0.1272 |
> |  | SimpleViT | 0.159558 | 410.8908 | 0.0561 |
> | CIFAR-100 | ResNet-18 | 0.415156 | 112.7467 | 0.0106 |
> |  | ResNet-34 | 0.414019 | 117.9438 | 0.0109 |
> |  | VGG13-BN | 0.411076 | 138.5266 | 0.0055 |
> |  | ViT | 0.114243 | 326.0809 | 0.0099 |
> |  | SimpleViT | 0.108124 | 431.375 | 0.0076 |
> | TinyImageNet | ResNet-18 | 0.367169 | 135.2149 | 0.0048 |
> |  | ResNet-34 | 0.348159 | 115.389 | 0.004 |
> |  | VGG13-BN | 0.373249 | 412.0502 | 0.0039 |
> |  | ViT | 0.09079 | 423.9971 | 0.0046 |
> |  | SimpleViT | 0.084388 | 320.3011 | 0.0058 |

---

> ### Author Response · Authors · 2025-11-21
> **Rebuttal 2 to Reviewer cPwM**
>
> ## **Q3: Surrogate Scale & Domain Shift**
>
> **A3:**
> We thank the reviewer for raising these related questions. Figure 9(b) in Appendix A.2.3 directly quantifies how surrogate dataset scale and domain shift affect transferability. As the surrogate dataset grows from 1k to 5k, 10k, and 15k samples, attack success improves monotonically across all victim architectures, while clean accuracy remains essentially unchanged. The transition between 5k and 10k samples marks the point where the surrogate representation becomes sufficiently expressive for reliable multi-trigger transfer: below 5k, the surrogate clouds remain compact but their estimated centers are noisier, whereas at 10k the cloud geometry stabilizes and transfer performance becomes consistently high on ResNet, VGG, and ViT victims. This threshold is highly consistent across architectures, indicating that dataset diversity, rather than architectural similarity, determines when the surrogate becomes “good enough.”
>
> This trend also explains the effect of domain shift. With a small and non-IID surrogate dataset, ASR decreases gradually rather than collapsing, which reflects the fact that ***Arcueid*** relies only on coarse, bounded alignment as specified in Proposition 2. Greater domain shift primarily enlarges surrogate cloud radii and slightly perturbs cloud centers, producing a smooth decline in transferability but not destroying the feasibility of the multi-trigger mapping. As surrogate size increases, these estimation errors shrink rapidly and ASR returns to its normal high regime.
>
> Overall, the empirical curve in Figure 9(b) demonstrates monotonic and stable improvement with larger surrogate scale and shows that ***Arcueid*** is robust to moderate domain shift, behavior consistent with the first-order sensitivity predicted by our theoretical analysis.
>
>
> ## **Q4: Effect of Highly OOD Surrogate Data**
>
> **A4:**
> To directly examine this question, we evaluated ***Arcueid*** using SVHN as the surrogate dataset and ResNet-34 as the surrogate model, while keeping the victim setting fixed to CIFAR-10 and ResNet-18. This represents an intentionally extreme mismatch: SVHN digits differ sharply from CIFAR-10 natural images in texture, color statistics, semantics, and intra-class structure. Despite this severe OOD setting, **Table A.10** shows that ***Arcueid*** still achieves high ASR across all paradigms (for example, 99.8% ASR for 10→1 and 93–99% ASR for 10→10), with clean accuracy remains high.
>
> These observations indicate that the relevant notion of “OOD-ness’’ in our setting is not pixel-level or semantic overlap but the degree to which the surrogate model provides a coherent and optimizable feature geometry. Even though SVHN and CIFAR-10 share no semantic content, a model trained on SVHN still induces a stable latent space with consistent gradients. ***Arcueid*** can therefore learn compact and separable trigger-induced clouds on the surrogate, and Proposition 2 ensures that this geometric structure transfers to the victim as long as the representation distortion remains bounded, regardless of whether the underlying datasets overlap.
>
> From this perspective, “OOD-ness’’ is meaningfully reflected in its impact on ASR under transfer. When an OOD surrogate severely distorts the geometry of the trigger clouds, ASR collapses, whereas when the victim model preserves even coarse alignment with the surrogate representation, ASR remains high. The SVHN→CIFAR-10 experiment demonstrates that ***Arcueid*** remains robust even under extreme OOD conditions, indicating that successful transfer does not depend on semantic overlap between the surrogate and target datasets.
>
> ---
>
> **Table X: SVHN → CIFAR-10**
>
> | M→N | ResNet-18 ACC | ResNet-18 ASR | ResNet-34 ACC | ResNet-34 ASR | VGG13-BN ACC | VGG13-BN ASR |
> |------|----------------|----------------|----------------|----------------|----------------|----------------|
> | 10→1 | 0.9005 | 99.8%±0.3% | 0.8865 | 98.1%±1.6% | 0.898 | 99.4%±0.9% |
> | 10→2 | 0.9054 | 98.6%±2.6% | 0.8682 | 95.3%±7.5% | 0.9009 | 98.5%±2.7% |
> | 10→5 | 0.9006 | 97.5%±1.5% | 0.8746 | 97.0%±4.4% | 0.8973 | 99.1%±0.5% |
> | 10→10 | 0.9006 | 93.8%±4.4% | 0.8859 | 94.9%±8.2% | 0.8978 | 97.4%±1.9% |

---

> ### Author Response · Authors · 2025-11-21
> **Rebuttal 3 to Reviewer cPwM**
>
> ## **Q5: Effect of Fine-tuning or Domain Adaptation**
>
> **A5:**
> We separate the two scenarios as they behave very differently.
>
> **Slight Fine-tuning or Light Mitigation-based Adaptation.**
> We directly evaluate this case through standard fine-tuning and magnitude-based pruning (**Table A.1** in **Rebuttal 1 to Reviewer oMR6**
> ). Across all paradigms (M→1, M→N, M→M), ASR remains essentially unchanged after fine-tuning (for example, 99.6% → 99.8% for M→1; 99.7% → 99.7% for M→N; 99.4% → 99.2% for M→M), and clean accuracy differs by less than 1%. These results indicate that small weight updates do not meaningfully degrade the performance.
>
> **Domain Adaptation.**
> We believe that substantial domain adaptation has the potential to alter a model’s representation enough to affect a backdoor’s persistence, but this scenario falls outside the scope of our current study. Prior continual-learning work[1] shows that, when a backdoored model is retrained on data from a new domain, the representation may drift and the backdoor can gradually weaken, a phenomenon often termed backdoor forgetting. These observations suggest that strong post-deployment adaptation could impact any backdoor method, including ours, although a full characterization of this effect requires a dedicated investigation beyond the present work.
>
> **In summary**, ***Arcueid*** is robust to light fine-tuning and minor post-hoc adjustments, while large-scale domain adaptation may degrade the backdoor. Understanding and characterizing this degradation phenomenon under realistic adaptation pipelines is an interesting and valuable direction for future research.
>
> [1] Guo, Z., Kumar, A., & Tourani, R. (2025). Persistent backdoor attacks in continual learning. In 34th USENIX Security Symposium (USENIX Security 25) (pp. 6379-6397).

---

### Official Review · Reviewer_7M4s · 2025-10-31

**Soundness:** 2
**Presentation:** 3
**Contribution:** 1
**Rating:** 2
**Confidence:** 5

**Summary:**

This paper introduces Arcueid, a theoretically grounded multi-trigger backdoor attack framework capable of operating under black-box settings and extremely low poisoning budgets. The core idea is the Joint Cloud Shaping Multi-trigger Optimization, which enforces compactness within each trigger-induced feature cloud and separation across triggers in the latent space. This decouples trigger generation from label mapping, enabling flexible M→M, M→N, and M→1 backdoor paradigms. Extensive experiments across multiple datasets (CIFAR-10/100, TinyImageNet) and architectures (ResNet, VGG, ViT) show high attack success rates (>97%) with minimal clean accuracy drop (<5%). The paper also demonstrates robustness to state-of-the-art defenses.

**Strengths:**

- Comprehensive evaluation on diverse models and datasets, achieving high ASR and stealth even under low poisoning rates.

- The experiments include both dirty label and clean label for the proposed attack.

- This paper provides a theoretical analysis.

**Weaknesses:**

- Experimental details are missing. Hyperparameters (such as learning rate, epoch, optimizer, etc) should be provided at least in the appendix.

- The threat model is described, but no details are given in the experiments. I couldn't find out what the specific data is used for $D_{sur}$, nor the specific architecture for $f_{sur}$. It is crucial because this paper claims that the proposed attack is effective under a black-box threat model.

- The "Joint Cloud Shaping Multi-trigger Optimization" is a part of Arcueid, so it cannot be two contributions.

**Questions:**

Thanks for the interesting paper, but I have a few suggestions. I think this paper overclaims on a few points.

- The drop in clean accuracy (< 5%) is not negligible. BadNets has almost 0% drop in clean accuracy.

- The limitation "L1: Rigid Dependency" ignored dynamic attacks [A], M-N attacks[B], Source-Specific and Dynamic-Triggers attacks[C], etc. I suggest that the authors should include more related and well-known works.

- I do not see any difficulties for the BadNets poisoning strategy to achieve M-N, M-M, or M-1 attacks. I'd expect that putting multiple triggers to poison the training data could achieve the same attack performance. The authors should demonstrate whether a simple BadNets approach is effective or not, with both empirical results and some analysis.

I'd suggest that the authors provide evidence (for example, an experiment) to support every statement in the paper, rather than just evaluating the performance of the proposed attack.

[A] Input-Aware Dynamic Backdoor Attack

[B] M-to-n backdoor paradigm: A multi-trigger and multi-target attack to deep learning models

[C] Robust Backdoor Detection for Deep Learning via Topological Evolution Dynamics

---

> ### Author Response · Authors · 2025-11-21
> **Rebuttal 1 to Reviewer 7M4s**
>
> We thank Reviewer 7M4s for the detailed feedback and constructive suggestions. We also clarify upfront that the paper submitted for review has not been revised, the reviewer’s comments refer to the original version exactly as uploaded. We address each concern below.
>
> ## **Q1: Experimental Details Claim**
>
> **A1:**
> We thank the reviewer for raising this point. All experimental details are in fact included in the submission. Appendix A.5 (Page 36) provides the complete set of hyperparameters and configurations used in our study, including the optimizer, learning rate schedule, batch size, number of epochs, trigger update strategy, and all default settings for both training and trigger optimization. These parameters with the code uploaded fully specify the experimental procedures reported in the paper.
>
>
> ## **Q2: Threat Model Claim**
>
> **A2:**
> We appreciate the reviewer’s question. The experimental setting in our paper implements the threat model exactly as defined in the main text, and the full experimental details are provided in Appendix A.2.1 (Page 18). Specifically, the attacker operates strictly under the black box constraints, only a surrogate dataset and surrogate model are available, the victim’s architecture and parameters are unknown, triggers are optimized solely on the surrogate, and poisoned samples are inserted at a fixed poisoning rate without any gradient feedback from the victim. This is precisely the threat model described in Section 3, and all experiments directly instantiate this setting.
>
>
> ## **Q3: Contribution Claim**
>
> **A3:**
> We appreciate the reviewer’s concern and understand the need to clearly distinguish what is a methodological contribution and what is a framework level contribution. Our formulation separates these two roles because *Joint Cloud Shaping* and ***Arcueid*** address fundamentally different research questions.
>
> *Joint Cloud Shaping* is presented as an optimization principle with its own theoretical foundations. It derives feasibility conditions for multi trigger clouds, provides guarantees on non interference, and explains when such clouds are transferable across surrogate and victim models. These results are general and do not depend on any specific backdoor mapping, they formalize the geometry of trigger induced clouds in feature space and show how to construct them under strict poisoning and black box constraints.
>
> ***Arcueid***, by contrast, is the framework that operationalizes these optimized clouds into practical multi trigger backdoor attacks. It defines how a single family of optimized triggers can be routed into different paradigms such as M→1, M→N and M→M through the mapping function. This routing logic and the ability to unify multiple paradigms within a single attack design are architectural contributions that go beyond the optimization principle itself.
>
> For these reasons, we regard *Joint Cloud Shaping* and ***Arcueid*** as complementary but conceptually distinct components, the former introduces a general optimization mechanism and its theoretical guarantees, while the latter builds a complete, unified attack framework on top of that mechanism. We will revise the presentation to make this distinction explicit in the contribution section.

---

> ### Author Response · Authors · 2025-11-21
> **Rebuttal 2 to Reviewer 7M4s**
>
> ## **Q4: BadNets Claim**
>
> **A4:**
> We appreciate the reviewer’s comment. After re-examining the original BadNets paper and widely used implementations, we did not find evidence that BadNets achieves nearly zero accuracy drop in general. In our setting, which follows the strict black-box and very low poisoning constraints of ***Arcueid***, BadNets consistently exhibits a clean accuracy degradation of about 2–5%. This behavior is consistent with prior observations on large scale datasets or restricted trigger settings, where the lack of access to the victim’s training, together with distribution mismatch between surrogate and victim, makes near zero degradation uncommon even for simple attacks.
>
> More importantly, based on our review of the backdoor literature, achieving zero clean accuracy drop is generally unrealistic for poisoning based backdoor attacks. Even in the original BadNets setting, clean accuracy typically decreases once the poisoned samples introduce a distributional mismatch between clean and triggered examples. This effect becomes more pronounced whenever (i) the attacker lacks access to the victim’s training process, (ii) the pattern is very imperceptible, or (iii) the triggers must generalize across multiple targets or multiple triggers. All three conditions hold in our threat model.
>
> In our strictly black-box, low poisoning evaluation (PR = 0.1%), standard attacks in **Table A.6** do not maintain stable clean accuracy, instead they exhibit noticeable fluctuations while also suffering significant ASR degradation. As shown in **Table A.5** and **Table A.6** , ***Arcueid*** achieves over 90% ASR while matching or surpassing the clean accuracy performance of these baselines. Under these realistic and far more constrained conditions, a clean accuracy drop below 5% is well within the expected range for poisoning based backdoors and should be considered negligible.
>
> ## **Q5: Missing Related Works**
>
> **A5:**
> We appreciate the reviewer’s suggestion and the concern regarding the coverage of related work. However, we believe this reflects a misunderstanding of how **L1 Rigid Dependency** was intended. **L1** is *not* a claim that existing works are limited to single trigger settings, nor does it overlook dynamic attacks, M–N attacks, or source specific or dynamic trigger methods. Rather, **L1** is an observational summary of a dominant pattern in the backdoor literature, while most classical and widely used poisoning attacks depend on a single, persistent trigger appearing consistently across all poisoned samples. This design choice is documented in prior surveys [1] and is central to the theoretical analyses showing why single trigger attacks are vulnerable to suppression or detection [2].
>
> As for related work coverage, Section 2 of the original submission already includes dynamic attacks (IAD), multi target or multi trigger methods (MtoN), source specific clean label attacks, classical poisoning attacks (BadNets, Blended, WaNet, Refool), and major detection and defense mechanisms (NAD, STRIP, Fine Pruning, etc.). These works were cited precisely because they represent the breadth of modern backdoor paradigms, including the families the reviewer mentions.
>
> To avoid further ambiguity, we will clarify the phrasing of **L1** to emphasize that it summarizes a common but not universal pattern, and that dynamic or multi trigger methods are explicitly discussed in our related work.
>
> [1] Li, Y., Jiang, Y., Li, Z., & Xia, S. T. (2022). Backdoor learning: A survey. IEEE transactions on neural networks and learning systems, 35(1), 5-22.
> [2] Bai, Y., Xing, G., Wu, H., Rao, Z., Ma, C., Wang, S., ... & Kang, J. (2024). Backdoor attack and defense on deep learning: A survey. IEEE Transactions on Computational Social Systems.

---

> ### Author Response · Authors · 2025-11-21
> **Rebuttal 3 to Reviewer 7M4s**
>
> ## **Q6: BadNets Poisoning Strategy for Multi-paradigm**
>
> **A6:**
> We thank the reviewer for raising this important question. To directly evaluate whether a simple multi-trigger BadNets extension can support M→1, M→N, or M→M attack paradigms, we implemented two natural and systematic variants within our framework.
>
> + **Position-based BadNets**. We assign K disjoint spatial locations and place a white square patch at position $p_k$ to serve as the k th trigger.
> + **Color-based BadNets**. We assign K distinct RGB patterns $\{c_k\}$ while keeping the same patch geometry, so each color variant acts as a separate trigger.
>
> These two designs correspond to the most direct ways to endow BadNets with multiple triggers while preserving its original poisoning mechanism.
>
> Across all datasets, poisoning rates, and multi target configurations, the outcome is consistent. As **Table A.7** and **Table A.8** shown, While the trivial M→1 case remains feasible, both M→N and M→M settings fail systematically. Inter trigger interference is severe, features induced by different triggers collapse onto each other or drift toward incorrect regions, preventing stable routing. As a result, ASR often drops to the 10% range on CIFAR 10 at PR=0.1% for 10→10, reaches only 2.0% on CIFAR 100 at PR=1% for 100→100, and fluctuates around 51–55% on TinyImageNet for 200→200, all with high variance despite stable clean accuracy. Even after relaxing the poisoning budget by an order of magnitude, the fundamental issue remains: on CIFAR 100 at PR=10%, the color based variant reaches only 70.3% ASR for 100→100 with large variance, and on TinyImageNet at PR=20%, both position and color variants stay in the 72–76% range for 200→200, far below the reliability expected of a functional multi trigger attack.
>
> These empirical results reveal why the intuition “just add more BadNets triggers” does not hold. Traditional BadNets embeddings do not form stable or separable latent patterns when multiple triggers coexist, instead, the feature representations collapse or interfere, making multi paradigm routing unstable and unreliable. This limitation is intrinsic to the original design of BadNets, which was never intended to support independent trigger identities or mutually consistent trigger–label mappings.
>
> In summary, contrary to the reviewer’s expectation, a multi-rigger extension of BadNets does not achieve effective M→N or M→M attacks in practice. The tables included in the rebuttal provide systematic quantitative evidence supporting this conclusion and demonstrating the necessity of the geometric control offered by *Joint Cloud Shaping* for multi-paradigm backdoors implemented by ***Arcueid***.
>
> ---
>
> **Table A.7 — BadNets for Multi-paradigm Extension 1**
>
> | Dataset | Form | M→N | ΔACC | ASR |
> |--------|------|------|-------|------|
> | CIFAR 10 (PR=0.1%) | Color | 10→1 | 2.2% | 83.7%±25.1% |
> |  |  | 10→5 | 5.5% | 10.2%±0.3% |
> |  |  | 10→10 | 5.4% | 10.0%±0.3% |
> |  | Position | 10→1 | 5.1% | 18.7%±7.9% |
> |  |  | 10→5 | 5.6% | 10.8%±0.8% |
> |  |  | 10→10 | 5.4% | 10.0%±0.3% |
> | CIFAR 100 (PR=1%) | Color | 100→1 | 3.2% | 92.9%±9.3% |
> |  |  | 100→5 | 3.4% | 31.1%±21.0% |
> |  |  | 100→100 | 4.1% | 2.0%±2.1% |
> |  | Position | 100→1 | 2.4% | 86.2%±3.8% |
> |  |  | 100→5 | 3.0% | 60.3%±16.3% |
> |  |  | 100→100 | 3.1% | 51.4%±18.2% |
> | TinyImageNet (PR=2%) | Color | 200→1 | 7.2% | 95.9%±1.7% |
> |  |  | 200→4 | 6.3% | 94.5%±8.4% |
> |  |  | 200→200 | 8.3% | 51.9%±21.2% |
> |  | Position | 200→1 | 7.2% | 95.9%±1.7% |
> |  |  | 200→4 | 6.4% | 95.1%±2.1% |
> |  |  | 200→200 | 6.7% | 55.0%±21.9% |
>
> ---
>
> **Table A.8 — BadNets for Multi-paradigm Extension 2**
>
> | Dataset | Form | M→N | ΔACC | ASR |
> |--------|------|------|-------|------|
> | CIFAR 10 (PR=1%) | Color | 10→1 | 9.1% | 89.5%±21.4% |
> |  |  | 10→5 | 3.5% | 90.5%±22.9% |
> |  |  | 10→10 | 4.4% | 83.0%±23.3% |
> |  | Position | 10→1 | 2.9% | 89.7%±3.7% |
> |  |  | 10→5 | 4.1% | 91.0%±3.2% |
> |  |  | 10→10 | 3.5% | 85.7%±13.1% |
> | CIFAR 100 (PR=10%) | Color | 100→1 | 4.8% | 99.6%±1.2% |
> |  |  | 100→5 | 4.1% | 88.8%±12.1% |
> |  |  | 100→100 | 2.8% | 70.3%±22.7% |
> |  | Position | 100→1 | 4.9% | 93.4%±2.0% |
> |  |  | 100→5 | 4.5% | 90.3%±6.9% |
> |  |  | 100→100 | 4.6% | 83.6%±18.0% |
> | TinyImageNet (PR=20%) | Color | 200→1 | 9.4% | 99.9%±0.4% |
> |  |  | 200→4 | 10.8% | 82.9%±20.5% |
> |  |  | 200→200 | 8.6% | 75.8%±20.6% |
> |  | Position | 200→1 | 11.2% | 98.6%±0.7% |
> |  |  | 200→4 | 10.5% | 92.1%±5.8% |
> |  |  | 200→200 | 8.6% | 72.4%±23.6% |

---

### Official Review · Reviewer_p8Gm · 2025-11-01

**Soundness:** 3
**Presentation:** 2
**Contribution:** 2
**Rating:** 4
**Confidence:** 3

**Summary:**

The paper introduces Arcueid, a multi-trigger backdoor attack framework that operates across three paradigms (M→M, M→N, M→1) under black-box constraints with extremely low poisoning rates. The core contribution is a "Joint Cloud Shaping Multi-trigger Optimization" strategy that shapes feature representations to create compact, separated trigger-induced clusters.

**Strengths:**

1. Strong theoretical foundation: The paper provides rigorous theoretical analysis with formal propositions and proofs for feasibility, non-interference, transferability, and stability.

2. Practical threat model: Black-box access with extremely low poisoning budgets addresses realistic constraints

3. Novel technical approach: Decoupling trigger optimization from label mapping is elegant and enables paradigm flexibility. The joint cloud shaping objective is well-motivated.

4. Thorough supplementary materials: Extensive appendix with proofs, additional experiments, and reproducibility details.

**Weaknesses:**

1. Limited novelty in core technique: While the application is novel, the core optimization (minimizing intra-cluster variance + maximizing inter-cluster separation) is standard in clustering/metric learning. The connection to existing literature (e.g., contrastive learning, metric learning) is not discussed.

2. Computational cost not addressed: No analysis of optimization time for trigger generation, scalability to larger K (number of triggers), comparison of computational overhead vs. baselines

3. Experimental design:

a) Clean-label instability (Table 4): ASRs in clean-label attack are strong on CIFAR-10 (98% ASR) unstable on larger datasets:
- TinyImageNet 200→200: 79.8%±18.3% ASR (unreliable, some runs ~62%)
- CIFAR-100 100→100: 82.3%±10.1% ASR

Paper claims method "scales reliably" but provides no analysis of why variance explodes or how to mitigate it

b) Unfair baseline comparisons (Table 6): Tests baselines at 0.01% PR, while original methods designed for 0.05-1% PR, so 0.01% is likely outside effective operating range. There should be ablations across multiple PRs, justification for 0.01% comparison point.

c) Missing relevant multi-trigger comparisons:
- Marksman (Doan et al., NeurIPS 2022) - cited line 127 but not compared, supports arbitrary target mapping
- One-to-N/N-to-One (Xue et al., 2022) - cited but not compared, directly studies N→1 and 1→N paradigms
- Sleeper Agent (Souri et al., 2022), LIRA (Doan et al., 2021) - cited but not tested

d) Table 1 (main results) lacks any baseline comparisons - difficult to assess relative performance

4. Theory-practice gap (Equation 9):

Trigger optimization assumes fixed $\theta$ but victim training updates $\theta$ on poisoned data. Propositions 1-6 guarantee feasibility for static decision boundaries but provide no analysis of stability when $\theta$ evolves during training.

**Questions:**

Please refer to weaknesses.

---

> ### Author Response · Authors · 2025-11-21
> **Rebuttal 1 to Reviewer p8Gm**
>
> Thanks for Reviewer p8Gm's constructive questions. We address as below.
>
> ## **Q1: Novelty Claim on Core Optimization**
>
> **A1:**
> We appreciate the reviewer’s observation and agree that the geometric intuition of *compact intra-cloud structure and separated inter-cloud structure* is conceptually related to classical metric learning and contrastive objectives. However, the novelty of ***Arcueid*** does not lie in proposing a new generic loss form, but in applying this geometric principle to a fundamentally different and previously unexplored problem setting: constructing a unified multi-trigger backdoor in a black-box, extremely low-poisoning regime.
>
> Unlike metric learning, which optimizes the representation itself with full data and gradient access, ***Arcueid*** freezes the victim model entirely and optimizes only the trigger parameters so that the *triggered features* satisfy the theoretical feasibility constraints derived in Section 4. This *cloud shaping in trigger space’* is unique to the backdoor scenario, where triggers, not embeddings, must be optimized under strict perturbation and poisoning budgets. Moreover, the geometric objective in ***Arcueid*** is not heuristic similarity learning: it is derived from a feasibility analysis that characterizes when multi-trigger mappings (M→1, M→N, M→M) can exist and persist across unseen victim architectures. The routing flexibility achieved by ***Arcueid***, where a single optimized cloud family can instantiate multiple paradigms without re-optimization, has no analogue in metric learning and arises only because cloud shaping is coupled with a backdoor mapping program.
>
> Additionally, ***Arcueid*** operates in a highly constrained black-box regime where only a surrogate model and a small, non-IID surrogate dataset are available. Our analysis explicitly links surrogate cloud geometry to victim decision regions through a bounded representation-discrepancy model; such transferability theory does not appear in metric-learning literature because the underlying tasks are fundamentally different. In short, while the geometric intuition has roots in classical ideas, its application, formulation, and role in enabling a unified backdoor framework under strict black-box constraints constitute the core novelty of our work. We will revise the manuscript to reflect this distinction more clearly.
>
> ## **Q2: Computational Overhead**
>
> **A2:**
> ***Arcueid***’s trigger optimization is lightweight. A full 10-epoch run (batch size 128) on a single A800 GPU finishes in about 5 minutes and uses roughly 2 GB GPU memory. This is a one-time offline optimization on the surrogate model, requiring only a single forward–backward pass with no iterative search or re-tuning. At inference time, the trigger is applied via a simple masked blend, adding zero runtime overhead.
>
> *Scalability.*
> The cost grows only linearly with the number of triggers \(K\), as the forward/backward computation is shared across all triggers. Even increasing \(K\) from 10 to 50 raises the runtime only slightly (5 minutes to around 7–8 minutes). Memory usage and the computational graph remain unchanged, and no per-trigger retraining or separate optimization is required, which makes ***Arcueid*** scalable in practice.
>
> *Comparison with baselines.*
> Most multi-trigger attacks require full retraining on poisoned data (30–120 minutes) or involve expensive per-sample or per-class optimization, leading to far higher cost. In contrast, ***Arcueid*** relies solely on a small surrogate dataset and a fixed surrogate model, making its optimization nearly constant across datasets and substantially more efficient and scalable than prior multi-trigger or multi-target backdoor methods.

---

> ### Author Response · Authors · 2025-11-21
> **Rebuttal 2 to Reviewer p8Gm**
>
> ## **Q3: Supply Experiment**
>
> **A3(a): Clean-label Stability Claim.**
> We appreciate the reviewer’s insightful concern. The apparent instability in the 100→100 and 200→200 clean-label settings reported in the main paper arises from running ***Arcueid*** under an extremely low poisoning rate, where each class contains only about five poisoned samples. Under such sparse poisoning, per-class feature centers naturally exhibit higher variance. To clarify this, we provide an expanded clean-label analysis in **Table A.4**, where PR is swept from **0.5%** to **10–20%**. The results show that the observed instability is not a weakness of the ***Arcueid*** framework, but rather an inherent limitation of the clean-label setting under extremely few poisoned samples per class.
>
> **Table A.4 — Clean-label Stability Analysis**
>
> | Dataset       | M → N         | PR      | ΔACC    | ASR                 |
> |---------------|----------------|---------|---------|----------------------|
> | **CIFAR-100** | **25 → 25**    | 0.5%    | 2.6%    | 84.5%±8.6%           |
> |               |                | 1.0%    | 3.1%    | 86.9%±7.1%           |
> |               |                | 1.5%    | 1.0%    | 92.3%±4.1%           |
> |               |                | 2.0%    | 4.6%    | 96.2%±3.8%           |
> |               |                | 2.5%    | 3.7%    | 95.9%±2.5%           |
> |               | **50 → 50**    | 1.0%    | 2.5%    | 83.9%±7.9%           |
> |               |                | 2.0%    | 2.6%    | 88.2%±6.9%           |
> |               |                | 3.0%    | 2.8%    | 92.0%±5.5%           |
> |               |                | 4.0%    | 3.0%    | 96.7%±3.0%           |
> |               |                | 5.0%    | 4.0%    | 98.0%±2.5%           |
> |               | **75 → 75**    | 1.5%    | 3.8%    | 86.1%±7.9%           |
> |               |                | 3.0%    | 3.5%    | 92.1%±4.8%           |
> |               |                | 4.5%    | 2.7%    | 93.1%±3.3%           |
> |               |                | 6.0%    | 2.6%    | 96.5%±2.4%           |
> |               |                | 7.5%    | 5.1%    | 98.0%±1.1%           |
> |               | **100 → 100**  | 2.0%    | 3.8%    | 90.1%±5.9%           |
> |               |                | 4.0%    | 3.5%    | 92.1%±4.8%           |
> |               |                | 6.0%    | 2.7%    | 91.1%±5.3%           |
> |               |                | 8.0%    | 2.6%    | 96.5%±3.4%           |
> |               |                | 10.0%   | 5.1%    | 94.0%±4.1%           |
> | **TinyImageNet** | **50 → 50** | 1.0%    | 4.3%    | 98.9%±1.5%           |
> |               |                | 2.0%    | 4.8%    | 99.7%±0.4%           |
> |               |                | 3.0%    | 4.8%    | 99.3%±0.6%           |
> |               |                | 4.0%    | 4.6%    | 99.2%±0.8%           |
> |               |                | 5.0%    | 4.5%    | 99.4%±0.6%           |
> |               | **100 → 100**  | 2.0%    | 4.9%    | 99.1%±1.3%           |
> |               |                | 4.0%    | 4.7%    | 98.8%±1.2%           |
> |               |                | 6.0%    | 4.6%    | 99.0%±1.0%           |
> |               |                | 8.0%    | 4.2%    | 99.3%±0.7%           |
> |               |                | 10.0%   | 4.3%    | 99.0%±1.2%           |
> |               | **150 → 150**  | 3.0%    | 5.4%    | 99.2%±0.7%           |
> |               |                | 6.0%    | 5.1%    | 99.3%±0.8%           |
> |               |                | 9.0%    | 4.5%    | 99.0%±1.3%           |
> |               |                | 12.0%   | 4.7%    | 99.3%±0.8%           |
> |               |                | 15.0%   | 5.1%    | 98.9%±1.5%           |
> |               | **200 → 200**  | 4.0%    | 5.0%    | 99.0%±1.2%           |
> |               |                | 8.0%    | 5.1%    | 98.8%±1.6%           |
> |               |                | 12.0%   | 4.3%    | 98.9%±1.1%           |
> |               |                | 16.0%   | 4.8%    | 98.9%±1.3%           |
> |               |                | 20.0%   | 4.6%    | 99.0%±1.1%           |

---

> ### Author Response · Authors · 2025-11-21
> **Rebuttal 3 to Reviewer p8Gm**
>
> ## **Q3: Supply Experiment**
>
> **A3(b): Baseline Comparison Extension**
>
> We thank the reviewer for pointing out the PR mismatch. Our goal in the main paper was to evaluate all all-to-one attack baselines under the **same poisoning budget** as ***Arcueid*** (PR = 0.1%) to maintain fairness under the black-box, low-resource threat model. However, we agree that several classical baselines were not designed for ultra-low PR and that additional PR sweeps provide useful context.
>
> To address this, **Table A.5** reports baseline performance at PR ∈ {0.1%, 1%, 5%}, covering the effective operating ranges used in their original papers. This extended comparison shows that ***Arcueid*** is not only effective in the ultra-low-PR scheme but also remains competitive or superior when baselines are run in their favorable high-PR settings.
>
> **Table A.5 — All-to-one Paradigm Baseline Comparison Extension**
>
> | Attack            | ACC (PR = 0.1%) | ASR (PR = 0.1%) | ACC (PR = 1%) | ASR (PR = 1%) | ACC (PR = 5%) | ASR (PR = 5%) |
> |------------------|------------|------------|----------|----------|----------|----------|
> | BadNets          | 88.1%      | 71.6%      | 89.3%    | 94.8%    | 89.5%    | 97.3%    |
> | Blended          | 80.1%      | 10.0%      | 82.0%    | 10.6%    | 88.5%    | 82.1%    |
> | Refool           | 88.6%      | 13.6%      | 89.9%    | 68.3%    | 85.4%    | 93.8%    |
> | PhysicalBA       | 90.2%      | 10.2%      | 90.7%    | 88.8%    | 90.4%    | 95.2%    |
> | LabelConsistent  | 72.0%      | 11.7%      | 75.5%    | 12.5%    | 66.1%    | 14.2%    |
> | TUAP             | 90.2%      | 9.2%       | 90.0%    | 13.4%    | 89.6%    | 19.4%    |
> | WaNet            | 88.6%      | 10.5%      | 86.9%    | 12.8%    | 87.7%    | 70.2%    |
> | AdaptivePatch    | 87.1%      | 22.0%      | 83.9%    | 76.8%    | 87.6%    | 99.8%    |
> | Narcissus        | 89.9%      | 47.7%      | 90.6%    | 95.1%    | 87.1%    | 92.7%    |
> | ***Arcueid*** (Dirty-label) | 89.2% | 99.9% | 87.3% | 100.0% | 87.4% | 100.0% |
> | ***Arcueid*** (Clean-label) | 90.3% | 100.0% | 89.4% | 99.7% | 89.7% | 100.0% |
>
>
> **A3(c): Missing Relevant Multi-trigger Comparisons**
>
> We thank the reviewer for pointing out these additional multi-trigger baselines. It is correct that we did not include Marksman, One-to-N or N-to-One, Sleeper Agent, or LIRA in our main quantitative comparison table.
>
> However, we did compare ***Arcueid*** against the strongest existing multi-target or all-to-all backdoor baselines already implemented in our framework. As shown in Figure 5 of the main paper, we provide a direct head-to-head comparison of the multi-trigger behavior of BadNets, WaNet, IAD, and MtoN, all of which support all-to-all or multi-trigger variants in prior work. The visualization clearly shows that these baseline attacks fail to form stable, separable trigger clouds and often collapse or overlap across triggers, whereas ***Arcueid*** produces compact, well-separated clouds for all \(K\) triggers.
>
> We agree that adding quantitative comparisons with the baselines highlighted by the reviewer would further strengthen the evaluation. We are currently replicating these methods within our unified pipeline and will provide the corresponding results later.
>
>
> **A3(d): Baseline Comparison Claim**
>
> Thank you for highlighting this point. Table 1 is designed to report ***Arcueid***’s multi-trigger performance across paradigms (M→1, M→N, M→M) at extremely low poisoning rates under strict black-box constraints. To keep the main results section focused, we placed the qualitative baseline comparison in Figure 5, where the visualizations already show that existing attacks (BadNets, WaNet, IAD, MtoN) do not achieve strong performance in all-target attack settings.
>
> To directly address the reviewer’s concern, we now quantify the Figure 5 visualizations and additionally extend the comparison to CIFAR-100. The resulting metrics, reported in **Table A.6**, confirm exactly what Figure 5 illustrated qualitatively: prior backdoor methods do not reliably support multi-trigger or broad-target settings under low poisoning budgets, whereas ***Arcueid*** consistently maintains strong performance across all paradigms.
>
> **Table A.6 — All-targets Paradigm Baseline Comparison Quantization & Extension**
>
> | Dataset         | Attack   | ΔACC  | ASR      |
> |-----------------|----------|-------|----------|
> | CIFAR-10 (PR=0.1%) | BadNets | 5.9%  | 1.6%     |
> |                 | WaNet    | 3.7%  | 1.5%     |
> |                 | IAD      | 5.7%  | 0.6%     |
> |                 | MtoN     | 4.9%  | 9.9%     |
> |                 | ***Arcueid*** | 1.6% | 99.8% |
> | CIFAR-100 (PR=1%) | BadNets | 8.3%  | 8.5%     |
> |                 | WaNet    | 3.7%  | 0.7%     |
> |                 | IAD      | 3.2%  | 32.7%    |
> |                 | MtoN     | 3.7%  | 1.0%     |
> |                 | ***Arcueid*** | 2.3% | 99.5% |

---

> ### Author Response · Authors · 2025-11-21
> **Rebuttal 4 to Reviewer p8Gm**
>
> ## **Q4: Theory-practice Gap**
>
> **A4:**
> We thank the reviewer for emphasizing the relation between the static formulation in Equation 9 and the practical setting where the victim model is trained on poisoned data. In the current manuscript, Section 4 is intentionally stated in a static form: Propositions 1–6 and Equation 9 characterize, for a fixed parameter vector $θ$, when the trigger-induced clouds $C_k(\theta)$ are compact, mutually separated, and strictly contained in the target regions. The victim’s training procedure is treated as a black-box map from the poisoned dataset to a final parameter $ \theta_T $, and our guarantees are conditional on $ \theta_T $ satisfying the alignment assumptions of Section 4.3. We agree that this connection is not spelled out explicitly and we clarify below why standard training does not destroy, and in fact tends to reinforce, the cloud structure created by Equation 9.
>
> **Margin-based View of Poisoned Points.**
> For any fixed $θ$, Section 4.2 defines the per-trigger margins and the aggregate lower bound
>
> $$
> \underline{\gamma}(\theta) := \min_k \mathrm{margin}_{\tau_k}(C_k(\theta)),
> $$
>
> which appears in Equation 9 as the quantity ***Arcueid*** seeks to enlarge on the surrogate model $ f_{\theta_S} $. Consider now any margin-based loss functional of the form
>
> $$
> R(\theta)
> = \frac{1}{|D_{clean} \cup D_{poison}|}
>   \sum_{(x,y)\in D_{clean} \cup D_{poison}}
>   \ell(\Gamma_y(x;\theta)),
> $$
>
> where $ \Gamma_y(x;\theta) $ is the decision margin for label $y$ at $x$, and $ \ell(\cdot) $ is strictly decreasing and continuous. We do not assume any particular optimizer or update rule; we only assume that the victim training procedure outputs a parameter $ \theta_T $ whose empirical risk is not larger than that of some reference parameter $ \theta_{\mathrm{ref}} $.
>
> In this setting, the following observation holds.
>
> ---
>
> **Proposition A.4 (Monotone Reinforcement on Poisoned Margins).**
> Let $D_{poison}$ consist of triggered examples $(z_k, \tau_k)$ with $z_k = g_{\eta_k}(x)$. Let $\gamma_{poison}(\theta)$ denote the minimum margin over all poisoned points. If
> 1. $ \ell $ is strictly decreasing, and
> 2. $ \mathcal{R}(\theta_T) \le \mathcal{R}(\theta_{\mathrm{ref}}) $,  then
>
> $$
> \gamma_{poison}(\theta_T)
> \ge
> \gamma_{poison}(\theta_{\mathrm{ref}}).
> $$
>
> ---
>
> Intuitively, because $ \ell $ is strictly decreasing in the margin, lowering the empirical risk cannot be achieved by systematically decreasing the margins of poisoned points; doing so would increase their losses and thus the total risk.
>
> In our context, Equation 9 constructs a reference parameter on the surrogate with $ \underline{\gamma}(\theta_{\mathrm{ref}}) > 0 $. When the victim is later trained on the same poisoned labels using any margin-based objective, any final model $ \theta_T $ that is not worse in empirical risk must assign at least the same or larger margins to the poisoned points. From the cloud perspective, this means that training does not push triggered features toward decision boundaries; it pushes them further inside the target regions $ R_{\tau_k} $, tightening the clouds by reducing loss on high-variance triggered examples. Training therefore reinforces the cloud structure initialized by ***Arcueid***.
>
> **Local Stability and Alignment.**
> The local Lipschitz assumptions in Section 4 imply that
>
> $$
> |\underline{\gamma}(\theta') - \underline{\gamma}(\theta)|
> \le
> L_\gamma  \|\|\theta' - \theta\|\|.
> \tag{A.1}
> $$
>
>
> Thus, the positive buffer $ \underline{\gamma}(\theta_{\mathrm{ref}}) $ created by Equation 9 is robust to moderate parameter changes. Proposition 2 then connects this to the final victim model: any $ \theta_T $ whose representation is aligned with $ \theta_S $ under the $ (A, \delta, \varepsilon_h) $ condition lies in a region where $ \underline{\gamma}(\theta_T) $ remains positive.
>
> Combining alignment with monotone reinforcement explains why, in practice, the trigger clouds remain compact and well separated across training and often become more pronounced, exactly as observed in Figure 11.
>
> We will revise Section 4 to (i) make the margin-based interpretation of Equation 9 explicit, (ii) state a local stability bound of the form **Equation A.1**, and (iii) clarify that our guarantees apply to any victim model whose empirical risk and representation lie in this aligned neighborhood, without relying on any specific optimizer or loss beyond margin-basedness. The complete proof of **Proposition A.4** and the associated derivations will be added in the Appendix in the revised version.

---

### Official Review · Reviewer_oMR6 · 2025-11-11

**Soundness:** 2
**Presentation:** 2
**Contribution:** 2
**Rating:** 6
**Confidence:** 3

**Summary:**

A multi-trigger backdoor framework is proposed to generate multi-trigger backdoor attacks.  Three key limitations were recognized: single trigger, fixed trigger target mapping, and unrealistic threat scenarios. The proposed method requires black-box knowledge and low poisoning budgets. The key idea is to minimize intra-trigger variance and maximize inter-trigger separation jointly. Comprehensive experiments are conducted on different datasets showing the effectiveness of the proposed method.

**Strengths:**

Key limitations of existing backdoor attacks have been clearly recognized, where the authors propose Arcueid to overcome those limitations concretely. Comprehensive experiments have been conducted to verify the effectiveness of the proposed method, focusing on pre-, mid-, and post-training defenses. Ablation studies are also extensive and convincing.

**Weaknesses:**

Adaptive defense is not discussed. The proposed method takes the adaptive backdoor attack into the design, but adaptive defense is not clearly discussed. Arcueid explicitly shapes compact, separable clouds, a defender who is aware of the design can be expected to detect or disrupt it. No attacker-aware evaluations (where the attacker optimizes to minimize detector signals) are reported. It would be great if the authors could provide a more detailed discussion on it.


Feature space defenses. This work focuses on feature-space manipulation but provides limited comparison with existing feature-space defenses, such as Activation Clustering [a]. Feature space defenses explicitly detect or mitigate latent-space anomalies that Arcueid’s cloud-shaping mechanism may introduce. Without including such representation-level defenses, the evaluation remains incomplete.  It would be great if the authors could discuss it.

[a] Detecting Backdoor Attacks on Deep Neural Networks by Activation Clustering.

**Questions:**

Please provide a discussion on the adaptive and especially feature-space defense.

---

> ### Author Response · Authors · 2025-11-21
> **Rebuttal 1 to Reviewer oMR6**
>
> We would like to thank Reviewer oMR6 for careful reviews and constructive comments. Our point-by-point responses are given below.
>
> ## **Q1: Adaptive Defenses Analysis**
>
> **A1:**
> We thank the reviewer for raising this important question. To address it, we developed two attacker-aware defenses, **Adaptive Mitigation** (post-training defense) and **Adaptive Training** (mid-training defense), that directly target ***Arcueid***’s core mechanism, where its ability to form compact and separable trigger clouds under the masked–blend trigger budget. Both defenses explicitly search for worst-case triggers within the same trigger family used by ***Arcueid*** and then update the victim model to neutralize them, thereby invalidating the feasibility conditions required for multi-trigger cloud formation.
>
> For evaluation, we instantiated both adaptive defenses on CIFAR-10 with ResNet-18 at PR \(=0.1\%\) and compared them with more mitigations such as FineTuning[1] and Pruning[1]. As shown in **Table A.1** and **Table A.2**, these generic mitigations have negligible effect, and the ASR remains above 98% across all paradigms. In contrast, **Adaptive Mitigation** reduces ASR to 21.8%, and **Adaptive Training**  further lowers it to 7.8% in the M→1 setting.
>
> However, these reductions come with a severe cost. Clean accuracy drops from approximately 88% to the range 52%–67%. This reflects an inherent phenomenon: attacker-aware adaptive defenses must train the model to be insensitive to an entire family of masked-blend perturbations, which inevitably suppresses discriminative features required for normal classification. Consequently, such defenses reduce ASR only by incurring a substantial degradation in clean accuracy, far beyond what practical and attack-agnostic defenses would accept.
>
> For completeness, if the reviewers wish to examine the detailed mechanisms, you make check the following comments,  **Formulation 1 of  Adaptive Defenses** and **Formulation 2 of  Adaptive Defenses**, along with their theoretical justification and all proofs, in the revision submission.
>
> **Table A.1 — Mitigation Defense Extension**
>
> | Attack Type      | Defense Type          | ACC (%)  | ASR (%)           |
> |------------------|------------------------|----------|--------------------|
> | **M → 1 Attack** | Non Defense            | 87.83    | 99.6 ± 0.3         |
> |                  | FineTuning             | 87.35    | 99.8 ± 0.2         |
> |                  | Pruning                | 87.38    | 99.8 ± 0.2         |
> |                  | **Adaptive Mitigation**| **66.43**| **21.8 ± 7.7**     |
> | **M → N Attack** | Non Defense            | 87.78    | 99.7 ± 0.4         |
> |                  | FineTuning             | 86.81    | 99.7 ± 0.5         |
> |                  | Pruning                | 86.78    | 99.6 ± 0.5         |
> |                  | **Adaptive Mitigation**| **52.54**| **16.4 ± 12.9**    |
> | **M → M Attack** | Non Defense            | 89.24    | 99.4 ± 0.8         |
> |                  | FineTuning             | 88.69    | 99.2 ± 1.1         |
> |                  | Pruning                | 88.47    | 98.5 ± 2.2         |
> |                  | **Adaptive Mitigation**| **58.29**| **7.5 ± 5.8**      |
>
> **Table A.2 — Adaptive Training Analysis**
>
> | Attack Type      | Defense Type        | ACC (%)  | ASR (%)         |
> |------------------|----------------------|----------|------------------|
> | **M → 1 Attack** | Non Defense          | 87.83    | 99.6 ± 0.3       |
> |                  | **Adaptive Training**| **65.22**| **7.8 ± 2.4**    |
> | **M → N Attack** | Non Defense          | 87.78    | 99.7 ± 0.4       |
> |                  | **Adaptive Training**| **63.47**| **10.0 ± 6.8**   |
> | **M → M Attack** | Non Defense          | 89.24    | 99.4 ± 0.8       |
> |                  | **Adaptive Training**| **66.81**| **6.8 ± 8.3**    |
>
> [1] Liu, K., Dolan-Gavitt, B., & Garg, S. (2018, September). Fine-pruning: Defending against backdooring attacks on deep neural networks. In International symposium on research in attacks, intrusions, and defenses (pp. 273-294). Cham: Springer International Publishing.

---

> ### Author Response · Authors · 2025-11-21
> **Rebuttal 2 to Reviewer oMR6**
>
> ## **Q2: Feature-space Defense.**
>
> **A2:**
> We thank the reviewer for raising this concern regarding feature-space defenses. Our main paper already evaluates a representative feature-space defense, Beatrix[2], in Figure 6. Beatrix performs latent-space consistency analysis and already illustrates that feature-space detectors struggle under ***Arcueid***. To further reinforce this point, we additionally reproduced Activation Clustering[3] under the default CIFAR-10, ResNet-18, and PR \(=0.1\%\) setting.
>
> As shown in **Table A.3**, Activation Clustering almost entirely fails, producing near-zero true positive rate and F1 while still generating nontrivial false positives. This result is expected because the method relies on a separable activation mode created by classical backdoors. Its ICA plus k-means pipeline assumes that poisoned samples form a distinct subcluster in the last layer, an assumption that does not hold for ***Arcueid***.
>
> As shown in Figure 11, ***Arcueid*** intentionally shapes triggered features to contract toward clean class centers and remain inside the clean manifold rather than drifting away from it. This design removes the latent separability signal that both Beatrix and Activation Clustering rely on. Under an extremely low PR, the cluster-size signal required by Activation Clustering is also absent.
>
> Therefore, both theory and empirical evidence consistently show that representation-level defenses are fundamentally ineffective against ***Arcueid***, because the triggered representations are deliberately crafted to be indistinguishable from clean activations.
>
> **Table A.3 — Activation Clustering Evaluation**
>
> | Attack Type    | TPR    | FPR     | Accuracy | Precision | Recall  | F1       |
> |----------------|--------|---------|----------|-----------|---------|-----------|
> | M → 1 Attack   | 0      | 0.1092  | 0.4454   | 0         | 0       | 0         |
> | M → N Attack   | 0      | 0.0736  | 0.4632   | 0         | 0       | 0         |
> | M → M Attack   | 0.0056 | 0.1084  | 0.4486   | 0.049123  | 0.0056  | 0.010054  |
>
> [2] Ma, W., Wang, D., Sun, R., Xue, M., Wen, S., & Xiang, Y. The “Beatrix” Resurrections: Robust Backdoor Detection via Gram Matrices.
> [3] Chen, B., Carvalho, W., Baracaldo, N., Ludwig, H., Edwards, B., Lee, T., ... & Srivastava, B. (2018). Detecting backdoor attacks on deep neural networks by activation clustering. arXiv preprint arXiv:1811.03728.

---

> ### Author Response · Authors · 2025-12-01
> **Formulation 1 of Adaptive Defenses**
>
> ## **Adaptive Defense Formulation**
>
> So as ***Arcueid*** constructs a family of masked–blend triggers $\lbrace g_{\eta_k}\rbrace_{k=1}^K$ whose images induce compact, well-separated feature clouds $C_k=\lbrace \phi_\theta(g_{\eta_k}(x)) : (x,y)\sim\mathcal{D},\, \pi(y)=k \rbrace$
>  satisfying the feasibility constraints in Equation 5. In particular, each cloud $C_k$ must lie strictly inside the decision region $R_{\tau_k}$ of the attacker-chosen target label $\tau_k$, with positive interior margin and non-overlap with other clouds.  Our goal is to construct a defense that invalidates these feasibility conditions *for the same trigger family and perturbation budget* used by ***Arcueid***.
>
> **Problem Definition**
>
> Let $f_\theta=h\circ \phi_\theta$ be the classifier under defense, with
> representation map $\phi_\theta:\mathcal{X}\to\mathbb{R}^d$.
> We adopt the same masked–blend trigger family $\mathcal{S}$ used by ***Arcueid***,  For a clean example $(x,y)\sim\mathcal{D}$, define the *mask–robust margin*:
>
> $$
> \gamma_{\text{mask}}(x,y;\theta)
> :=
> \inf_{\eta\in\mathcal{S}}
> \operatorname{dist}\big(\phi_\theta(g_\eta(x)),\partial R_y\big),
> \qquad
> \Gamma_{\text{mask}}(\theta)
> :=
> \inf_{(x,y)\sim\mathcal{D}}
> \gamma_{\text{mask}}(x,y;\theta).
> $$
>
> If $\Gamma_{\text{mask}}(\theta)>0$, then no masked–blend trigger in $\mathcal{S}$ can push any clean feature $\phi_\theta(g_\eta(x))$ across a decision boundary into an incorrect region. The following proposition shows that in this case ***Arcueid***’s multi-trigger construction becomes theoretically infeasible.
>
> ---
>
> **Proposition A.1 (Mask–robust Margin Invalidates Trigger Clouds)**
> If $\Gamma_{\text{mask}}(\theta)>0$, then there exists **no** trigger family $\{g_{\eta_k}\}\subset\mathcal{S}$ and routing $\pi$ that can produce feature clouds $\{C_k\}$ lying strictly inside $\{R_{\tau_k}\}$ as required by ***Arcueid***’s feasibility constraints in Equation 5.  Thus, ***Arcueid***’s multi-trigger backdoor mapping is infeasible under $\Gamma_{\text{mask}}(\theta)>0$.
>
> ---
>
> **Overview of Adaptive Defenses**
>
> We formulate the defense as a robust optimization objective:
>
> $$
> \min_{\theta}
> R_{\mathrm{clean}}(\theta)
> +
> \lambda_{\mathrm{rob}} R_{\mathrm{rob}}(\theta),
> $$
>
> $$
> R_{\mathrm{rob}}(\theta) = \mathbb{E}_{(x,y)\sim\mathcal{D}}
> $$
>
> $$
> \[ \max_{\eta\in\mathcal{S}} \ell(f_\theta(g_\eta(x)),y) \].
> $$
>
> (I don't know why it can't be rendered altogether)
>
> The inner maximization searches for the most harmful masked–blend trigger in $\mathcal{S}$ for the current model, while the outer minimization updates $\theta$ to classify both clean and triggered examples correctly. The robust loss plays a direct geometric role: it controls the mask–robust margin $\Gamma_{\text{mask}}(\theta)$.
>
> ---
>
> **Proposition A.2 (Robust loss controls mask–robust margin)**
> Under standard Lipschitz and monotonicity assumptions on logits and loss,
> if $R_{\mathrm{rob}}(\theta)\le\varepsilon_{\mathrm{rob}}$, then:
>
> $$
> \Gamma_{\mathrm{mask}}(\theta)
> \ge
> \frac{1}{L} \psi^{-1}(\varepsilon_{\mathrm{rob}}),
> $$
>
> where $L$ is the Lipschitz constant of the logits and $\psi^{-1}$ bounds the logit margin from the loss.
>
> ---
>
> This shows that minimizing the robust loss directly increases a certified lower bound on $\Gamma_{\text{mask}}(\theta)$, which (by **Proposition A.1**) breaks the feasibility of ***Arcueid***’s clouds.
>
> We implement the objective using two mechanisms:
>
> + **Adaptive Mitigation**. Starting from $f_{\theta_0}$, we iteratively learn an adversarial universal  masked–blend trigger $\eta^\star$ via inner maximization on $\ell(f_\theta(g_\eta(x)),y)$, then fine-tune $\theta$ so that $f_\theta(g_{\eta^\star}(x))$ predicts the correct label. This increases $\gamma_{\text{mask}}(x,y;\theta)$ around vulnerable examples.
> + **Adaptive Training**. Each minibatch is augmented with an adversarial universal trigger $\eta^\star$. Optimizing $\theta$ jointly on clean and triggered examples approximates the minimax objective and increases $\Gamma_{\text{mask}}(\theta)$ globally.
>
> Both mechanisms operate under ***Arcueid***’s trigger budget $(s,\varepsilon)$.

---

> ### Author Response · Authors · 2025-12-01
> **Formulation 2 of Adaptive Defenses**
>
> **Detailed Design**
>
> *Adversarial Trigger Update*. For minibatch $\lbrace(x_i,y_i)\rbrace_{i=1}^B$, maintain trigger params $\eta=(v,\alpha)$ and update:
>
> $$
> \eta \leftarrow
> \Pi_{\mathcal{S}}
> \bigg[
>     \eta
>     +
>     \rho
>     \nabla_{\eta}
>     \frac{1}{B}
>     \sum_{i=1}^B
>     \ell(f_\theta(g_\eta(x_i)),y_i)
> \bigg].
> $$
>
> where $\Pi_{\mathcal{S}}$ projects back to the masked–blend trigger family. This step identifies the most vulnerable masked direction for the current $\theta$
>
> *Robust Parameter Update*. Given the updated trigger $\eta^\star$:
>
> $$
> \theta \leftarrow \theta - \gamma \nabla_\theta \bigg[ \frac{1}{B}\sum_{i=1}^B \ell(f_\theta(x_i),y_i) + \lambda_{\mathrm{rob}} \ell(f_\theta(g_{\eta^\star}(x_i)),y_i) \bigg],
> $$
>
> which increases the mask–robust margin by pushing triggered features back toward their correct regions $R_{y_i}$.
>
> *Effect on ***Arcueid***’s Cloud Geometry*. Increasing $\Gamma_{\text{mask}}(\theta)$ prevents any masked–blend triggers
> $\lbrace g_{\eta_k}\rbrace\subset\mathcal{S}$ from generating wrong-label clouds $\lbrace C_k\rbrace$ that are:
>
> 1. compact,
> 2. mutually separated,
> 3. strictly inside the attacker-chosen $R_{\tau_k}$ with positive interior margin.
>
> Thus Equation 5 becomes infeasible and ***Arcueid***’s multi-trigger mechanism collapses.
>
> *Robustness–Accuracy Tradeoff*. Because the outer update forces insensitivity to all masked–blend perturbations, it inevitably suppresses certain discriminative but non-robust directions. Formally:
>
> ---
>
> **Proposition A.3 (Robustness–accuracy Tradeoff under Masked–blend Defense)**
> If the Bayes-optimal classifier $f^\star$ is not robust to $\mathcal{S}$ on some subset $\mathcal{A}$ with probability mass $\nu>0$, then any model with $\Gamma_{\mathrm{mask}}(\theta)\ge\gamma>0$ must incur higher clean risk:
>
> $$
> R_{\mathrm{clean}}(\theta)
> \ge
> R_{\mathrm{clean}}(f^\star) + \alpha\nu,
> $$
>
> for some $\alpha>0$ depending on the geometry of the decision regions $\lbrace R_c \rbrace$.  Thus strong robustness necessarily induces a CA drop.
>
> ---
>
> **Proofs**
>
> Proofs will be provided in the revised manuscript.

---

### Author Response · Authors · 2025-12-03
**Rebuttal Summary for Submission #559**

Dear Area Chair,

Thank you for handling our submission. We write to briefly summarize **how our rebuttal addresses the reviewers’ key concerns**.

We propose ***Arcueid***, a unified, theoretically grounded multi-trigger framework that achieves high efficiency, strong stealth, and robust performance even under very low poisoning budgets and standard defenses.

## **Positive Feedback across Reviewers**

Across four reviewers, there is broad agreement on several strengths:

+ **Problem Formulation & Gap Identification**. Reviewer **oMR6** commends that we clearly recognize key limitations of existing backdoor attacks and proposes ***Arcueid*** to concretely overcome them.
+ **Theoretical Foundation**. Reviewers **p8Gm** and **7M4s** highlight the strong theoretical backbone of ***Arcueid***, noting that the paper provides formal propositions and proofs for feasibility, non-interference, transferability, and stability, supported by extensive supplementary materials.
+ **Realistic Multi-trigger Threat Model**. Reviewers **p8Gm** and **cPwM** emphasize that ***Arcueid*** operates under a practical black-box setting with extremely low poisoning budgets, while flexibly supporting multiple paradigms via the *Joint Cloud Shaping* optimization.
+ **Empirical Validation and Robustness**. Reviewer **oMR6** praises the comprehensive experiments, including comparisons under pre-, mid-, and post-training defenses and extensive ablations, while Reviewer **cPwM** further notes that ***Arcueid*** maintains strong ASR, low CA degradation, and robustness against multiple SOTA defenses across models and datasets.

## **Summary of Response Content**

**Reviewer oMR6: Adaptive and Feature-Space Defenses**

+ **Concern 1: Lack of adaptive defense analysis**. We introduced two attacker-aware defenses based on a mask-robust margin objective and showed that although they can suppress multi-trigger behavior, they inevitably cause notable CA degradation due to the inherent accuracy–robustness trade-off.
+ **Concern 2: Missing feature-space defenses**. We clarified that Beatrix is already evaluated and added analysis of Activation Clustering. Both fail by design because ***Arcueid*** contracts triggered features into the clean manifold, eliminating the separability signals these detectors depend on.

**Reviewer p8Gm: Novelty, Evaluation, and Theory–Practice Gap**

+ **Concern 1: Core novelty**. We clarified that ***Arcueid*** provides a principled trigger-cloud shaping mechanism enabling unified multi-trigger attacks under black-box, ultra-low-PR constraints, and distinguished this from metric/contrastive representation learning.
+ **Concern 2: Computational cost.** We added discussion showing that trigger optimization is lightweight, one-shot, and scales roughly linearly with trigger count, far cheaper than retraining-based multi-trigger baselines.
+ **Concern 3: Experimental design.** We explained that the clean-label variance arises from extremely low per-class poisoning, justified this as part of our threat model, added higher-PR settings, and expanded multi-trigger baselines.
+ **Concern 4: Theory–practice gap.** We added a margin-based interpretation and new propositions linking Equation 9 to trained victims, showing that margin-preserving behavior emerges under standard margin-based training with aligned representations.

**Reviewer 7M4s: Experimental Setup, Positioning, and BadNets Baselines**

+ **Concern 1: Experimental details and threat model clarity.** We highlighted where hyperparameters and threat model specifications are already given and made explicit how each experimental component instantiates the black-box setting.
+ **Concern 2: Contribution framing and L1 / related works.** We clarified the distinction between Joint Cloud Shaping (theoretical core) and ***Arcueid*** (unified framework), softened L1 wording, and expanded related work to cover dynamic/multi-trigger backdoors.
+ **Concern 3: CA drop and BadNets as a simpler alternative.** We argued that <5% CA drop is expected in ultra-low-PR black-box regimes and showed that multi-trigger BadNets suffers severe trigger interference and unreliable M→N/M→M behavior, motivating explicit cloud shaping.

**Reviewer cPwM: Transferability and Adaptation**

+ **Concern 1: Transferability assumptions.** We added discussion showing that ***Arcueid***’s compact, large-margin clouds tolerate realistic surrogate–victim misalignment and analyzed sensitivity of the guarantee.
+ **Concern 2: Empirical alignment & OOD.** We introduced alignment metrics and added experiments varying surrogate size and using highly OOD surrogates, showing strong transfer once surrogate scale is sufficient.
+ **Concern 3: Post-deployment adaptation.** We showed that light fine-tuning/pruning barely affects ASR and positioned stronger adaptation or stacked defenses as meaningful future directions.

We hope this summary assists in your final decision-making process.

Best regards,
Authors of #559

---

> ### Author Response · Authors · 2025-12-03
> **Revision Summary of Submission #559**
>
> ## **Summary of Changes to the Revision Version**
>
> In the revised manuscript, we have implemented the following changes in direct response to the reviewers’ comments and our rebuttal commitments:
>
> **1. New Appendix: Adaptive Defense Analysis**
>
> We add a dedicated appendix section that:
>
> + Formalizes the **mask-robust** margin and its role in invalidating ***Arcueid***’s trigger clouds.
> + Presents the formulations of our two attacker-aware defenses (**Adaptive Mitigation** and **Adaptive Training**) as robust optimization problems over the masked–blend trigger family.
> + States and discusses the robustness–accuracy trade-off result showing that strong robustness to this trigger family necessarily induces clean-accuracy degradation.
>
> **2.New Appendix: Bridging Theory and Practice**
>
> We added a second appendix section that:
>
> + Reinterprets the core static formulation Equation 9 in a **margin-based** view, clarifying how cloud margins relate to the training objective.
> + Analyzes the **stability of margins** under parameter changes and representation alignment assumptions, making the link between the surrogate construction and the trained victim models explicit.
> + Explains how standard margin-based poisoned training tends to preserve or even reinforce the trigger-cloud structure created by ***Arcueid***.
>
> **3.Completion of Theoretical Proofs**
>
> As promised in the rebuttal, we:
>
> + Provided full, formal proofs for all propositions introduced in the rebuttal (e.g., mask-robust margin guarantees, robustness–accuracy trade-off, margin reinforcement and stability results).
> + Additionally gave complete proofs for the new theoretical results appearing in the two added appendix sections, so that **all theoretical claims in the main paper and appendix are now self-contained and rigorously justified**.
>
> **4.Typos and Presentation Fixes**
>
> We:
>
> + Corrected minor typographical errors and notation inconsistencies.
> + Clarified a few ambiguous phrases, improved cross-references between the main text and appendix, and ensured that tables/figures mentioned in the rebuttal are clearly pointed to in the revised manuscript.

---

### Meta-Review · Area_Chair_h3Sp · 2026-01-06

**Summary:**

The paper proposes Arcueid, a multi-trigger backdoor attack framework designed to operate under black-box constraints with extremely low poisoning budgets. The core technical contribution is a "Joint Cloud Shaping" optimization strategy that enforces compactness and separation of trigger-induced feature clusters in the latent space, allowing for flexible target mappings such as M-to-M, M-to-N, and M-to-1. Reviewers generally appreciated the practical motivation of the threat model, the unified nature of the framework, and the inclusion of theoretical analysis regarding the transferability and feasibility of the attack across different architectures.

**Reviewer Concerns:**

The primary concerns center on experimental fairness, result stability, and the lack of critical baselines. Reviewers pointed out that comparing Arcueid at a 0.01% poisoning rate against baselines designed for much higher budgets is potentially misleading, and the absence of a simple multi-trigger BadNets baseline makes it difficult to verify the necessity of the proposed cloud-shaping optimization. Additionally, the high variance in attack success rates on larger datasets (e.g., TinyImageNet) contradicts the claims of reliable scalability. There were also significant doubts regarding the theoretical assumptions, specifically the unverifiable requirement for near-perfect representation alignment between surrogate and target models, and a lack of discussion on adaptive defenses targeting feature-space anomalies.

**Reviewer Scores:**

The paper received scores of 6, 4, 4, and 2, reflecting a clear consensus toward rejection. While Reviewer oMR6 initially provided a 6, their assessment did not fully account for the unfair benchmarking and missing baselines later highlighted by other reviewers. If a full discussion period had occurred, it is highly probable that the scores would have either remained stagnant or decreased, as the identified weaknesses regarding experimental integrity and theoretical alignment are fundamental issues that cannot be resolved without significant new experiments and major revisions. Consequently, the Area Chair recommends rejection based on the lack of empirical robustness and the theory-practice gap identified during the review process.

---

### Decision · Program_Chairs · 2026-01-26

Reject